



# Application of a new distributed hydrological model based on soil–gravel structure in the Niyang River Basin, Qinghai–Tibet Plateau

Pengxiang Wang[1,2,3], Zuhao Zhou[1*], Jiajia Liu[1], Chongyu Xu[3,7], Kang Wang[3], Yangli Liu[4], Jia Li[5], Yuqing Li[6], Yangwen Jia[1], Hao Wang[1]

[1]State Key Laboratory of Simulation and Regulation of Water Cycle in River Basin, China Institute of Water Resources and Hydropower Research, Beijing 100038, China
[2]China Three Gorges Corporation, Beijing 100038, China
[3]State Key Laboratory of Water Resources and Hydropower Engineering Science, Wuhan University, Wuhan 430072, China
[4]China Power Construction Group Guiyang Engineering Corporation Limited, Guiyang 550081, China
[5]Bureau of South to North Water Transfer of Planning, Designing and Management, Ministry of Water Resources, Beijing 100038, China
[6]Department of Water Resources and Civil Engineering, Tibet Agriculture and Animal Husbandry College, Nyingchi 860000, China
[7]Department of Geosciences, University of Oslo, Oslo, Norway

*Correspondence to*: Zuhao Zhou (zhzh@iwhr.com)

**Abstract.** The Qinghai-Tibet Plateau has a thin soil layer on top of a thick gravel layer, while its unique geological structure, combined with the snow and frozen soil in this area, significantly affect water circulation in the entire region. To investigate the mechanism of the underlying surface structure on the hydrothermal migration and water circulation process in the Qinghai-Tibet Plateau, we performed comprehensive study combining field experiments of the water and heat transfer processes and development of a Water and Energy transfer Processes model in the Qinghai–Tibet Plateau (WEP-QTP), based on the original Water and Energy transfer Processes model in Cold Regions (WEP-COR). The Niyang River Basin located on the Qinghai-Tibet Plateau was selected as the study area to evaluate the agreement between theoretical hypothesis, observation, and modeling results. This model divides the single soil into a dualistic structure of soil and gravel. In the non-freeze–thaw period, two infiltration models based on the dualistic soil-gravel structure were developed using the Richards equation in a non-heavy rain scenario and the multi-layer Green-Ampt model in a heavy rain scenario. During the freeze–thaw period, a hydrothermal coupling model based on the continuum of the snow-soil-gravel layer was constructed. The addition of the gravel layer corrected the original model's overestimation of the moisture content below the surface soil and reduced the moisture content relative error (RE) from 33.74 to -12.11%. The addition of the snow layer not only reduced the temperature fluctuation of the surface soil, but also, with the help of the gravel layers, revised the original model's overestimation of the freeze-thaw speed. The temperature RE was reduced from -3.60 to 0.08%. In the non-freeze-thaw period, the dualistic soil-gravel structure improved the regulation effect of groundwater on flow, stabilizing the flow process. The maximum RE at the flow peak and valley decreased by 88.2 and 21.3%, respectively. In the freeze-thaw period, by considering the effect of the snow-soil-gravel layer continuum, the freezing and thawing process of WEP-QTP lagged that of WEP-COR by approximately one month. The groundwater simulated by WEP-QTP had more time to recharge the river,



which better showed the "tailing" process after October. The flow simulated by the WEP-QTP model was more accurate and closer to the actual measurements, with Nash > 0.75 and |RE| < 10 %. The improved model reflects the effects of the Qinghai-Tibet Plateau special environment on the hydrothermal transport and water cycle process and can be used for hydrological simulation of the Qinghai-Tibet Plateau.

## 1 Introduction

The Qinghai-Tibet Plateau (QTP), known as the "Asian Water Tower," is a typical cold mountainous area with low latitude and high altitude. This region has a unique geology and landform, is sensitive to climate change (Liu et al., 2019), and plays an important role in ensuring the security of water resources in China and Southeast Asia. The impact of the extensive glacier, snow cover, and permanent and seasonal frozen soil on the water circulation processes of the entire area cannot be ignored. On the surface of seasonal frozen soil and permafrost, seasonal thaw layers alternately freeze and thaw as seasons

change. Almost all ecological, hydrological, soil, and biological activities in the soil in cold regions occur here, hence it has been the focus of hydrological research in cold regions (Chen et al., 2014; Kurylyk et al., 2014). During soil freezing, ice in the seasonal thaw layer blocks most of the pores in the soil, hinders infiltration, and affects water movement in the soil (Cheng and Jin, 2013). As for the hydrological cycle, the frozen soil layer prevents rainfall and snowmelt infiltration, forcing surface runoff down the slope, which may lead to severe flash floods. In addition, it affects the quantity of groundwater

recharge supplemented by infiltration and the distribution ratio of surface runoff between rivers and lakes (Ireson et al., 2013; Larsbo et al., 2019).

The simulation accuracy of the water and heat transport in the seasonal thaw layer directly affects the evaluation of water resources in this area and the analysis of the runoff evolution. However, soil water and heat transfer are relatively complex and influenced by several factors (Watanabe and Kugisaki, 2017) such as soil structure (Dai et al., 2019; Franzluebbers,

2002) and heat conduction under snow cover (Lundberg et al., 2016). Soil mixed with gravel has different hydraulic and thermodynamic properties compared with soil alone, which is affected by the sand and gravel content (Zhang et al., 2011). When this content is low, the gravel changes the soil structure and increases the distance of soil water movement; the saturated hydraulic conductivity of soil decreases as the gravel content increases (Childs and Flint, 1990; Mehuys et al., 1975). When the sand and gravel content exceeds a certain level, connected macropores are formed in the soil, and the soil's

saturated hydraulic conductivity increases along with the content (Beibei et al., 2009). In the heat transfer process, the greater thermal conductivity and heat capacity of gravel compared with those of dry soil affect the geothermal flux (Yi et al., 2013).

During the continuous QTP uplift, a series of ascending areas (denuded areas) and descending areas (deposited areas) have been formed. Quaternary deposits are generally thinner in denuded areas and thicker in deposited areas (valleys, plain). The

thickness of Quaternary deposits varies greatly at the transition between the denuded and deposited areas controlled by the fault zone. In the QTP Quaternary sediments, there are many gravel and rock fragments due to the surface uplift and





collision of the Indian plate with the Eurasian plate (Arocena et al., 2012; Sun, 1996). In addition, under strong freeze-thaw conditions in the cold plateau region, the humus accumulation of herbaceous plants is slow, while the decomposition of minerals is weak, resulting in slow soil development on the surface of Quaternary deposits and thin soil layers, which is
prevalent in the QTP (Deng et al., 2019; Yang et al., 2009; Chen et al., 2015).

The existing hydrological models in cold regions, such as the SHAWDHM model (Zhang et al., 2013), GEOtop model (Rigon et al., 2006), Cold Regions Hydrological Model (CRHM) (Pomeroy et al., 2007), and Variable Infiltration Capacity (VIC) model (Cherkauer and Lettenmaier, 2003), consider the process of water and heat transport in the soil in cold regions and can simulate the water cycle processes in general cold regions to a certain extent. However, these models defined the
simulated object of hydrothermal coupled transport process as a one-dimensional homogeneous medium, while ignoring the upper and lower layered structure of the QTP. The geological features of the QTP are generally thin soil layers above the thick gravel layers with clear boundaries between them. Compared with soil, gravel layers have different porosity and density as well as water and thermal properties, which have a greater effect on hydraulic conductivity and water retention curve. By adjusting the model parameters, the hydrothermal simulation effect of the QTP can be improved to a certain extent
(Zhu and Shao, 2010; Pan et al., 2017). However, there remain some errors in the simulation, and it is difficult to objectively reflect the influence of the upper and lower geological structures on hydrothermal migration and the hydrological cycle.

Therefore, it is important to consider the influence of the dualistic soil–gravel structure on the hydrothermal coupling and flow simulation of the hydrological model when simulating the hydrology in the QTP. The purpose of this study is to: (1) develop infiltration models based on the dualistic soil–gravel structure in non-heavy and heavy rain scenarios during the
non-freeze–thaw period (all soil/gravel layer temperatures of the calculation unit were greater than 0 ℃, and all of the water was in a liquid state), (2) develop a hydrothermal coupling method based on the continuum of snow–soil–gravel layer through field water and heat monitoring experiments during the freeze–thaw period (the soil/gravel layer temperature of the calculation unit was lower than 0 ℃), and (3) study the effect of the dualistic soil–gravel structure on the hydrological cycle by building the distributed water cycle model (WEP-QTP) for the Niyang River Basin, a tributary of the Yarlung Zangbo
River in the QTP.

## 2 Materials and methods

### 2.1 Study sites and data

#### 2.1.1 Study area

The Niyang River is located on the left bank of the lower reaches of the Yarlung Zangbo River, between 29°28'–30°38' N
and 92°10'–94°35' E in the Linzhi area of southeastern Tibet. It originates from Cuomoliang Mountain on the west side of the Mila Mountain in the Tibet Autonomous Region of China at an altitude of approximately 5,000 m above sea level (a.s.l.). The Niyang River flows through Gongbujiangda County and Bayi Town from west to east and finally flows into the Yarlung


Zangbo River in the Bayi District of Nyingchi City, with a drop of 2,080 m and an average slope drop of 0.73 %. The basin is approximately 230 km long from east to west and 110 km wide from north to south. The area of the watershed is 17,535 km², ranking fourth among the five tributaries of the Yarlung Zangbo River, and its runoff is second only to that of Palungzangbu. The Niyang River Basin is located at the intersection of Tibet's east–west and north–south mountain ranges. The basin sediments consist of greywacke and litharenitewith low-moderate maturities of textures and minerals (Huyan et al., 2022). The terrain in the watershed is complex, with staggered large and small mountains and large elevation fluctuations. The elevation of the river valley is generally 3,000–4,000 m a.s.l. The elevation of most mountain peaks on both sides of the valley is approximately 5,000 m a.s.l, reaching up to 6,870 m a.s.l. The Niyang River Basin belongs to the plateau temperate monsoon climate zone. The multi-year average precipitation is affected by the Indian Ocean tropical ocean monsoon. Under the effect of the Indian low pressure, the southwest monsoon pushes a large amount of warm and humid air from the Bay of Bengal along the Yarlung Zangbo River Valley to the Niyang River Basin, causing precipitation in the basin with heavy rainfall and large vertical changes. The average annual precipitation in the basin is 1,416 mm, and the average annual temperature is approximately 8 °C. Obvious temperature changes occur from east to west with elevation. The frozen soil in the study area is mainly seasonal. Permafrost accounts for approximately 23.65%, mainly distributed in the upper reaches of the basin and the high-altitude areas on both sides the mainstream. The annual average temperature of the experimental site is 5.28 °C, which is a seasonally frozen soil area.

In this study, a monitoring experiment of the hydrothermal coupling process during the freeze‐thaw period was carried out on the mountainside of the Sejila Mountain in the lower reaches of the Niyang River Basin. The longitude and latitude of the study site are 94°21′45″ E and 29°27′12″ N, respectively, and the altitude is 4,607 m a.s.l. The experimental period was 2016–2017, while the freeze–thaw period was from November 2016 to March 2017. The basic situation of the watershed and the location of the experimental points are shown in Figure 1. Before the field experiment, nuclear magnetic resonance was used to calibrate the water and heat transport monitoring instruments under seasonal freezing and thawing soil conditions on the plateau. A working area with a length of 1.0 m, a width of 1.0 m, and a depth of 2.0 m was excavated at the experimental site. A time-domain reflectometry sensor was used to monitor the contents of the liquid water, a PT100 sensor was used to measure the temperature, and a TensionMark sensor was used to measure the potential of the substrate, all of which were installed every 10 cm vertically in the experimental pit to a depth of 1.6 m. Following installation, the pit was backfilled with undisturbed soil and the data were collected automatically.



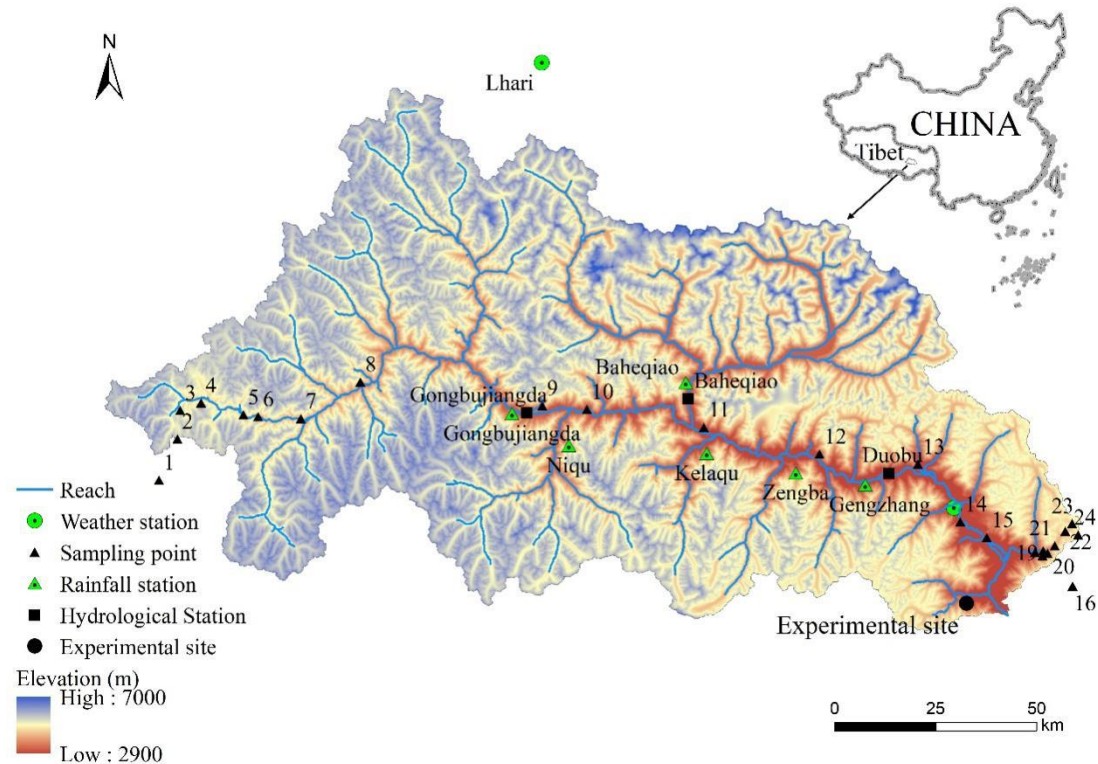


**Figure 1: Basic situation and station distribution of the Niyang River Basin**

**2.1.2 Data description**

The data required for this study were mainly divided into two categories: the first was the data required for model construction (mainly including meteorology, geology and landform, terrain, soil type, land-use type, vegetation index, and

glacier data), while the second was the data used to verify the model results (including historical and experimental monitoring data).

Meteorological data: Temperature, relative humidity, sunshine hours, and wind speed data were collected by the Nyingchi Meteorological Station in the basin and the Jiali Meteorological Station outside the basin, from 1961–2018. The data were obtained from the China Meteorological Data Network (http://data.cma.cn). In addition to the two meteorological stations in

Nyingchi and Jiali, the rainfall data sources also included six rainfall stations (2013–2015) including Gongbujiangda, Gengzhang, Baheqiao, Niqu, Kelaqu, and Zengba in the watershed and the contour map of annual precipitation in the Tibet Water Resources Bulletin (2012–2017). The temperature, precipitation, relative humidity, sunshine duration, and wind speed in the basin were interpolated from the meteorological station data by the reversed distance squared method. Temperature and precipitation were additionally corrected for elevation. The temperature correction factor was -6 °C/km. As for the

precipitation data, the precipitation-elevation relationship was determined according to the contour map of annual precipitation in the Tibet Water Resources Bulletin and the precipitation station data. Then, the daily precipitation data were





obtained in the basin through elevation interpolation (Wang et al., 2017) of the precipitation data from six rainfall stations and two meteorological stations. Avoid precipitation errors at high altitudes caused by altitude limitations of precipitation stations.

Geology and landform: Due to the combined effects of plate tectonics, weathering, and erosion, a unique geological structure was formed in the QTP with a thin soil layer on the top and a thick gravel layer (mixed layer of soil and gravel, and gravel accounts for more than half) on the bottom (Fig. 2). According to the geological characteristics of the QTP, we selected 24 sampling points at different altitudes from the source to the estuary of the Niyang River basin for field investigation on soil texture (Fig. 1). Among them, points 1–16 were along the river, while 17–24 were from the foot of the mountain to the peak.

The soil thicknesses and compositions at the 24 sampling points were measured and analyzed. The soil layer of the Niyang River Basin is mainly sandy loam, with average sand, silt, and clay contents of 55.89, 31.2, and 12.91%, respectively. The gravel layer is mainly round gravel containing pebbles; the gravel content was approximately 50–65%, the clay content was 5–10%, and the pores are filled with medium and fine-grained sand. The thickness of the soil layer gradually decreases from the foot to the peak of the mountain, and0 is approximately 40 cm on the hillside with higher altitude and increases to more

than 100 cm in the valley.

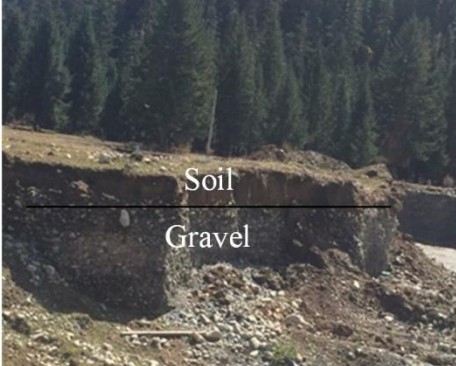
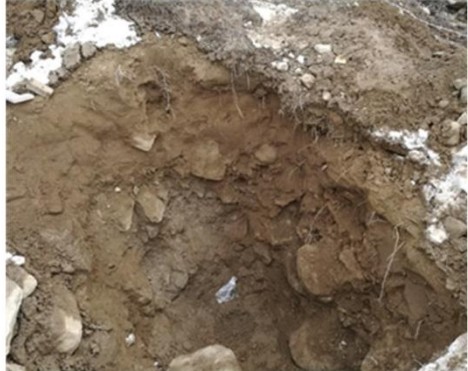

**Figure 2: Soil and gravel structure of the Niyang River Basin**

Terrain data: The elevation data (Digital Elevation Model, DEM) used in this study were obtain from the SRTM90 (Shuttle Radar Topography Mission), jointly measured by the National Aeronautics and Space Administration (NASA) and the

National Imagery and Mapping Agency (NIMA) with an accuracy of 90 m.

Soil date: Soil type data were obtained from the second national soil census and "Chinese Soil Records". Land-use data came from the Resource Environment Science and Database Center, Institute of Geographic Sciences and Natural Resources Research, Chinese Academy of Sciences (http://www.resdc.cn), and the data resolution was 30 m.

Vegetation index: Moderate-resolution Imaging Spectroradiometer (MODIS) data from 2000–2017 were selected as the data

source. Among them, the leaf area index accuracy was 500 m and the normalized difference vegetation index accuracy was 250 m; these were mainly used to calculate evaporation and vegetation interception processes, respectively.



Glacier data: The glacier data included China's second glacier inventory data set (1:100,000) and Landsat TM/ETM+/OLI remote sensing images. The second glacier inventory data comes from the China Cold and Arid Regions Science Data Center (http://westdc.westgis.ac.cn/). Landsat data were obtained from the data sharing platform of the United States
Geological Survey (USGS) (http://glovis.usgs.gov/). ENVI software was used to extract glaciers, and the boundaries of the glaciers were finally determined with reference to Google Earth imagery. According to China's second glacier cataloging rules, the glaciers in the basin were classified and the glacier area was calculated. The volume of the glacier was calculated by the area-volume empirical formula (Grinsted, 2013; Radić and Hock, 2010).

The distribution of the main hydrologically-relevant features of the basin used in the model construction is shown in
Appendix A.

Model verification data included historical data and experimental monitoring data. Historical data included daily measured flow data from the Gongbujiangda Hydrological Station (2013–2016, 2018), the Baheqiao Hydrological Station (2013–2014), and the Duobu station (2013–2018). The experimental monitoring data included the soil temperature and volumetric water content of the experimental site from 2016–2017. A time-domain reflectometry sensor for monitoring the water content of
the liquid water, a PT100 sensor for measuring the temperature, and a TensionMark sensor for measuring the potential of the substrate were installed every 10 cm in the vertical depth of the experimental pit at a depth of 1.6 m. Water, heat, and potential energy were automatically monitored during freezing and thawing.

## 2.2 Model improvement

Based on the Water and Energy transfer Processes in Cold Regions (WEP-COR) model, this study developed the improved
Water and Energy transfer Processes in the Qinghai–Tibet Plateau (WEP-QTP) model. An introduction to the WEP-COR model structure and simulation method is provided in the Appendix B. In contrast to the general cold areas where the WEP-COR model is applied, the widespread dualistic soil–gravel structure in the QTP has a great impact on the water cycle processes in the basin. According to the geological characteristics of the QTP, this study improved the hydrothermal simulation methods of the non-freeze–thaw period and the freeze–thaw period.
The seasonal thaw layers above the impervious boundary, including the aquifer and vadose zone, were investigated (in permafrost regions, the impervious boundary is the permafrost layer). This part contains both soil and gravel layers with a strong freeze-thaw effect, closely interacts with the surface hydrological process, and is the key link affecting the hydrological cycle in this area. In the non-freeze–thaw period, the calculation object of water movement was defined as the dualistic soil–gravel structure (Fig. 3a). The upper layer is soil whose thickness and number of layers are determined by the
location of the calculation unit; the thickness of the soil layer gradually decreases from the foot to the peak of the mountain. The lower layer is the gravel layer (mixed layer of soil and gravel). During the freeze–thaw period, in addition to considering the impact of gravel on hydrothermal transfer, the contribution of snow to thermal insulation and its higher reflectivity to shortwave solar radiation were also considered. A snow layer was added on top of the dualistic soil–gravel structure. The hydrothermal coupling simulation object was defined as the snow–soil–gravel layer continuum (Fig. 3b); among them,


$R_{surface}$ represents runoff from the surface, and $R_i$ represents lateral flow or soil flow in the $i$-th layer of soil. $R_i$ is related to slope and soil moisture content. $E_1$ represents soil evaporation, $E_r$ represents vegetation transpiration, $Q_i$ is gravity drainage of the $i$-th layer, $P$ is precipitation, $T_a$ is atmospheric temperature, $T_i$ is the temperature of the $i$-th layer, and $G_i$ is the heat flux caused by the temperature difference between the $i$-th layer of soil and the adjacent soil layers.

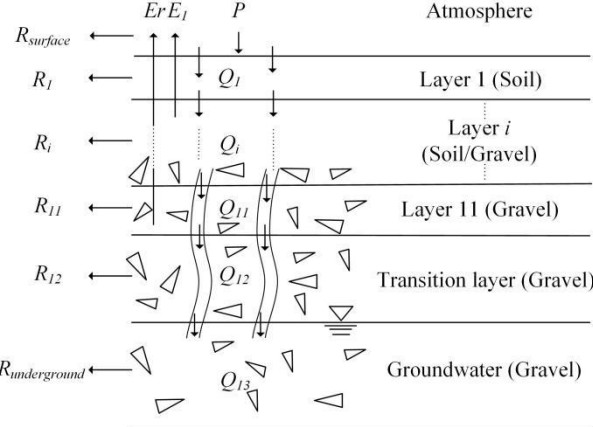


(a)

(b)

**Figure 3: Layered calculation structure of the dualistic "soil–gravel" structure (a) and the "snow–soil–gravel layer" continuum (b)**

The division of soil layer thickness was mainly based on the convenience of accurately simulating the hydrothermal

migration process. As the surface soil is more sensitive to changes in atmospheric temperature, the thicknesses of the first and second layers were set to 10 cm each, while the third through eleventh layers were divided evenly, each with a thickness





---

of 20 cm. The depth from the surface to the impermeable layer was an adjustable parameter in the model that can be adjusted based on the actual basin conditions. Considering the active range of seasonal frozen soil, the depth of the hydrothermal numerical simulation in the model was set 2 m. When the depth of impermeable layer was greater than 2 m, the excess
comprised the transition layer. The lower boundary at the bottom was the impermeable layer, assuming that it maintains a constant temperature (seasonal frozen soil regions > 0 °C, permafrost regions < 0 °C). In regions with seasonally frozen soil, the model considered the entire freezing and bidirectional thawing processes of the soil. For the permafrost regions, the surface layer freezing and thawing processes were considered, while the lower permafrost layer was used as the lower boundary condition.


### 2.2.1 Non-freeze–thaw period

In the non-heavy rain infiltration scenario, the basic equations describing water movement were the same as the WEP-COR model. However, since gravel can neither conduct nor store water, the gravel, which accounts for 50–65% of the gravel layer, hinders the movement of water and affects the water retention curves (Cousin et al., 2003). Therefore, the
revised formula for water retention properties of the soil–gravel mixture was used to describe the lower gravel layer water retention curves (Wang et al., 2013):

$$\frac{\theta_l - \theta_r}{\theta_s - \theta_r} = Ah^{-\lambda}\left(1 - \omega_{gravel}\right) \tag{1}$$

where $A$ is an empirical parameter, $h$ is the matric suction (cm), $\lambda$ is the pore-size distribution parameter ($\lambda < 1$), and $\omega_{gravel}$ is the volume ratio of the gravel in the gravel layer.
In the heavy rain infiltration scenario, the multi-layer Green–Ampt equation (Jia and Tamai, 1998) was used to calculate the infiltration process when the infiltration front (INF) was in the soil layer, which is the same as in WEP-COR. When the INF moves to the interface of the soil and gravel (Layer *itf*), the front movement slows down because the water suction of the gravel layer is less than that of the soil (Mao and Shang, 2010). Until the water has the same potential energy in the soil and the gravel, the INF breaks through the critical surface, and then the infiltration rate stabilizes (Fig. 4). Therefore, a new
multi-layer Green–Ampt model based on the soil–gravel structure was proposed. The stable infiltration rate after INF breaking through Layer *itf* was calculated as follows:

$$f_{gravel} = k_{soil}\left(1 + \frac{A_{itf}}{B_{itf} + F_{itf}}\right) \tag{2}$$

where $f_{gravel}$ is the stable infiltration rate after breaking through Layer *itf*; $k_{soil}$ is the saturated hydraulic conductivity of the soil layer (mm/h); $A_{itf}$ is the total water capacity of the soil above the interface (mm); $B_{itf}$ is the error caused by the different
soil moisture content of the soil above the interface (mm); $F_{itf}$ is the cumulative infiltration when the front breaks through Layer *itf* (mm).



The large portion of gravel in the gravel layer causes the formation of macropores, which are connected to form a fast channel for transporting water during heavy rains (Fig. 3a). After the INF breaks through the interface, the infiltration water preferentially recharges the groundwater through the macropores. The accumulated infiltration quantity is as follows:

$F = F_{itf} + Q_{gd}$                                                                                         (3)

where $Q_{gd}$ is the quantity of groundwater recharge by infiltration, $Q_{gd} = f_{gravel}(t - t_{itf})$, and $t_{itf}$ is the time when the INF breaks through the interface.

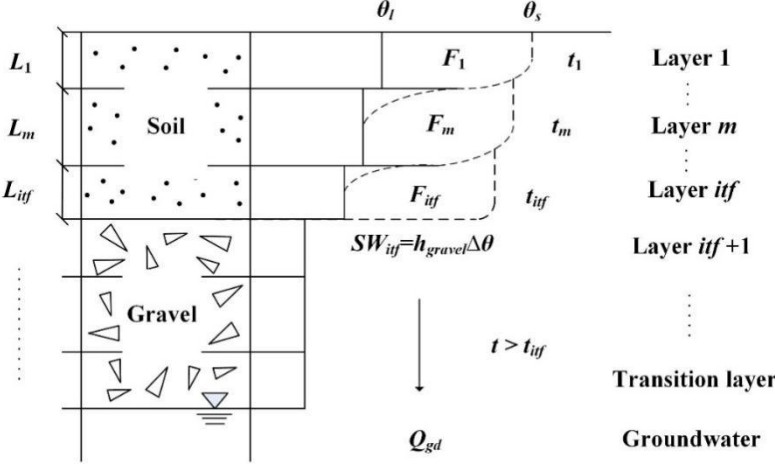

**Figure 4: Cumulative infiltration process of the WEP-QTP model**

**2.2.2 Freeze–thaw period**

During the freeze-thaw period, the hydrothermal coupling simulation object was defined as the snow–soil–gravel layer continuum (Fig. 3b). A snow layer was added on top of the dualistic soil–gravel structure, the thickness of which was determined by the snow water equivalent and snow density. In the QTP, differences in temperature and precipitation caused by altitude differences cause the calculation units at higher altitudes to accumulate more snow and less melting. Calculation

units (the contour bands) with lower elevations accumulate less snow and more melting. In this model, we established a thickness threshold. When the snow thickness difference between two calculation units exceeded this threshold, snow meltdown occurred. The snow in the higher-altitude calculation unit slides into the next unit until the two units had the same snow thickness. The daily variation of snow water equivalent was calculated as follows:

$S = S_p - S_d - S_m$                                                                                         (4)

where $S$ is the daily variation of snow water equivalent (mm/d); $S_p$ is the snow water equivalent from precipitation (mm/d), $S_p$ is equal to the daily precipitation when the average temperature of the day was $T_a < 2$ °C, otherwise $S_p = 0$; $S_d$ is the snow water equivalent variation due to snow sliding down (mm/d), when the difference in snow thickness between contour bands in the same sub-basin exceeds the threshold, the snow slides downwards until the snow thickness is the same; $S_m$ is the quantity of snow melting equivalent (mm/d) calculated using the degree-day factor method (Hock, 1999) as follows:





$$S_m = d_f(T_a - T_S) \tag{5}$$

where $d_f$ is the degree-day factor (mm/[°C·day]); $T_S$ is the critical temperature of snow melting (°C), assuming snow melt starts when $T_a$ is $> T_S$. $S_m$ was treated as precipitation in the model.

The simulation method of water transport under the snow layer was the same as that in non-freeze-thaw period. But the saturated hydraulic conductivity ($K_s$) of the soil or gravel layer was corrected by temperature and can be calculated as follows

(Chen et al., 2008; Jansson, 2004):

$$K_s = \begin{cases} K & T > 0 \\ K(0.54 + 0.023T) & T_f \leq T \leq 0, \\ K_0 & T < T_f \end{cases} \tag{6}$$

where $K$ is the initial saturated hydraulic conductivity (cm/s) and $K_0$ is the minimum hydraulic conductivity (cm/s) under freezing conditions. Considering the difference in the hydrodynamic properties of the soil and gravel layer, the $K_0$ value for soil was 0 cm/s. For the gravel layer, due to the larger pores, $K_0$ has a value of $> 0$ cm/s; $T$ is the temperature of the soil or

gravel layer (°C), while $T_f$ is the critical temperature (°C) corresponding to the minimum hydraulic conductivity.

For the heat transfer process, assuming that the upper boundary of the system is the atmosphere, which controls the input and output of the system energy. In the presence of snow on the surface, the atmosphere first exchanges energy with the snow layer, and then the snow layer exchanges energy with the soil. Conversely, in the absence of snow, the atmosphere directly exchanges energy with the soil, and the upper boundary energy can be calculated by meteorological elements. The lower

boundary at the bottom is the impermeable layer, assuming that it maintains a constant temperature (seasonal frozen soil regions $> 0$ °C, permafrost regions $< 0$ °C). The energy balance equation of the surface can be expressed as follows (Jia et al., 2001):

$$RN = LE + H + G \tag{7}$$

$$LE = L_{vi}E_{subl} + L_{vl}E_{evap} \approx L_{vi}E \tag{8}$$

$$H = \frac{\rho_a C_P (T_S - T_a)}{r_a} \tag{9}$$

where $RN$ is the net radiation flux (MJ/m²/d); $LE$ is the latent heat flux (MJ/m²/d), which was calculated from Eq. (8), $L_{vi}$ is the latent heat of sublimation of ice ($2.838 \times 10^6$ J/kg at 0°C); $L_{vl}$ is the latent heat of evaporation of water ($2.505 \times 10^6$ J/kg at 0 °C); $E$ is the sum of the surface sublimation and evaporation rates; $H$ is the sensible heat flux (MJ/m²/d), which was obtained from Eq. (9), $\rho_a$ is the air density; $C_P$ is the constant pressure specific heat of the air; $T_s$ is the surface

temperature; $T_a$ is the air temperature; $r_a$ is the aerodynamic impedance, and $G$ is the heat flux (MJ/m²/d) conducted into the snow or soil, which was determined by the temperature difference between the soil or snow and the atmosphere near the surface. The above equation was combined with the ground heat conduction and energy balance equations and solved using the iterative method, which requires extensive calculations. Therefore, in this study, we approximately deduced the heat into the ground according to the daily temperature change and simplified the calculation by solving the $H$ according to the energy

balance equation after calculating the $LE$.





For soil and gravel layers, the average temperature was represented by the temperature in the middle of the layer. The temperature difference between the atmosphere and the surface is the source of heat conduction; after the heat flux conducted into the snow or soil was determined, the heat flux and temperature of each layer were calculated via the following equation (Shang et al., 1997; Wang et al., 2014):

$$C_v \frac{\partial T}{\partial t} = \frac{\partial}{\partial z}\left[\lambda \frac{\partial T}{\partial z}\right] + L_f \rho_I \frac{\partial \theta_I}{\partial t} \tag{10}$$

where $C_v$ and $\lambda$ are the volumetric heat capacity (J/[m³·°C]) and thermal conductivity (W/[m·°C]) of the soil or gravel layer, respectively; $L_f$ is the latent heat of ice melting (3.35 × 10⁵ J/kg); $T$ is the temperature (°C) of the soil or gravel layer; $\rho_I$ is the ice density (kg/m³); $\theta_I$ is the volumetric content of ice in the soil or gravel layer (cm³/cm³); $z$ is the layer thickness (m).

For the snow layer added to improve the model, the main hydrothermal parameters include thermal conductivity, volumetric

heat capacity, and snow density. The calculation formulas of each parameter are as follows:

Snow density was calculated via equation (11) (Hedstrom and Pomeroy, 1998):

$$\rho_s = \begin{cases} 67.9 + 51.3 e^{T_a/2.6} & T_a \leq 0 \\ 119.2 + 20 T_a & T_a > 0 \end{cases}. \tag{11}$$

The thermal conductivity and volumetric heat capacity of snow were calculated as follows (Goodrich, 1982; Ling and Zhang, 2006):

$$\lambda_s = \begin{cases} 0.138 - \frac{1.01 \rho_s}{1000} + 3.233\left(\frac{\rho_s}{1000}\right)^2 & 156 < \rho_s \leq 600 \\ 0.023 + \frac{0.234 \rho_s}{1000} & \rho_s \leq 156 \end{cases} \text{ and} \tag{12}$$

$$C_{Vs} = 2.09 \rho_s \times 10^3, \tag{13}$$

where $\rho_s$ is the snow density (kg/m³); $T_a$ is the atmospheric temperature (°C); $\lambda_s$ is the thermal conductivity of the snow (W/[m·°C]); $C_{Vs}$ is the volumetric heat capacity of the snow (J/[m³·°C]).

As opposed to a single soil medium, the presence of gravel also has a great influence on the heat conduction of the soil–

gravel layer. The main thermal parameters of the soil–gravel layer include volumetric heat capacity and thermal conductivity. The calculation formulas for each parameter were as follows:

Volumetric heat capacity were calculated via equation (14) (Chen et al., 2008):

$$C_V = (1 - \theta_s) \times C_s + \theta_l \times C_l + \theta_I \times C_I, \tag{14}$$

where $\theta_s$, $\theta_l$, and $\theta_I$ are the saturated volumetric water content, volumetric liquid water content, and volumetric ice content of

the soil or gravel layer, respectively; $C_s$, $C_l$, and $C_I$ are the volumetric heat capacity (J/[m³·°C]) of the soil or gravel layer, water, and ice, respectively; at 0 °C, the soil and gravel layers have values of 1.93×10⁶ J/[m³·°C] and 3.1×10⁶ J/[m³·°C], respectively; and water and ice have values of 4.213×10⁶ J/[m³·°C] and 1.94×10⁶ J/[m³·°C], respectively.

The thermal conductivity calculation referred to the IBIS model, as follows (Foley et al., 1996):

$$\lambda = \lambda_{st} \times (56^{\theta_l} + 224^{\theta_I}), \text{ and} \tag{15}$$

$$\lambda_{st} = \omega_{gravel} \times 1.5 + \omega_{sand} \times 0.3 + \omega_{silt} \times 0.265 + \omega_{clay} \times 0.25, \tag{16}$$





where $\lambda$ and $\lambda_{st}$ are the actual thermal conductivity of the soil or gravel layer and the thermal conductivity in the dry state (W/[m·°C]), respectively, and $\omega_{gravel}$, $\omega_{sand}$, $\omega_{silt}$, and $\omega_{clay}$ are the volume ratios of the gravel, sand, silt, and clay, respectively.

## 2.3 Model evaluation criteria

Data from January 2013 to December 2018 were used to evaluate the simulation results of daily flow rates at Gongbujiangda, Baheqiao, and Duobu stations. The performance of the model was first evaluated using a qualitative assessment via graphs and then assessed quantitatively using statistical metrics including the Nash–Sutcliffe efficiency (NSE) and relative error (RE). The NSE and RE were calculated as follows:

$$NSE = 1 - \frac{\sum_{i=1}^{N}(O_i - S_i)^2}{\sum_{i=1}^{N}(O_i - \overline{O_i})^2} \tag{17}$$

$$RE = \frac{\sum_{i=1}^{N} S_i - \sum_{i=1}^{N} O_i}{\sum_{i=1}^{N} O_i} \times 100\% \tag{18}$$

where $N$ is the number of observations; $O_i$ is the observed value; $\overline{O}$ is the mean observed value; and $S_i$ is the simulated value.

## 3 Results and discussion

### 3.1 Model calibration and validation

We calibrated and verified the daily flow process of Gongbujiangda, Baheqiao, and Duobu stations—located upstream, on
the largest tributary, and downstream, respectively—from 2013 to 2018. The data from Duobu station were split into two parts: data from 2013 to 2015, which were used for calibration, while those from 2016 to 2018 for validation. The discontinuous, measured flow data from Gongbujiangda and Baheqiao stations from 2013 to 2018 were used to verify the model.

The parameters of the model were divided into four categories namely underlying surface parameters, vegetation parameters,
soil parameters and aquifer parameters. All parameters have physical meaning and can be estimated based on observational experimental data or remote sensing data. The sensitivity of the above four types of parameters was analyzed (Jia et al., 2006), and the sensitivity of these parameters was divided into three levels namely high, medium, and low. Highly sensitive parameters included soil thickness, soil saturated hydraulic conductivity, and riverbed material permeability coefficient. The model was calibrated according to the runoff process. The saturated hydraulic conductivity of the soil layer was 0.648 m/d,
that of the gravel layer was 4.32 m/d, and the riverbed conductivity was approximately 5.184 m/d. The thickness of the soil layer at the mountaintop, mountainside, and foot of the mountain was 0.4 m, 0.6 m, and 1.0 m, respectively. In addition, for model improvement, the main parameter calibration results are as follows: the degree-day factor of snow was 4 mm/[°C·day], the critical temperature of snow melting was −1 °C, and the critical value of non-heavy and heavy rain periods was 15 mm/day.





Figure 5 and Table 1 present the results of both calibration and verification periods of the daily flow data from Duobu station and only the latter from Gongbujiangda and Baheqiao stations. The simulation results of the WEP-QTP model from the three stations were consistent with the measured flow data. During the verification period, compared with the WEP-COR model, the NSE of WEP-QTP increased and the RE decreased, thereby considerably improving the simulation effect of the model. The WEP-QTP simulation flow process was smoother, and a large flow peak was not easily formed. However, we also found

that WEP QTP underestimated the river discharge during the frozen period, which may be attributed to the model not explicitly considering the outflow of sub-permafrost water, which could also supplement the river discharge in the frozen period through the macropores in the bedrock fracture zone. In general, the WEP-QTP model delivered an acceptable performance for the Niyang River Basin and achieved NSE > 0.75 and RE < 10 % for the validation period. The improved model could be used for further analysis.

**Table 1: Model validation results for Gongbujiangda, Baheqiao, and Duobu stations**

| Model | Duobu | | | | Gongbujiangda | | Baheqiao | |
|---|---|---|---|---|---|---|---|---|
| | Calibration | | Validation | | Validation | | Validation | |
| | NSE | RE | NSE | RE | NSE | RE | NSE | RE |
| WEP-QTP | 0.89 | −5.8 % | 0.76 | 3.4 % | 0.79 | 0.01 % | 0.75 | −5.47% |
| WEP-COR | 0.69 | −4.65% | 0.31 | 0.01 % | 0.67 | 1.66 % | 0.40 | −2.38 % |



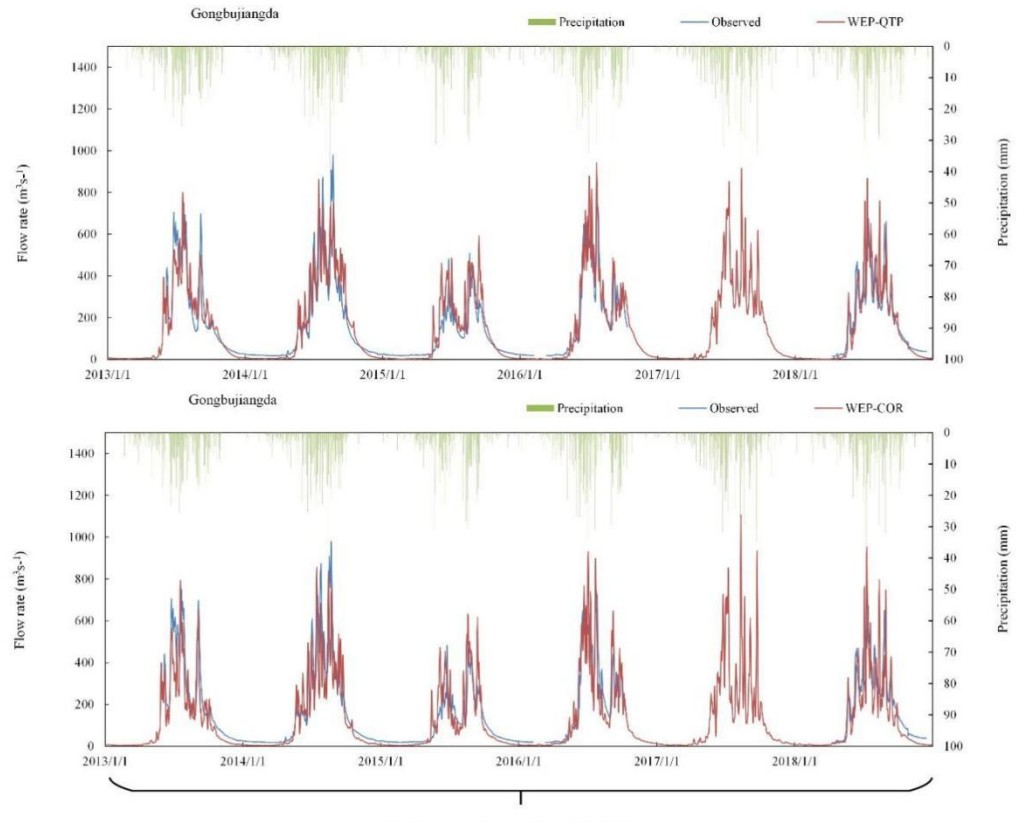

(a)



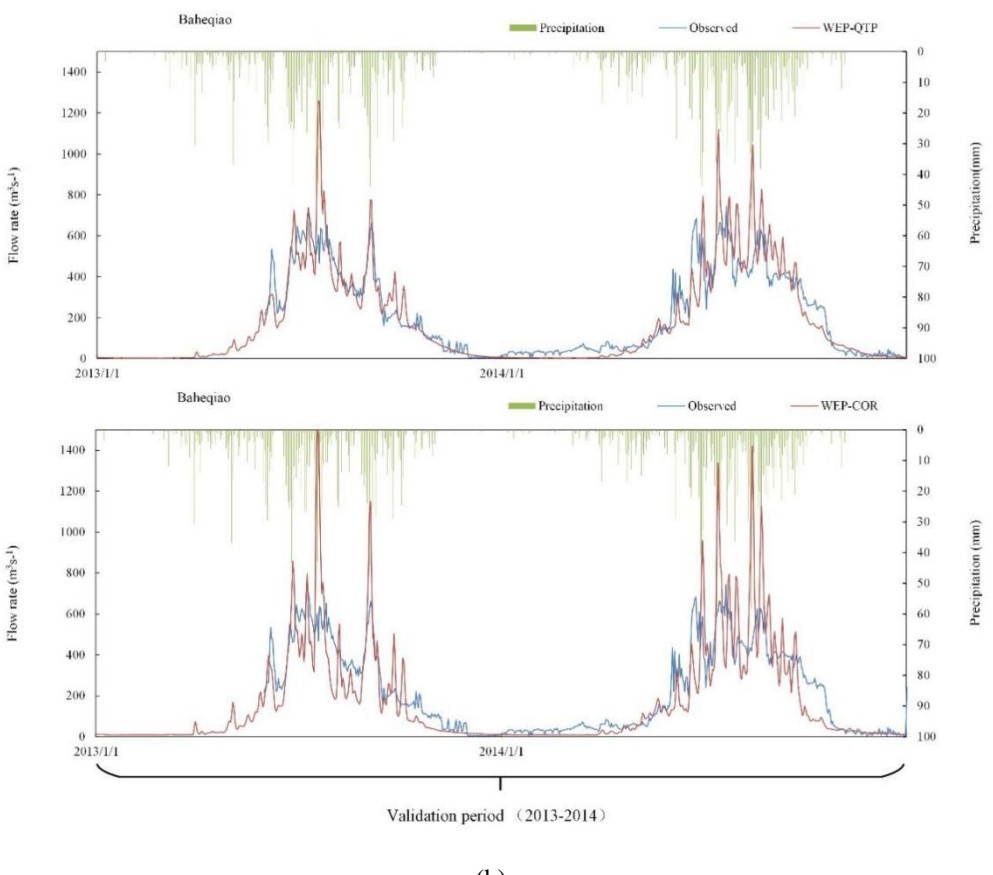


(b)



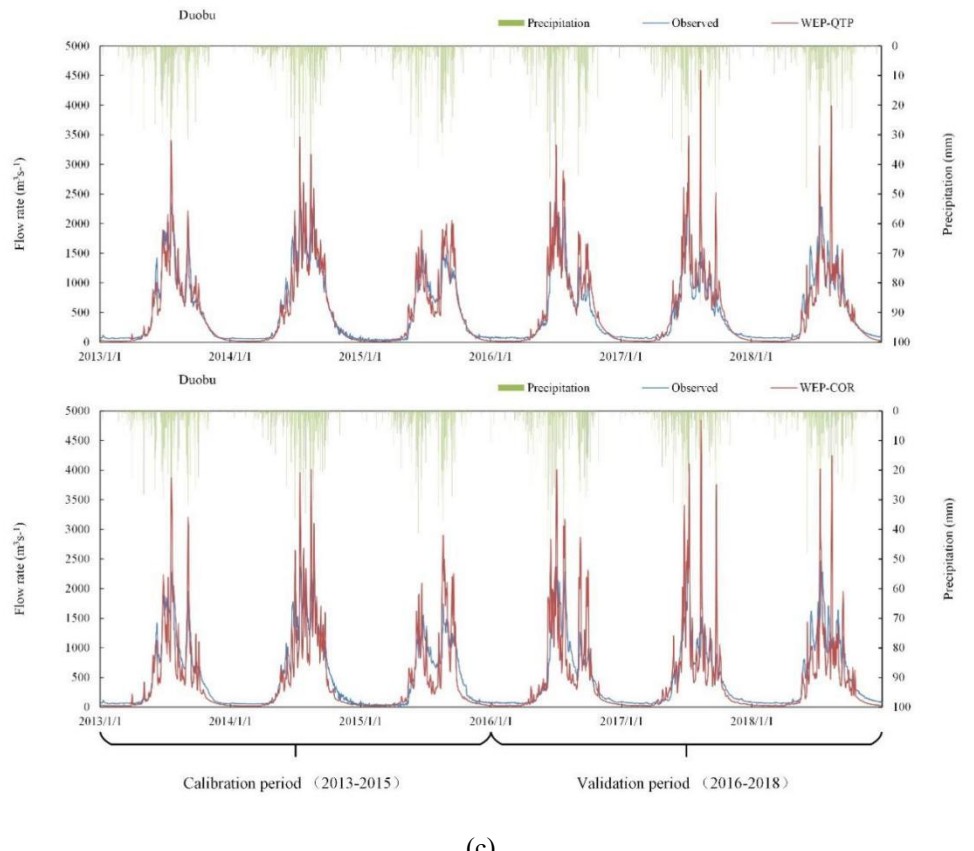

(c)

**Figure 5: Verification results of the WEP-QTP and WEP-COR models at (a) Duobu, (b) Gongbujiangda, and (c) Baheqiao stations**

### 3.2 Simulation and comparison of soil–gravel hydrothermal data at test sites

Figure 6 shows the air temperature and the simulated and measured snow thickness during freezing and thawing at the test point. The snow began to accumulate on 3 December 2016 and was completely melted by 4 April 2017. The maximum snow thickness was 12.4 cm, and the simulated snow thickness was consistent with the measured value. The temperature and moisture of the soil–gravel layer at the experimental point during the freezing and thawing period of the soil were compared with the measured results.


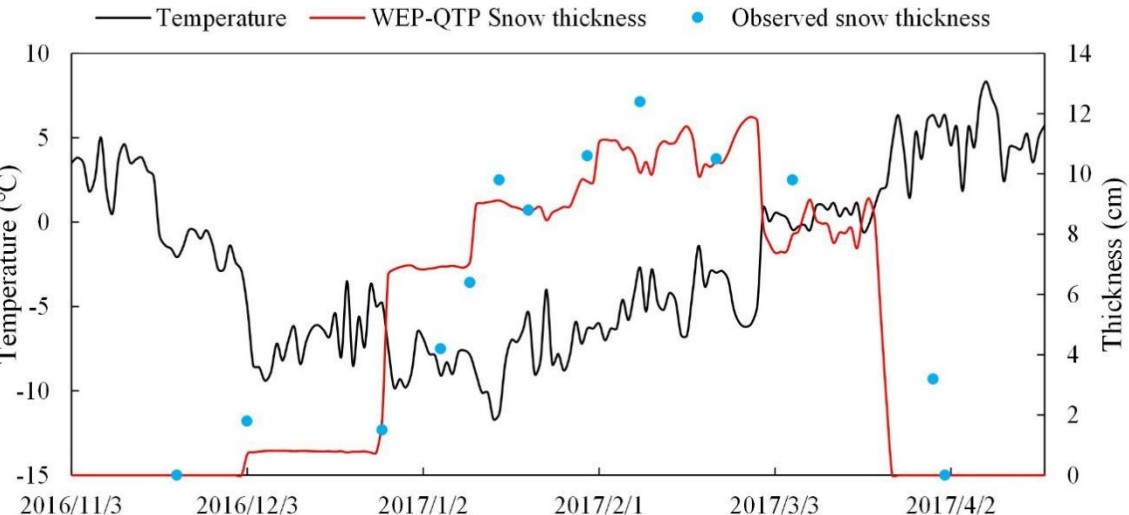

**Figure 6: Snow thicknesses and temperatures during freezing and thawing at the experimental sites from observations and the WEP-QTP model simulated results**

### 3.2.1 Soil–gravel temperature

The soil–gravel temperature simulation results of the WEP-QTP and WEP-COR models are shown in Figure 7. The soil thickness at the test site was approximately 40 cm. For the 10 cm layer, because the simulation parameters of the WEP-QTP and WEP-COR models were the same, the temperature simulation results were consistent except for the snow cover period (i.e., the period when the snow thickness was > 5 cm, 27 December 2016–20 March 2017). Due to the heat preservation effect of the snow, the heat transfer and temperature fluctuations of the surface soil were reduced. The RE of WEP-QTP in the 10 cm soil layer was 11.1% during the snow cover period, which was much less than the 46.0% of WEP-COR. The simulation results of the two models in the 20 cm layer did not differ considerably. For the 40 cm and 60 cm layers, the moisture content of the soil simulated by WEP-COR was greater than that of the gravel layer simulated by WEP-QTP. In the early stage of freezing, the soil in the WEP-COR had a higher water content than the gravel in the WEP-QTP, masking the difference in simulated temperature between the two models (the volumetric heat capacity of water is greater than that of gravel and soil). However, as the temperature decreased, the moisture in the soil was converted into ice with a smaller heat capacity; thus, the difference in thermodynamic properties between the gravel and the soil gradually increased. During this period, the simulation difference between the two models reached a maximum of 1.41 °C (40 cm layer, 26 January 2017). For the temperature simulation below 60 cm, because the temperature was higher compared with that in the 40–60 cm layer, the thermodynamic properties between the two models were not significantly different due to water phase change, and the difference in the non-snow cover period was not as great as in the 40–60 cm layer. During the snow cover period, the effect





of snow on the temperature was also reduced due to the weakening of the upper soil–gravel layers. The snow cover reduces the heat transfer and temperature fluctuations of the soil layer, which improves the simulation accuracy of the surface soil temperature. In addition, by neglecting the special hydraulic and thermodynamic properties of the gravel layer, the WEP-

COR underestimated the soil temperature. The snow and gravel layer collectively resulted in the temperature simulation difference between the two models. The average RE of WEP-COR was -3.60%, while that of WEP-QTP was 0.08%. The results of the WEP-QTP simulation were closer to the actual measurement; thus, it could accurately reflect the temperature changes of each layer during freeze–thaw period.

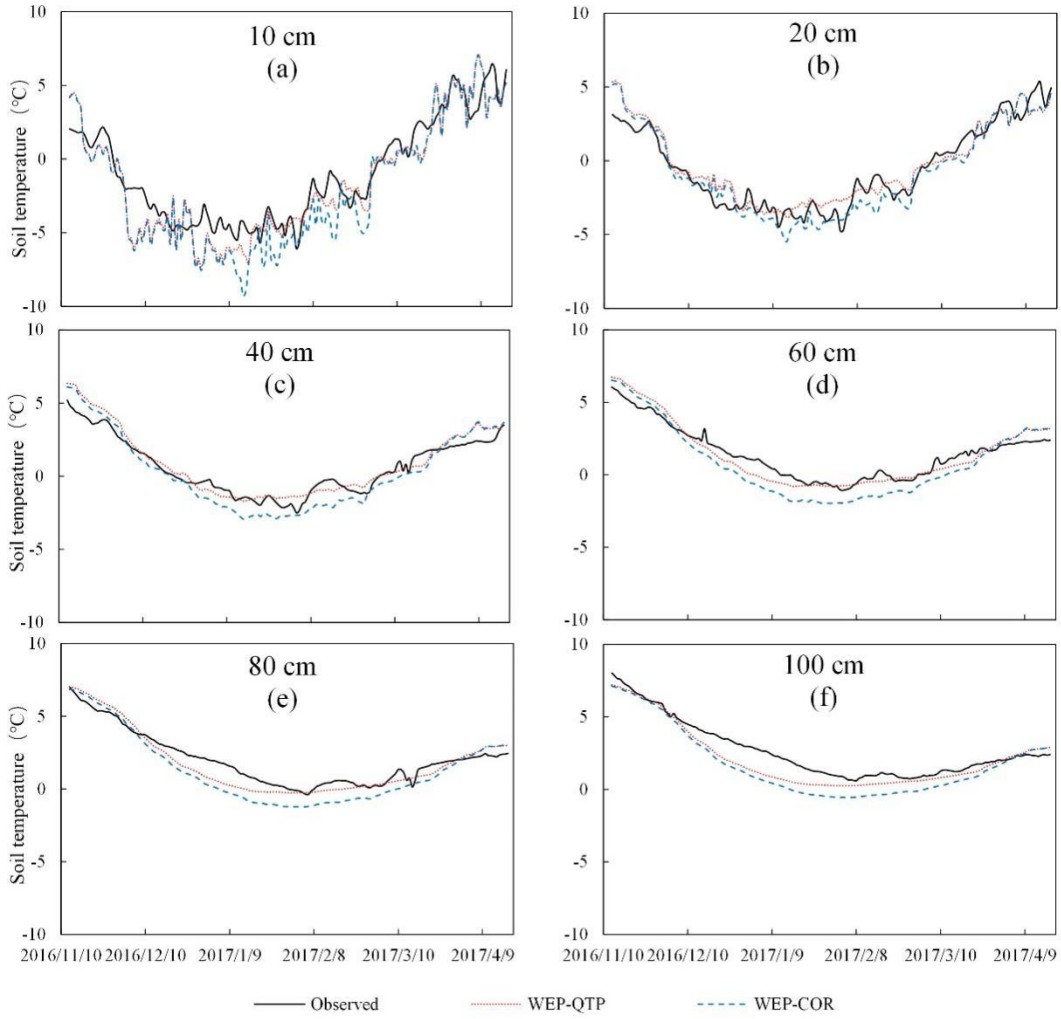


**Figure 7: Simulated (WEP-QTP and WEP-COR models) and observed temperatures of the soil–gravel layer at different depths**





### 3.2.2 Soil–gravel moisture

Figure 8 shows the comparison between the simulated and measured values of the liquid water content in the freeze–thaw

period of the WEP-QTP and WEP-COR models at the experimental points in 2016–2017. During the freezing period (December–March), the upper layer liquid water content first dropped due to the temperature drop, followed by the lower gravel layer liquid water content drop, thereby stabilizing in January–February. When the temperature increased in March, the upper layer initially began to melt, thereby increasing the liquid water content and the subsequent melting of the lower layer. After the thawing period, the upper part of the soil–gravel layer had a higher water content than that in the lower part

due to the infiltration of snow melt, and was also higher than before freezing.

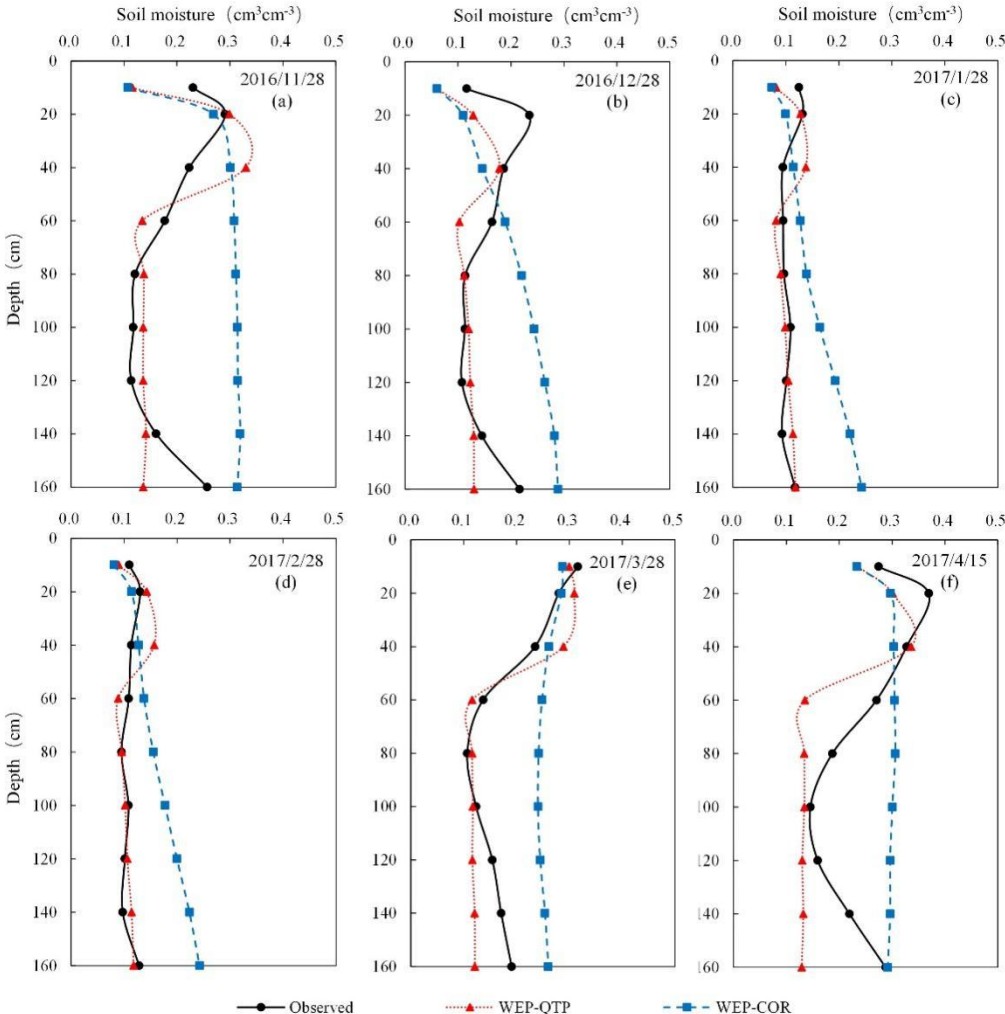

**Figure 8: Simulated (WEP-QTP and WEP-COR models) and observed moisture contents of the soil–gravel layer during the freezing and thawing periods**





During the entire freeze–thaw period, the lower gravel layer did not affect the simulation of soil moisture content in the 10

cm layer; however, for the 20–40 cm layers, when the lower layer was gravel, the moisture was more easily retained in the upper soil because its water holding capacity was greater than that of gravel. When the soil starts to freeze or the snow starts to melt, the WEP-QTP simulates a larger moisture content close to the measured value. Below 60 cm, the WEP-COR did not consider the impact of the gravel layer, and the water content was higher than the measured value throughout the freeze– thaw period. However, due to the large uncertainty of the compositions of the soil and gravel layer, the unstable water-

holding capacity of the soil–gravel layer cannot be accurately reflected when the model is generalized, which also leads to a certain difference between the WEP-QTP simulation and the measured values. There might have been a soil interlayer at 160 cm, and the measured water content was between the simulated values of the WEP-QTP and WEP-COR. The average RE of WEP-COR was 33.74%, and that of WEP-QTP was smaller at -12.11%. WEP-QTP could reflect the influence of gravel on the vertical migration of water.

**3.3 Simulation and comparison of watershed flow process**

To further analyze the improved WEP-QTP compared with the original WEP-COR, three sites with data for 2014 were selected, and the daily flow data of the three stations were compared with the simulated data of the two models (Fig. 9). The simulation difference between the two models was from June to November. The simulation performance of the WEP-QTP was better than that of the WEP-COR in three aspects: the peak value of the flood season was not too high; the valley value

of the flow process was higher than that of the WEP-COR; there was a tailing process after October.

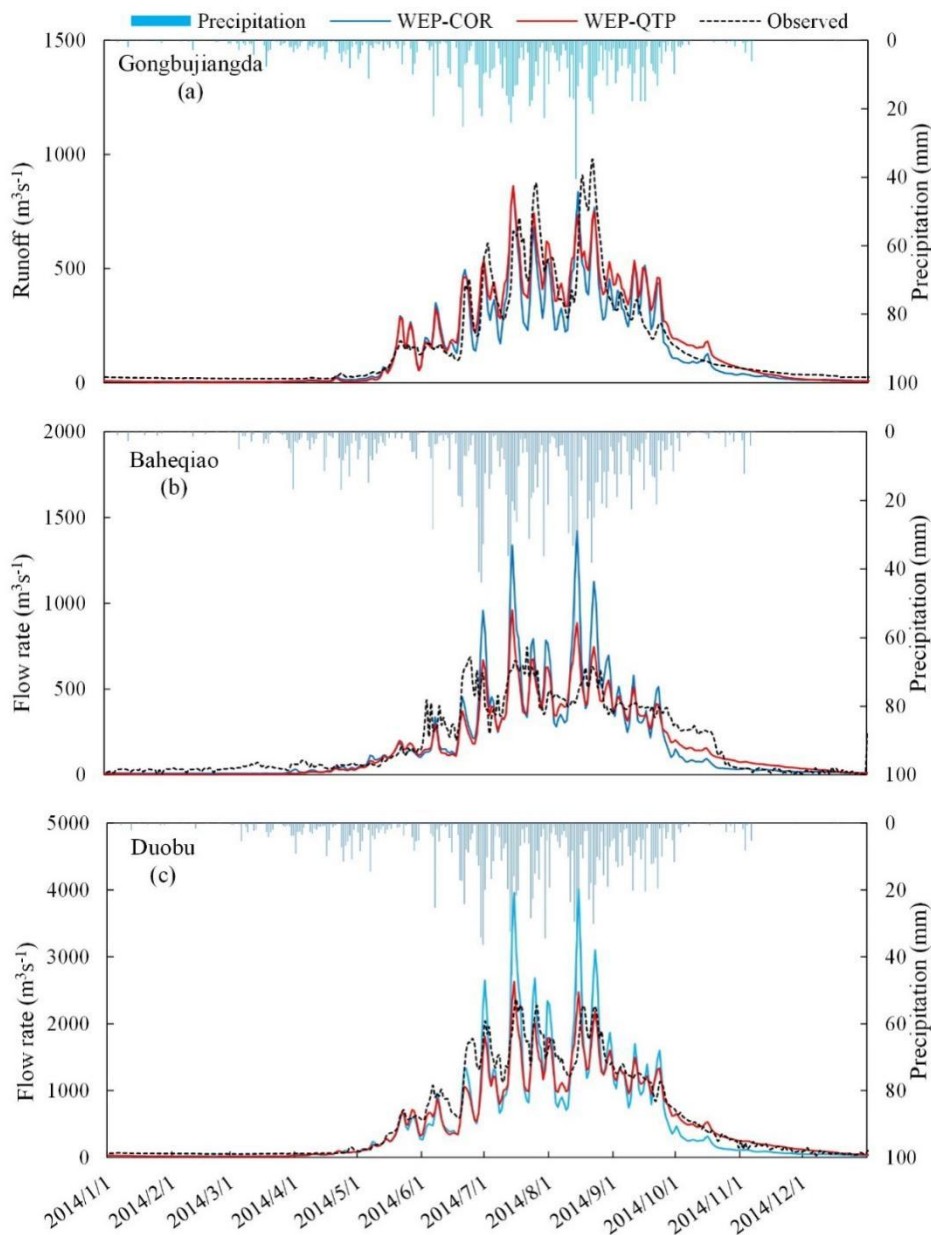

**Figure 9: Simulated (WEP-QTP and WEP-COR models) and observed flow rates at (a) Gongbujiangda, (b) Baheqiao, and (c) Duobu stations in 2014**

Figure 10 shows the comparison and analysis of the changes in hydrological cycle flux. The surface runoff from precipitation simulated by WEP-QTP was smaller than that by WEP-COR since the precipitation in WEP model can recharge groundwater more rapidly during heavy rains through macropores in gravel layer, while flow peaks are not easily



formed. However, in the WEP-COR model, this part of water mostly formed the peak flow which was inconsistent with the measured value. Therefore, the WEP-QTP performed better in the peak simulation during the flood season, and could
simulate more groundwater discharge to river, which in turn perform better when simulating low-flow.

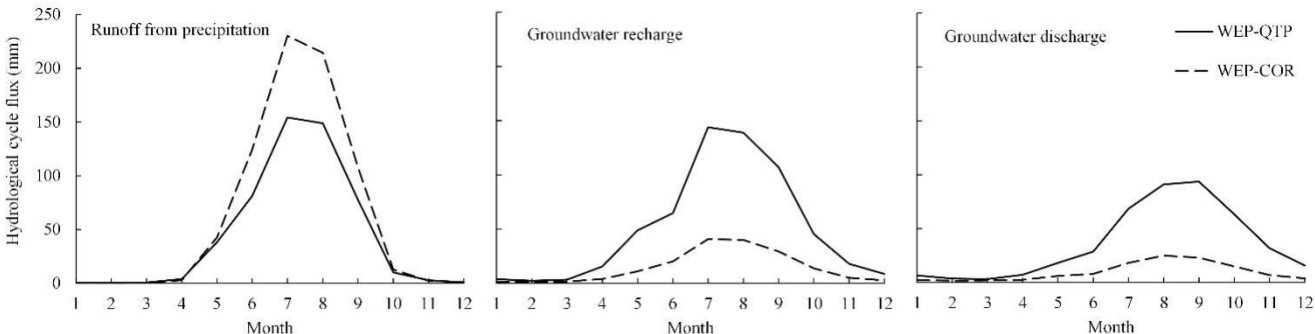

**Figure 10: Monthly change process of hydrological cycle flux in the basin**

In addition, snow cover significantly contributed to the inconsistencies between the temporal and spatial changes of the frozen soil in the two models, which in turn caused variations in the groundwater recharge and discharge process. The WEP-COR did not consider snow cover, the upper boundary was the atmosphere, and the soil responds quickly to the change of atmospheric temperature. During the thawing process, the area of frozen soil with different thickness was decreased and then increased with the change of temperature from January to December (Fig. 12). Similar to the change of frozen soil area
proportion simulated by the WEP-COR, the snow cover rate in the valley (< 3,800 m a.s.l.) began to melt and decreased after February. However, inconsistent with the valley, in the high-altitude area (> 4,600 m a.s.l.) the temperature was lower than the valley, and the snow thickness continued to increase (Fig. 11). The proportion of the area with snow thickness greater than 1.5 m reached the maximum in April (38.51%), while the area without snow cover reached its maximum in July and then gradually decreased after August. This was similar to the change of the proportion of the area with the frozen soil
thickness greater than 1.5 m simulated by the WEP-QTP (Fig. 12). The snow made the frozen soil area change simulated by the WEP-QTP lag the WEP-COR, as well as the groundwater discharge process (QTP's groundwater discharge reached its maximum in September, while COR's in August). Water in the WEP-QTP model had more time for groundwater recharge and river recharge, thus showing a better "tailing" process compared with the WEP-COR.





**Figure 11. Temporal and spatial variation of snow cover thickness in 2014**

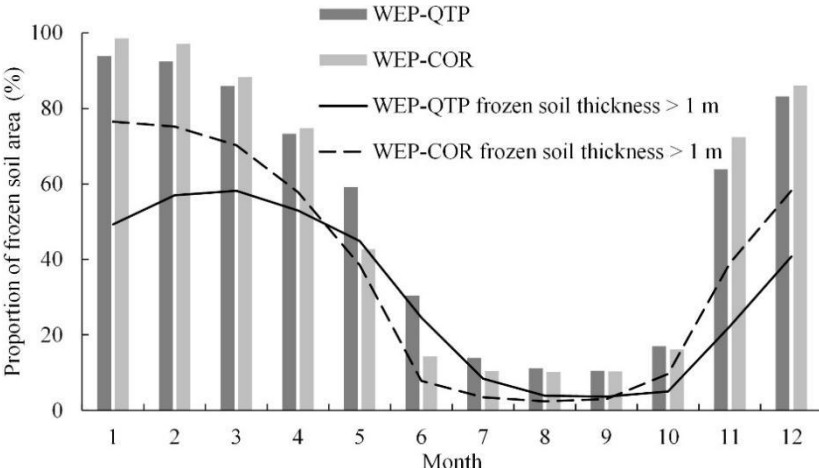

**Figure 12: Monthly change process of the frozen soil area proportion in the basin**

In cold regions, climate change affects high and low flows differently. As the warming and wetting continue, changes in high
flows show spatial heterogeneity, while low flow shows a consistent increasing trend (Song et al., 2021). Warming enhances
subsurface flow connectivity and increases groundwater flow by increasing the thickness of the active layer (St. Jacques and
Sauchyn, 2009; Walvoord and Kurylyk, 2016). As the medium connecting groundwater and surface water, the hydrothermal
transport of the active layer has a great influence on the water cycle process in cold regions (Chen et al., 2014; Kurylyk et al.,
2014). Our study demonstrates that the soil–gravel layer structure in Qinghai Tibet Plateau is different from the single soil
layer structure in general cold regions. In terms of heat transfer, snow and gravel block heat conduction and slow down the
freezing and thawing rates of aquifers. For water transfer, the large pores of the lower gravel layer increase the regulation
and storage effect of groundwater of the flow processes. Under the future climate conditions, the participation of
groundwater in the water cycle of the Qinghai Tibet plateau may be higher than we expected. Ignoring the snow–soil–gravel
layer structure will affect the hydrological forecast, reservoir regulation, and water resource utilization.

**4 Conclusions**

This study combined the geological characteristics of the thin soil layer on the thick gravel layer and the climate
characteristics of the long snow cover period in the QTP. Using the Niyang River Basin as the research area, the WEP-QTP
model was constructed based on the original WEP-COR model. The improved model divides the single soil structure into
two types of media: soil and gravel layers. In the non-freeze–thaw period, two infiltration models based on the dualistic soil–
gravel structure were developed based on the Richards equation in non-heavy rain scenarios and the multi-layer Green–Ampt
model in heavy rain scenarios. During the freeze–thaw period, a hydrothermal coupling model based on the continuum of the

snow–soil–gravel layer was constructed. This model was used to simulate the water cycle processes of the Niyang River Basin, and the improvement effect of the model was analyzed via comparison with the WEP-COR.

Compared with the simulation results prior to improvement, the addition of snow not only reduced the surface soil temperature fluctuations, but also interacted with the gravel layer to reduce the soil freezing and thawing speed. The low estimation of temperature by WEP-COR was corrected, while the RE was reduced from -3.60% to 0.08%. At the same time, the WEP-QTP model can reflect the impact of the gravel layer under the soil on the vertical movement of water and accurately describe the dynamic changes in moisture in the soil and gravel layers; the RE of the moisture content was reduced from 33.74% to −12.11%.

According to the comparison of the WEP-QTP simulation and measured results of the main stations in the Niyang River Basin, the daily flow process simulated by the model is in line with the actual situation, while the flow simulation result was more accurate (Nash > 0.75 and |RE| < 10 %), which is a considerable improvement compared with the WEP-COR model. In the non-freeze–thaw period, the dualistic soil–gravel structure increased the recharge and discharge of groundwater and improved the regulation effect of groundwater on flow, thereby stabilizing the water flow process. The maximum RE at the

flow peak and valley decreased by 88.2% and 21.3%, respectively. In the freeze–thaw period, by considering the effect of the snow–soil–gravel layer continuum on soil freezing and thawing processes, changes in frozen soil depth simulated by WEP-QTP lagged those of WEP-COR by approximately one month. There was more time for the river recharge by the groundwater, which showed a better tailing process after October.

In contrast to the general cold area, the special geological structure and climatic characteristics of the QTP change the water

cycle processes in the basin. Ignoring the influence of the dualistic soil–gravel structure greatly impacts the hydrological forecast and water resource assessment.

**Data and code availability**

The datasets and model code relevant to the current study are available from the corresponding author on reasonable request.

**Author contribution**

PW performed the model programming and simulations. ZZ, KW and YL conducted the field experiments. ZZ, JL, YL, and JL contributed to the model programming. The writing was performed by PW and ZZ. CX, YJ and HW contributed to the writing of the paper.

**Competing interests**

The authors declare that they have no conflict of interest.



**Acknowledgements**

This work was partly supported by grants from the National Natural Science Foundation of China (91647109, 51879195) and the National Key Research and Development Program of China (2016YFC0402405).

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





## Appendix A Distribution of major hydrologically-relevant features in the basin

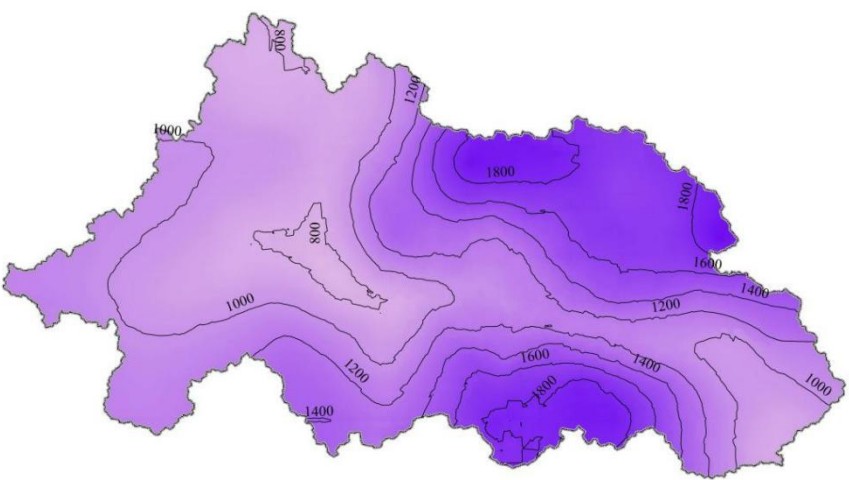

**Figure A1. Annual precipitation contour map (mm/year).**

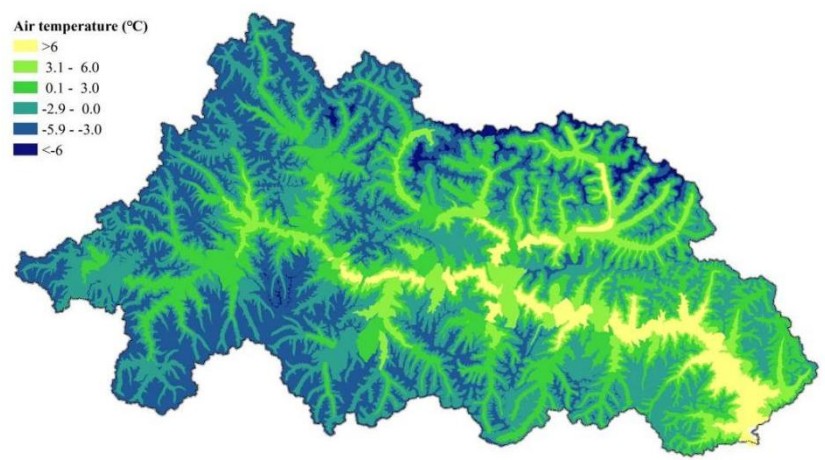


**Figure A2. Mean annual temperature (°C).**

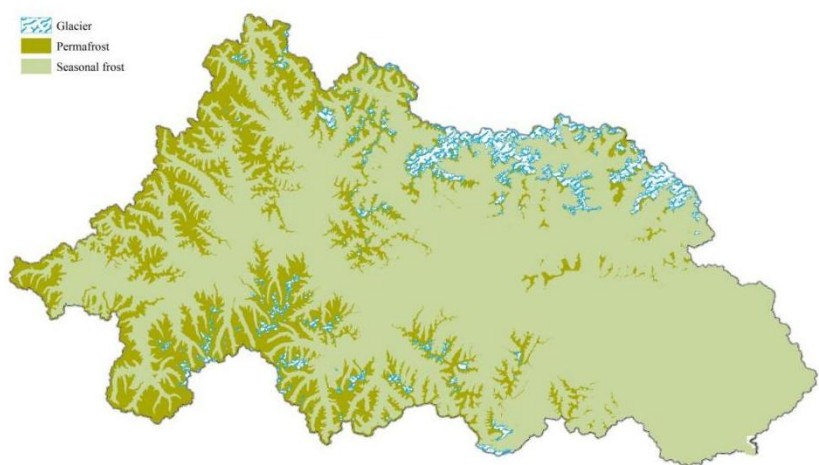

**Figure A3. Glacier and permafrost distributions.**

**Appendix B Introduction of WEP-COR model**

The WEP-QTP model was developed based on the Water and Energy transfer Processes in Cold Regions (WEP-COR) model. For better understanding and comparison, the WEP-COR model is briefly introduced in this section.

**Model structure**

The WEP-COR model is a distributed hydrological model. In terms of the horizontal structure, the WEP-COR uses the contour bands inside small sub-basins as the basic calculation unit (Fig. B1a), and fully considers the vertical changes of

vegetation, soil, air temperature, precipitation, and other factors in the basin with the elevation. Each unit is divided into five types according to the land-use type: water body, soil–vegetation, irrigated farmland, non-irrigated farmland, and impervious area. The calculation result of the water and heat flux in each type was weighted by area to obtain the water and heat flux of the contour band. For glacier-covered areas, the "degree-day factor method" (Hock, 1999) was used to calculate the quantity of glacier melting, and the runoff from the melting of glaciers was directly added to the corresponding hydrological

calculation unit.

The vertical structure of WEP-COR was divided into the vegetation canopy or building interception layer, the surface depression storage layer, the aeration layer, the transition zone layer, and the groundwater layer. To accurately simulate the changes in soil moisture and heat from the surface to the deep layers and to reflect the influence of soil depth on the evaporation of bare soil and the water absorption and transpiration of vegetation roots, the aerated zone soil was divided into

11 layers (Fig. B1b). Among them, $R_{surface}$ represents runoff from the surface, and $R_i$ represents lateral flow or soil flow in the $i$-th layer of soil. $R_i$ is related to slope and soil moisture content. $E_l$ represents soil evaporation, $E_r$ represents vegetation transpiration, $Q_i$ is gravity drainage of the $i$-th layer, $P$ is precipitation, $T_a$ is atmospheric temperature, $T_i$ is the temperature of





the *i*-th layer, and $G_i$ is the heat flux caused by the temperature difference between the *i*-th layer of soil and the adjacent soil layers. The thickness of the first and second layers was set to 10 cm, and the thickness of layers 3–11 was set to 20 cm.


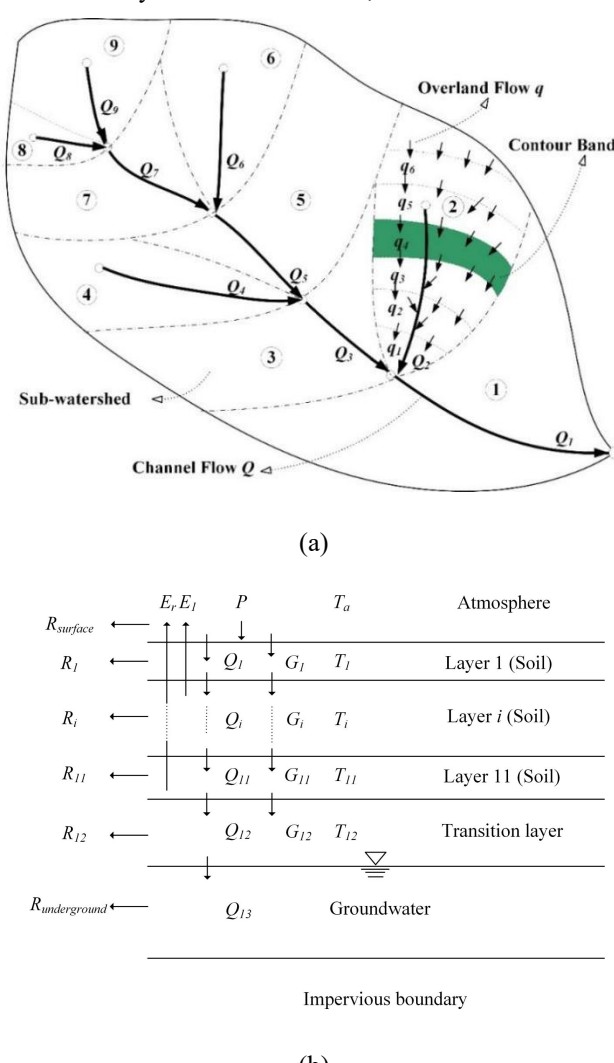

(a)

(b)

**Figure B1: Horizontal (a) and vertical (b) structure of the WEP-COR model**

**Hydrothermal Transport Simulation**

The WEP-COR model divides the soil infiltration process into two scenarios for simulation: the heavy rain infiltration and non-heavy rain infiltration. The division standard is whether the daily rainfall exceeds 20 mm. In the non-heavy rain infiltration scenario, only liquid water migrates. The soil vertical water flux transfer can be written as follows (Shang et al., 1997; Wang et al., 2014):

$\frac{\partial \theta_l}{\partial t} = \frac{\partial}{\partial z}\left[D(\theta_l)\frac{\partial \theta_l}{\partial z} - K(\theta_l)\right] - \frac{\rho_I}{\rho_l}\frac{\partial \theta_I}{\partial t}$ (B1)





where $\theta_l$ is the volumetric content of liquid water in the soil or gravel layer (cm³/cm³); $D(\theta_l)$ and $K(\theta_l)$ are the unsaturated soil hydraulic diffusivity (cm²/s) and hydraulic conductivity (cm/s); $t$ and $z$ are the time and space coordinates (positive vertically downward), $\rho_l$ is the water density (kg/m³).

The Van Genuchten function (Van Genuchten, 1980) was used to describe the upper soil water retention curves:

$$\frac{\theta_l-\theta_r}{\theta_s-\theta_r} = \frac{1}{[1+(\alpha h)^n]^m} \tag{B2}$$

where $\theta_s$ is the saturated water content (cm³/cm³); $\theta_r$ is the residual water content (cm³/cm³); $h$ is the matric suction (cm); $\alpha$ is an empirical parameter (cm⁻¹); $n$ and $m$ are empirical parameters affecting the shape of the retention curve; $m = 1–1/n$.

The hydraulic conductivity was calculated as follows (Chen et al., 2008):

$$K(\theta_l) = \begin{cases} K_S & \theta_l = \theta_s \\ K_S\left(\frac{\theta_l-\theta_r}{\theta_s-\theta_r}\right)^n & \theta_l \neq \theta_s \end{cases}, \tag{B3}$$

where $\theta_r$ is the residual water content of the soil or gravel layer; $K(\theta_l)$ is the hydraulic conductivity (cm/s) of the soil or gravel layer when the liquid water content is $\theta_l$; $K_s$ is the saturated hydraulic conductivity of the soil temperature correction (cm/s); $n$ is Mualem's constant.

In the heavy rain infiltration scenario, the multi-layer unsteady rainfall Green–Ampt model proposed by Jia and Tamai is used (Jia and Tamai, 1998) to calculate the infiltration process. When the infiltration front (INF) reached the $m$-th layer of soil, the soil infiltration capacity was calculated by the following formulas:


$$f = k_m(1 + \frac{A_{m-1}}{B_{m-1}+F}) \tag{B4}$$

$$A_{m-1} = (\sum_1^{m-1} L_i - \sum_1^{m-1} \frac{L_i k_m}{k_i} + SW_m)\Delta\theta_m \tag{B5}$$

$$B_{m-1} = \left(\sum_1^{m-1} \frac{L_i k_m}{k_i}\right)\Delta\theta_m - \sum_1^{m-1} L_i\Delta\theta_i \tag{B6}$$

where $f$ is the infiltration capacity (mm/h); $A_{i-1}$ is the total water capacity of the soil above the $i$ layer (mm); $B_{i-1}$ is the error caused by the different soil moisture content of the soil above the $i$ layer (mm); $F$ is the cumulative infiltration (mm); $k_i$ is the hydraulic conductivity of the $i$-th soil layer (mm/h); $L_i$ is the soil thickness of the $i$-th layer (mm); $SW_m$ is the capillary suction pressure at the INF of the $m$-th layer (mm); $\Delta\theta_i = \theta_s - \theta_l$.


The cumulative infiltration quantity $F$ when the INF reaches the $m$-th layer is calculated based on whether there is water accumulation on the ground surface. If the ground surface has accumulated water when the INF reaches the $m$–1th layer, Eq. (B7) was used; otherwise, Eq. (B8) was used:


$$F - F_{m-1} = k_m(t - t_{m-1}) + A_{m-1}\ln\left(\frac{A_{m-1}+B_{m-1}+F}{A_{m-1}+B_{m-1}+F_{m-1}}\right) \tag{B7}$$

$$F - F_p = k_m(t - t_p) + A_{m-1}\ln\left(\frac{A_{m-1}+B_{m-1}+F}{A_{m-1}+B_{m-1}+F_p}\right) \tag{B8}$$

$$F_{m-1} = \sum_1^{m-1} L_i\Delta\theta_i \tag{B9}$$

$$F_p = A_{m-1}(\frac{I_p}{k_m} - 1) - B_{m-1} \tag{B10}$$





$t_p = t_{m-1} + (F_p - F_{n-1})/I_p$                 (B11)

where $t$ is the time; $t_{m-1}$ is the time when the INF reaches the interface between the $m-1$ and $m$ layers; $t_p$ is the start time of the water accumulation; $F_p$ is the cumulative infiltration quantity at $t_p$; $I_p$ is the precipitation intensity at $t_p$.

The relationship between the water and heat transport of frozen soil is mainly manifested in the dynamic balance of the moisture content of the unfrozen water and the negative temperature of the soil. According to the principle of energy balance,

the energy change of each layer in the freeze–thaw system was used for the soil temperature change and water phase change in the system. The heat flux conducted into the soil was calculated using the forced recovery method (Douville et al., 1995; Pitman et al., 1991). The heat flux and temperature of each layer were calculated using the following equation (Chen et al., 2008):

$H_{i,i+1} = 0.1782 \frac{(\lambda_i Z_i + \lambda_{i+1} Z_{i+1})(T_i - T_{i+1})}{(Z_i + Z_{i+1})^2}$ and                 (B12)

$T_i = \frac{H_{i-1,i} - H_{i,i+1}}{C_{vi} Z_i},$                           (B13)

where $i$ is the number of soil layers; $H_{i,i+1}$ represents the sensible heat flux between the i layer and the $i+1$ layer (MJ/m²);   $\lambda$ represents the thermal conductivity (W/[m·°C]); $Z$ is the thickness (cm); $C_v$ is the soil volume heat capacity (J/[m³·°C]); $T$ is the temperature (°C).

Temperature is the driving force of the water phase change. The relationship between the water and heat transport of frozen

soil is mainly manifested in the dynamic balance of the moisture content of the unfrozen water and the negative temperature of the soil:

$\theta_l = \theta_m(T)$                                    (B14)

where $\theta_m(T)$ is the maximum unfrozen water moisture content corresponding to a negative soil temperature.

For other details of the WEP-COR model, please refer to Li et al. (2019).
