# Peer review of "Application of an improved distributed hydrological model based on soil-gravel structure in the Niyang River basin, Qinghai-Tibet Plateau"

_Hydrology and Earth System Sciences, 2022_

## Author Comment (AC1)

The paper is interesting to read, which presented the development of Water and Energy transfer Processes model in the Qinghai–Tibet Plateau (WEP-QTP) that modified based on the original Water and Energy transfer Processes model in Cold Regions (WEP-COR). In the presented model, the vadose zone processes considered three strategies under different conditions: (1) a dualistic soil-gravel structure using the Richards equation under non-heavy rain in the nonfreeze–thaw period; (2) a multi-layer Green-Ampt model in a heavy rain scenario in the nonfreeze–thaw period; and (3) a hydrothermal coupling model based on the continuum of the snow-soil-gravel layer during the freeze—thaw period. The modified model was then verified with measured river discharge in Niyang River Basin by comparing the simulated groundwater.

The study adopted a new conceptualization of the water and energy transfer in Qinghai–Tibet Plateau, which is considered as novel. However, significant improvement is needed before the consideration of publication to Hydrology and Earth System Sciences.

**Dear Reviewer:**

We appreciate the detailed and valuable comments, which have considerably improved the quality of our manuscript. Our responses to the comments are provided below.

**Major comments:**

1. Author stated in Line 52-62 that the existence of gravel in soil can significantly affect the soil water content and water transport. However, coupling of soil water and heat transport may be still not fully achieved in the modified version of WEP-QTP. When the dualistic soil-gravel structure was used in the nonfreeze-thaw period, the soil water transport may be decoupled with the thermal transport (see Line 296: "for soil and gravel layers, the average temperature was represented by the temperature of the middle layer"). It seems that the full coupling of water and heat transport can only be achieved for freeze—thaw period? Author should at least state whether the neglection of heat transport in nonfreeze-thaw period affect hydrological processes.

Reply: Thanks for the comment and suggestion. In the non-freeze-thaw period, all the water was in a liquid state, and the heat conduction had a minor effect on the water migration process. The model detected the non-freeze-thaw period based on ice content and temperature of each computing unit. In that period, only the moisture simulation was performed for simulation efficiency. We will make the supplementary modifications in Section 2.2 to address this point.

2. During the nonfreeze-thaw period, the soil hydrology was simulated with either a dualistic soil-gravel model or a Green-Ampt equation, and the selection of the two options depend on whether the rainfall intensity exceeded 20 mm/day (Line 677). Why was such threshold selected? Would the dualistic model more suit to the high intensity rainfall?

Reply: The runoff generation mechanism is different for the non-heavy and heavy rain scenarios: during non-heavy rain, there is saturation-excess, while during heavy rain, there is infiltration-excess. In heavy rain scenario, the Richards model is unstable for soil hydrology simulation while the Green-Ampt model is stable and has high computational efficiency. Therefore, in the WEP model, this threshold value was used to divide the simulations into two scenarios.

3. In the schematic figure shown in Fig. 3 (a), the author proposed a dualistic soil-gravel model, it is not clear whether the dualistic model is similar to the dual-porosity model proposed by Greke and van Genuchten (1993). Moreover, author should clearly state how to separate the water flow in such dualistic pore system.

Reply: Thanks for the insightful comment. In the dual-porosity model proposed by Greke and van Genuchten, the water transport medium is a mixture of soil and rock that is consistent from top to bottom; hence, the model generalized the medium into two systems: macropore and matrix pore. Our research object was the upper and lower layered medium with the thin soil layer and thick gravel layer in the Qinghai–Tibet Plateau. Based on this, we generalized the medium as an upper and lower dualistic soil–gravel structure to simulate the process of water and heat transport in different periods. In this model, the water flow is not separated like in the dual-porosity model. In future research, we will refer to the dual-porosity model to improve the mathematical description method of water transport in the gravel layer.

4. The soil water retention curve was described with van Genuchten model in Eq. B2 (Line 685), while the soil hydraulic conductivity function adopted a power functionï¼^B3ï¼‱which is similar to that was used in Brooks-Corey model. Besides, the parameter n in Eq.B3 also adopted Mualem's constant (Line 692). Such combination may be acceptable only if more cautions were taken for the parameterizations. Author should clarify why chosen to combine the selected soil water retention curve and soil hydraulic function, and how these soil hydraulic parameters were specified for distributed hydrological modeling.

Reply: We apologize for the unclearness on this part. The meanings of the two n values in Equation B2 and Equation B3 are different. In Equation B2, n and m are empirical parameters affecting the shape of the retention curve; m = 1-1/n. In Equation B3, n is Mualem's constant. In the revised version, we will replace letters to clearly explain the meanings of different variables in the equation.

The combined application of the two models has been verified in the previous WEP COR model (Li 2019), which performs well in simulating water transport in frozen soil. These two models are mainly used to calculate the unsaturated soil hydraulic diffusivity  $D(\theta_l)$  and the hydraulic conductivity  $K(\theta_l)$  in Equation B1. Equation B1 was used to calculate the vertical movement of water in unsaturated soils.

5. In Page 20, Fig.8, why the simulated soil moisture differed between the two models in a freeze-thaw period (Line 414)? Modification in the proposed model

**may be solely focused on the nonfreeze-thaw period.**

Reply: Differences in simulated soil moisture were caused by different model structures. The WEP-COR model did not consider the layered geological features of Qinghai-Tibet Plateau; the simulation object was homogeneous soil. Therefore, the simulated moisture of the WEP-COR model changed gradually in the vertical direction, and a large difference between simulated and measured values occurred below 40 cm (the soil layer thickness at the experimental site is 40 cm, with gravel layer below 40 cm.). The WEP-QTP model took this geological structure into account. Gravel layer has higher hydraulic conductivity and lower water-retention capacity, which is manifested in the simulated difference in the water content of the gravel layer. The simulated results from the improved WEP-QTP model were closer to the measured values.

The model improved in both the freeze-thaw period and non-freeze-thaw period. Like in the non-freeze-thaw period, the revised formula for water retention properties of the soil–gravel mixture was used to describe water retention curves for the lower gravel layer during the freeze-thaw period (Equation 1). The saturated hydraulic conductivity ( $K_s$ ) of the soil or gravel layer was corrected by temperature (Equation 6). There are also improvements to the heat transport calculations; please see Section 2.2.2 for details.

**6. All the figures have poor resolution. Please consider replacing all of them.**

Reply: Thank you for your suggestion. In the revised version, we will replace all the figures with high-quality figures and improve figure layout.

**Minor comments:**

**7. Line 132: "Temperature" should be "temperature"**

Reply: Thanks, we will correct it and check for other potential errors.

**8. Line 154: The "0" may be redundant.**

Reply: Sorry for this mistake, we will correct it and check for other potential errors.

**9. Line 164: The citation of MODIS data should be added.**

Reply:ThecitationofMODISdatais:https://ladsweb.modaps.eosdis.nasa.gov/search/

**10. Line 166: The citation China's second glacier inventory data set should be added.**

Reply: Line 169 shows this citation (http://westdc.westgis.ac.cn/).

**11. Line 167: add the citation of Water and Energy transfer Processes in Cold Regions (WEP-COR) model.**

Reply: Thanks, we will add this citation in the revised version.

**12. Line 267: The unit of saturated hydraulic conductivity Ks and snow water equivalent S should be consistent.**

Reply: *S* is the daily variation of snow water equivalent (mm/d), which is consistent with the unit of daily precipitation.  $K_S$  is the saturated hydraulic conductivity, which was used for the calculation of water transport in Equation B1. Unit conversion was considered in the calculation process of the model.

**13. Line 289: The unit of $Ei'_{4}$ $\rho a$ , cp, and ra should be added.**

Reply: We apologize for these omissions. We will supplement these contents as follows: E is the sum of the surface sublimation and evaporation rates (mm/day);  $\rho_a$  is the air density (kg/m3); CP is the constant pressure specific heat of the air (MJ/kg/°C); ra is the aerodynamic resistance (day/m).

**14. Line 290: ra is aerodynamic resistance?**

Reply: Yes, we will supplement its definition in the revised version.

**15. Line 340: where can we found the calibrated parameters?**

Reply: The calibrated parameters were in Line 344-354.

16. Line 454: Figure 10 In order to prove the conclusion in this paper that WEP-QTP can better simulate the measured runoff, it was suggested to plot the measured runoff data in the figure.

Reply: The simulated and measured runoff data are compared in Figures 5 and 9. Due to the limitations of the experimental site conditions, the hydrological cycle fluxes in Figure 10 have no measured values. Figure 10 was provided to compare the effect of model improvement on the runoff process. We will supplement this in the revised version.

17. Figure11 Legend and the scale is too small to read; It is recommended to mark the location of the three stations. What the source of the plotted data, measured snow thickness or the model simulation? If it is a map of measured snow, it is recommended to put it in the appendix. If it is a map of modeled results, suggest making a comparison with the actual measurement.

Reply: We apologize for the unclearness on this part. Figure 11 shows the snow thickness simulated by the model. The measured snow thickness was calibrated at the experimental site (Figure 6). We will modify this map as you suggested and supplement the site location.

**References**

Li J, Zhou Z, Wang H, et al. Development of WEP-COR model to simulate land surface water and energy budgets in a cold region [J]. Hydrology Research, 2019, 50(1): 99-116.

---

## Author Comment (AC2)

*The manuscript "Application of a new distributed hydrological model based on soil-gravel structure in the Niyang River Basin, Qinghai-Tibet Plateau" applied a new model which considered the impact of gravel on water and heat transfer, as well as the snow cover in the study region. This work was necessary for the study region which has soils with large portion of gravel to a certain depth. However, when I read the manuscript, I felt that the work needs some major changes in order to make it clearer. The authors developed both the soil and snow processes, but they did not show how each process have improved the results. Besides, the description on the results are quite subjective and I did not see confident quantitative descriptions in multiple places, I will put the specific comments below. In general, I suggest a substantial revision to make this work more attractive and interesting to readers.*

Dear Reviewer:

We appreciate the detailed and valuable comments, which have considerably improved the quality of our manuscript. Our responses to your comments are provided below.

*Specific comments:*

*1. Line 63-70: I like this paragraph about the soil formation of QTP, but the position of this paragraph can be moved upward before introducing the gravel content impacts on soil heat transfer.*

Reply: Thank you for this suggestion. We will revise this paragraph as you suggest.

*2. Line 76-79: I think this sentence is repeated as it was already mentioned above.*

Reply: Thanks for pointing this out, we'll revise this section by remove the repetition.

*3. Line 79: how to adjust parameters? I think you mean calibration, but the calibration always depends on the function as the goal, i.e. soil temperature from surface or from lower layer, and/or soil water simulation accuracy.*

Reply: Yes, it is parameter calibration. For the water and heat simulation of the underlying surface in the hydrological model, with the goal of improving the accuracy of soil water and soil temperature simulation, the hydrothermal parameters of the underlying surface were calibrated. In the revised version, we will clarify this part.

***4. So I would suggest the authors to be more direct on demonstration of what you focus and why it is important. This would connect the whole storyline of this manuscript. Otherwise, I would not think you tell a good story on your work and the importance of it would be heavily lowered.***

Reply: Thank you for your constructive feedback. We will modify the structure of the Introduction in the revised version, as you suggested, to help readers better understand the importance of our work.

***5. Line 84: I did not see the impacts of rain intensity in your introduction.***

Reply: We are sorry that we failed to make this clear enough in the previously submitted version. The runoff generation mechanism is different between the non-heavy and heavy rain scenarios. The non-heavy rain scenario indicates saturation-excess, and the heavy rain scenario indicates infiltration-excess. Therefore, in the WEP model, the runoff generation process was divided into two scenarios. In the revised version, we will clarify this part.

***6. Line 143: to avoid***

Reply: Thanks, we will correct it and check for other potential errors.

***7. Line 154: what is "and0"?***

Reply: The "0" was redundant. We will remove it and check for other potential errors.

***8. Line 181-182: this is repeated.***

Reply: Thanks for pointing this out, we'll remove the repetition.

**9. Line 349-350: how were they determined?**

Reply: These parameters were calibrated with actual measurement.

**10. Line 420: Figure 8, again, interestingly, the model was good for frozen period because liquid water was very low, but during the thawing period, i.e. March and April, the model starts showing large discrepancy from observations, which I am curious about the possible causes.**

Reply: The soil moisture simulation results of the WEP-QTP model in March and April were indeed not satisfactory, but there is still a clear advantage compared to the WEP-COR model. The lower accuracy in simulation can be attributed to soil inclusions within the gravel layers. Gravel layers are not homogeneous, but they were generalized in the model as a set of unified parameters; hence, there is a discrepancy in the model simulation results.

**11. I just wonder the improvement is more about the hydraulic process than the thermal process? I can not agree that soil temperature was obviously improved from Figure 7 as the dynamics is still not well represented by the WEP-QTP model.**

Reply: The improvement of the heat transfer process includes two aspects: one is to consider the thermal insulation effect of snow, and the other is to consider the thermodynamic parameters difference between gravel and soil. The addition of snow mainly reduces the temperature fluctuation in the surface layer, while the addition of gravel affects the heat conduction in the underlying layer. Impacted by the heterogeneity of the gravel layer, the improved model shows a discrepancy from the observations, but it is closer to the observations than the WEP-COR model.

**12. Section 3.3: these are not quantitative and a little subjective. I would like to see how the new model is better and with some quantitative results.**

Reply: Thanks for the comment and suggestion. In the revised version, we will add the discussion of the quantitative results in this section 3.3 to help readers understand the model improvements.

**13. Line 457: I am wondering, how you compare the new model with the old model if it was changed in different processes, as you have developed the gravel related processes, and now you have a new snow model. I will be confused on what caused the differences in results.**

Reply: By analyzing the sensitivity of snow and gravel layers to temperature simulation, we found that when only considering the influence of the thermodynamic parameters of gravel, the average RE was -4.38%, which is close to the average RE of WEP-COR (-3.6%). When only the insulation effect of snow was considered, the average RE was 0.70%, which is close to the average RE of WEP-QTP (0.08%). Therefore, snow is the main factor affecting the temporal and spatial variation of permafrost.

**14. Conclusions: I am not so confident on the results for frozen period in the whole manuscript as I mentioned above, the snow and soil processes were both developed and I did not see the impact from one single process. I therefore additional sensitivity work should be done to quantitatively show how each process affects the results.**

Reply: Thank you for your professional comment. As mentioned in the reply to question 13, we will add additional sensitivity analysis in the revised version to quantitatively show how each process impacts the simulation results. In addition, we will focus on revising the discussion section based on your comments above to enhance the persuasiveness of our conclusions.

---

## Author Response (AR1)

**Authors Response to Reviewer #1**

*The paper is interesting to read, which presented the development of Water and Energy transfer Processes model in the Qinghai–Tibet Plateau (WEP-QTP) that modified based on the original Water and Energy transfer Processes model in Cold Regions (WEP-COR). In the presented model, the vadose zone processes considered three strategies under different conditions: (1) a dualistic soil-gravel structure using the Richards equation under non-heavy rain in the nonfreeze–thaw period; (2) a multi-layer Green-Ampt model in a heavy rain scenario in the nonfreeze–thaw period; and (3) a hydrothermal coupling model based on the continuum of the snow-soil-gravel layer during the freeze—thaw period. The modified model was then verified with measured river discharge in Niyang River Basin by comparing the simulated groundwater.*

*The study adopted a new conceptualization of the water and energy transfer in Qinghai–Tibet Plateau, which is considered as novel. However, significant improvement is needed before the consideration of publication to Hydrology and Earth System Sciences.*

Dear Reviewer:

We appreciate the detailed and valuable comments, which have considerably improved the quality of our manuscript. Our responses to the comments are provided below.

**Major comments:**

*1. Author stated in Line 52-62 that the existence of gravel in soil can significantly affect the soil water content and water transport. However, coupling of soil water and heat transport may be still not fully achieved in the modified version of WEP-QTP. When the dualistic soil-gravel structure was used in the nonfreeze–thaw*

*period, the soil water transport may be decoupled with the thermal transport (see Line 296: "for soil and gravel layers, the average temperature was represented by the temperature of the middle layer"). It seems that the full coupling of water and heat transport can only be achieved for freeze—thaw period? Author should at least state whether the neglection of heat transport in nonfreeze–thaw period affect hydrological processes.*

Reply: Thank you for the comment and suggestion. In the non-freeze–thaw period, all the water was in a liquid state, and the heat conduction had a minor effect on the water migration process. The model detected the non-freeze-thaw period based on ice content and temperature of each computing unit. In that period, only the moisture simulation was performed for simulation efficiency. We made supplementary modifications in Section 2.2 to address this point.

"Under the fully thawed conditions, all the water was in a liquid state, and the heat conduction had a minor effect on the water migration process. Therefore, for simulation efficiency, only the moisture simulation was performed during this period."

*2. During the nonfreeze–thaw period, the soil hydrology was simulated with either a dualistic soil-gravel model or a Green-Ampt equation, and the selection of the two options depend on whether the rainfall intensity exceeded 20 mm/day (Line 677). Why was such threshold selected? Would the dualistic model more suit to the high intensity rainfall?*

Reply: The runoff generation mechanism is different between the non-heavy and heavy rain scenarios: during non-heavy rain, there is saturation-excess, while during heavy rain, there is infiltration-excess. In the heavy rain scenario, the Richards model is unstable for soil hydrology simulation while the Green-Ampt model is stable and has high computational efficiency. Therefore, in the WEP model, this threshold value was used to divide the simulations into two scenarios. The added sentences in Appendix B are as follows:

"The WEP-COR model divides soil infiltration into two scenarios, heavy rain

infiltration and non-heavy rain infiltration, according to the different runoff generation mechanisms."

**3. In the schematic figure shown in Fig. 3 (a), the author proposed a dualistic soil-gravel model, it is not clear whether the dualistic model is similar to the dual-porosity model proposed by Greke and van Genuchten (1993). Moreover, author should clearly state how to separate the water flow in such dualistic pore system.**

Reply: Thank you for the insightful comment. In the dual-porosity model proposed by Greke and van Genuchten, the water transport medium is a mixture of soil and rock that is consistent from top to bottom; hence, the model generalized the medium into two systems: macropore and matrix pore. Our research object was the upper and lower layer medium, with the thin soil layer and thick soil-gravel mixtures (SGM) layer, in the Qinghai–Tibet Plateau. Based on this, we generalized the medium as an upper and lower dualistic soil–gravel structure to simulate the processes of water and heat transport in different periods. For the sake of clarity, we have redrawn Figure 3. In this model, the water flow is not separated like in the dual-porosity model. In future research, we will refer to the dual-porosity model to improve the mathematical description method of water transport in the SGM layer.

**4. The soil water retention curve was described with van Genuchten model in Eq. B2 (Line 685), while the soil hydraulic conductivity function adopted a power functionï¼ˆB3ï¼‰which is similar to that was used in Brooks-Corey model. Besides, the parameter n in Eq.B3 also adopted Mualem's constant (Line 692). Such combination may be acceptable only if more cautions were taken for the parameterizations. Author should clarify why chosen to combine the selected soil water retention curve and soil hydraulic function, and how these soil hydraulic parameters were specified for distributed hydrological modeling.**

Reply: We apologize for the lack of clarity in this part. Equation B3 is not Brooks-Corey model, but Mualem model (Mualem Y, 1986):

$$K(\theta_l) = \begin{cases} K_S & \theta_l = \theta_s \\ K_S \left( \dfrac{\theta_l - \theta_r}{\theta_s - \theta_r} \right)^{\Omega} & \theta_l \neq \theta_s \end{cases}$$

where K $(\theta_l)$ is the hydraulic conductivity (cm/s) of the soil layer when the liquid water content is $\theta_l$; $K_s$ is the saturated hydraulic conductivity of the soil temperature correction (cm/s); $\theta_r$ is the residual water content of the soil layer (cm$^3$/cm$^3$); $\theta_s$ is the saturated water content (cm$^3$/cm$^3$); $\Omega$ is Mualem's constant calculated as follows:

$$\Omega = 3 + 0.015 \int_{\theta_{15atm}}^{\theta_s} \gamma_\omega \, h d\theta$$

where $\gamma_\omega$ is the specific weight of water and $h$ is the capillary head (cm). $\Omega$ is a constant for the specified soil.

Van Genuchten model combined with Mualem model has been widely used (Khaleel R, 1995; Vereecken H, 2010), and the combined application of the two models has been verified in the previous WEP COR model (Li 2019), which performs well in simulating water transport in frozen soil. These two models are mainly used to calculate the unsaturated soil hydraulic diffusivity $D(\theta_l)$ and the hydraulic conductivity $K(\theta_l)$ in Equation B1. Equation B1 was used to calculate the vertical movement of water in unsaturated soils.

We supplemented a description of the method used to calculate the hydraulic conductivity, supplemented the references accordingly, and replaced the repeated variable names.

**References:**

[1] Mualem Y. Hydraulic conductivity of unsaturated soils: Prediction and formulas[J]. Methods of Soil Analysis: Part 1 Physical and Mineralogical Methods, 1986, 5: 799-823.

[2] Khaleel R, Relyea J F, Conca J L. Evaluation of van Genuchten – Mualem relationships to estimate unsaturated hydraulic conductivity at low water contents[J]. Water Resources Research, 1995, 31(11): 2659-2668.

[3] Vereecken H, Weynants M, Javaux M, et al. Using pedotransfer functions to estimate the van Genuchten – Mualem soil hydraulic properties: A review[J]. Vadose Zone Journal, 2010, 9(4): 795-820.

[4] Li J, Zhou Z, Wang H, et al. Development of WEP-COR model to simulate land surface water and energy budgets in a cold region [J]. Hydrology Research, 2019, 50(1): 99-116.

**5. In Page 20, Fig.8, why the simulated soil moisture differed between the two models in a freeze–thaw period (Line 414)? Modification in the proposed model may be solely focused on the nonfreeze–thaw period.**

Reply: Differences in simulated soil moisture were caused by different model structures. The WEP-COR model did not consider the layered geological features of the Qinghai-Tibet Plateau; the simulation object was homogeneous soil. Therefore, the simulated moisture of the WEP-COR model changed gradually in the vertical direction, and a large difference between simulated and measured values occurred below 40 cm (the soil layer thickness at the experimental site is 40 cm, with soil-gravel mixtures (SGM) layer below 40 cm.). The WEP-QTP model took this geological structure into account. The SGM layer has higher hydraulic conductivity and lower water-retention capacity, resulting in lower simulated values of water content than the WEP-COR model. The simulated results from the improved WEP-QTP model were closer to the measured values.

We have rewritten the relevant exposition of Section 3.2.2 as follows.

"For the 20−40 cm soil layers, their water holding capacity in the WEP-QTP model was greater than that of the SGM layer under it, so that the moisture simulated by the WEP-QTP model was greater than that of the WEP-COR model; Below 40 cm, the simulation difference between the two models starts to emerge clearly. The simulated moisture of the WEP-COR model changed gradually in the vertical direction and was greater than the measured value. While the moisture simulated by the WEP-QTP model was lower and closer to the measured value. This was attributed to the fact that the thickness of the soil layer at the experimental site was 40 cm, and below it is the SGM layer with higher hydraulic conductivity and lower water holding capacity. The WEP-COR model did not take into account the influence of the layered geological features on water migration."

The model improved in both the freeze-thaw period and non-freeze-thaw period. Like in the non-freeze-thaw period, the revised formula for water retention properties of the soil–gravel mixture was used to describe water retention curves for the lower gravel layer during the freeze-thaw period (Equation 1). The saturated hydraulic conductivity ($K_s$) of the soil or gravel layer was corrected by temperature (Equation 6). There are also improvements to the heat transport calculations; please see Section 2.2.2 for details.

**6. All the figures have poor resolution. Please consider replacing all of them.**

Reply: Thank you for your suggestion. In the revised version, we have replaced all figures with high-quality figures and improved the figure layout.

***Minor comments:***

**7. Line 132: "Temperature" should be "temperature"**

Reply: Thank you. We have corrected it and checked for other potential errors.

**8. Line 154: The "0" may be redundant.**

Reply: Sorry for this mistake. We have corrected it and checked for other potential errors.

**9. Line 164: The citation of MODIS data should be added.**

Reply: The citation of MODIS data is: https://ladsweb.modaps.eosdis.nasa.gov/search/. We added it in the revised version (Line 160).

**10. Line 166: The citation China's second glacier inventory data set should be added.**

Reply: Line 165 showed this citation (http://westdc.westgis.ac.cn/).

**11. Line 167: add the citation of Water and Energy transfer Processes in Cold Regions (WEP-COR) model.**

Reply: Thank you. We added this citation in the revised version.

**12. Line 267: The unit of saturated hydraulic conductivity Ks and snow water equivalent S should be consistent.**

Reply: $S$ is the daily variation of snow water equivalent (mm/d), which is consistent with the unit of daily precipitation. $K_S$ is the saturated hydraulic conductivity, which was used for the calculation of water transport in Equation B1. Unit conversion was considered in the calculation process of the model.

**13. Line 289: The unit of E, ρa, cp, and ra should be added.**

Reply: We apologize for these omissions. In the revised version, we rewrote the calculation method of energy and supplemented the relevant units.

**14. Line 290: ra is aerodynamic resistance?**

Reply: Yes, $r_a$ is aerodynamic resistance (day/m).

**15. Line 340: where can we found the calibrated parameters?**

Reply: The calibrated parameters were in Line 347-352.

**16. Line 454: Figure 10 In order to prove the conclusion in this paper that WEP-QTP can better simulate the measured runoff, it was suggested to plot the measured runoff data in the figure.**

Reply: The simulated and measured runoff data are compared in Figures 5 and 9. Due to the limitations of the experimental site conditions, the hydrological cycle fluxes in Figure 10 have no measured values. Figure 10 was provided to compare the effect of model improvement on the hydrological cycle flux.

Relevant sentences have been corrected to avoid ambiguity.

Change from

"However, in the WEP-COR model, this part of water mostly formed the peak flow which was inconsistent with the measured value"

to

"However, in the WEP-COR model, this part of water mostly formed the peak flow during the flood season, which far exceeds the measured value of the flow (Fig. 9)."

*17. Figure11 Legend and the scale is too small to read; It is recommended to mark the location of the three stations. What the source of the plotted data, measured snow thickness or the model simulation? If it is a map of measured snow, it is recommended to put it in the appendix. If it is a map of modeled results, suggest making a comparison with the actual measurement.*

Reply: We apologize for the lack of clarity in this part. Figure 11 shows the snow thickness simulated by the model. The measured snow thickness was calibrated at the experimental site (Figure 6). We have modified this map as you suggested, and supplemented the three stations.

**Authors Response to Reviewer #2**

*The manuscript "Application of a new distributed hydrological model based on soil-gravel structure in the Niyang River Basin, Qinghai-Tibet Plateau" applied a new model which considered the impact of gravel on water and heat transfer, as well as the snow cover in the study region. This work was necessary for the study region which has soils with large portion of gravel to a certain depth. However, when I read the manuscript, I felt that the work needs some major changes in order to make it clearer. The authors developed both the soil and snow processes, but they did not show how each process have improved the results. Besides, the description on the results are quite subjective and I did not see confident quantitative descriptions in multiple places, I will put the specific comments below. In general, I suggest a substantial revision to make this work more attractive and interesting to readers.*

Dear Reviewer:

We appreciate the detailed and valuable comments, which have considerably improved the quality of our manuscript. Our responses to your comments are provided below.

*Specific comments:*

*1. Line 63-70: I like this paragraph about the soil formation of QTP, but the position of this paragraph can be moved upward before introducing the gravel content impacts on soil heat transfer.*

Reply: Thank you for this suggestion. We revised this paragraph as you suggested.

*2. Line 76-79: I think this sentence is repeated as it was already mentioned above.*

Reply: Thank you for pointing this out. We removed the repetition.

*3. Line 79: how to adjust parameters? I think you mean calibration, but the*

***calibration always depends on the function as the goal, i.e. soil temperature from surface or from lower layer, and/or soil water simulation accuracy.***

Reply: Yes, it is parameter calibration. For the water and heat simulation of the underlying surface in the hydrological model, with the goal of improving the accuracy of soil water and soil temperature simulation, the hydrothermal parameters of the underlying surface were calibrated. In the revised version, we have clarified this part.

Change from

"By adjusting the model parameters, the hydrothermal simulation effect of the QTP can be improved to a certain extent"

To

"By calibrating the parameters, the difference in water and heat transfer between SGM and soil can be hidden to some extent, and the simulation effect can be improved."

***4. So I would suggest the authors to be more direct on demonstration of what you focus and why it is important. This would connect the whole storyline of this manuscript. Otherwise, I would not think you tell a good story on your work and the importance of it would be heavily lowered.***

Reply: Thank you for your constructive feedback. We pointed out the importance of this work at the beginning of the Introduction as follows:

"Unraveling the runoff formation and hydrological regulation mechanisms in this region is the basis for studying the response patterns of hydrological processes under climate change conditions."

At the end of the Introduction, we redescribed the key issues we focused on as follows:

"It is particularly important to consider the influence of the unique underlying surface of the QTP on the water-heat transport and water circulation, especially in the context of global climate change and frozen soil degradation. The objectives of the present study were to…"

In addition, we also modified the structure of the Introduction, as you suggested, to

help readers better understand the importance of our work.

**5. Line 84: I did not see the impacts of rain intensity in your introduction.**

Reply: We apologize for failing to make this clear enough in the previously submitted version. The runoff generation mechanisms are different between the non-heavy and heavy rain scenarios. The non-heavy rain scenario indicates saturation-excess, and the heavy rain scenario indicates infiltration-excess. Therefore, in the WEP model, the runoff generation process was divided into two scenarios. The added sentences in Appendix B are as follows:

"The WEP-COR model divides soil infiltration into two scenarios, heavy rain infiltration and non-heavy rain infiltration, according to the different runoff generation mechanisms."

**6. Line 143: to avoid**

Reply: Thank you. We corrected it and checked for other potential errors.

**7. Line 154: what is "and0"?**

Reply: The "0" was redundant. We removed it and checked for other potential errors.

**8. Line 181-182: this is repeated.**

Reply: Thank you for pointing this out. We removed the repetition.

**9. Line 349-350: how were they determined?**

Reply: These parameters were determined by calibrating runoff, soil moisture, and temperature simulation results on the basis of measured and empirical values.

**10. Line 420: Figure 8, again, interestingly, the model was good for frozen period because liquid water was very low, but during the thawing period, i.e. March and April, the model starts showing large discrepancy from observations, which I am curious about the possible causes.**

Reply: The soil moisture simulation results of the WEP-QTP model in March and April were indeed not satisfactory, but there is still a clear advantage compared to the WEP-COR model. The lower accuracy in simulation can be attributed to soil inclusions within the gravel layers. Gravel layers are not homogeneous, but they were generalized in the model as a set of unified parameters; hence, there is a discrepancy in the model simulation results. We added the following description in this part:

"However, as the actual condition of the underlying surface cannot be homogeneous like in the model generalization, there was a discrepancy in the model simulation results. Soil with low gravel content may be present near the 160 cm layer. In March and April after the snow melts, the snowmelt water infiltrates to this layer and was more likely to reside there, and the measured values of this layer was between the simulated values of the WEP-QTP and WEP-COR."

**11. I just wonder the improvement is more about the hydraulic process than the thermal process? I can not agree that soil temperature was obviously improved from Figure 7 as the dynamics is still not well represented by the WEP-QTP model.**

Reply: The improvement of the heat transfer process includes two aspects: one is to consider the thermal insulation effect of snow, and the other is to consider the thermodynamic parameter difference between gravel and soil. The addition of snow mainly reduces the temperature fluctuation in the surface layer, while the addition of gravel affects the heat conduction in the underlying layer. Impacted by the heterogeneity of the gravel layer, the improved model exhibits a discrepancy from the observations, but it is closer to the observations than the WEP-COR model.

**12. Section 3.3: these are not quantitative and a little subjective. I would like to see how the new model is better and with some quantitative results.**

Reply: Thank you for the comment and suggestion. In the revised version, we added the discussion of the quantitative results in this section to help readers understand the model improvements.

*13. Line 457: I am wondering, how you compare the new model with the old model if it was changed in different processes, as you have developed the gravel related processes, and now you have a new snow model. I will be confused on what caused the differences in results.*

Reply: By analyzing the sensitivity of snow and gravel layers to temperature simulation, we found that both snow and soil-gravel mixtures (SGM) improved the accuracy of temperature simulation, but the locations of their main effects on heat transfer were different. Snow cover reduces the heat transfer and temperature fluctuations of the surface layer, which improves the simulation accuracy of the surface soil temperature. For layers 40 cm and below, the influence of SGM was dominant. Overall, the addition of the snow layer reduces the temperature fluctuations in the topsoil and, in conjunction with the SGM layer, corrects the original model's overestimation of the freeze-thaw rate. We added a detailed discussion in the revised version in Lines 396-407.

*14. Conclusions: I am not so confident on the results for frozen period in the whole manuscript as I mentioned above, the snow and soil processes were both developed and I did not see the impact from one single process. I therefore additional sensitivity work should be done to quantitatively show how each process affects the results.*

Reply: Thank you for your professional comment. As mentioned in the reply to question 13, we added additional sensitivity analysis in the revised version to quantitatively show how each process impacts the simulation results. In addition, we revised the discussion section based on your comments above to enhance the persuasiveness of our conclusions.

**Authors Response to Reviewer #3**

*General comments*

*To tackle the question of the cryo-hydrology of Tibetan catchments under climate changes, the authors use an already publish cryo-hydrological model and improve it in two ways: they use a stratigraphy of a soil lying on a soil with more gravels that they identify as widespread across the QTP and they account for the yearly cycles of growth and melt of a snowpack both thermally and hydrologically.*

*I think overall that the study is interesting because it contributes to give visibility of catchment scale cryo-hydrological modeling which is a key approach, currently under development, to understand how climate change will impact the water cycle in high mountain regions. I also think that the field observations on the stratigraphy give an important added value to the paper and they should be presented with more details to better assess the characterization of this stratigraphy and its spatial distribution. The study also conveys the interesting message and demonstrates that the stratigraphy is important both regarding the hydrology and the thermal behavior of the model setup. The importance of representing the snowpack is also interesting but probably more obvious, as you cannot really model realistically the hydrology and thermal regime of a catchment with a significant seasonal snowpack without accounting for it.*

*So for me, this is where the added value of the article is: interesting field observations that motivate an interesting sensitivity test on stratigraphy. Even though visible, the improvements on the model outputs are not stunning on the provided graphs but I believe that if the authors could provide observations or something we could consider "reality" on figures 10 and 12 this feeling could be improved. I realize that this study represents an important amount of work, the objective is clear and I think the global structure of the article is relevant.*

*Yet the manuscript has important flaws that needs major modifications. I detail that below.*

Dear Reviewer:

We appreciate the detailed and valuable comments, which we addressed to substantially improve the quality of our manuscript. Our detailed responses to your comments are provided below.

*1. In the first place, according to me, the idea that the whole QTP, a region of millions of km2 presents a more or less uniform stratigraphy with a layer of soil on top of a layer of gravels makes very little sense to me. I had a look at the studied catchment on Google Earth and I saw steep rock walls, colluvium, torrential streams, glacial valleys, all types of moraines, alluvial fans, braided fluvial systems… Meaning I saw the normal variety of landscape processes I could have expected and there is no reason they all produce this uniform stratigraphy at the scale of the catchment, not counting the whole QTP ! Not to mention that most of the sampling points are in the low lying parts of each valley. So the authors need to present better their field observations of the stratigraphy and discuss them in a much more cautious way.*

Reply: There is indeed a variety of landscapes in the catchment, as you can see on Google Earth. The WEP-COR model generalizes this and divides the underlying surface into five classes: water body, soil-vegetation, irrigated farmland, non-irrigated farmland, and impervious area (Lines 673-675). The water body class includes rivers, lakes and glaciers. The soil–vegetation class includes bare land, grassland and woodland. Impervious area consists of urban buildings and impervious surfaces. These five classes of underlying surfaces represent all landscapes of the catchment in the model. Simulation of soil water and heat transport is not applicable for water body and impervious area classes, while for other underlying surfaces, the improvement of the water and heat transport model was achieved in this study.

Apologies for the lack of clarity regarding the surface classification in the previous

version of the paper. In the revised version of Section 2.2 (Lines 179-183, Lines 190-192), we added supplementary information accordingly.

***2. L76-77 "The geological features of the QTP are generally thin soil layers above the thick gravel layers with clear boundaries between them." -> Any study supporting this at a large scale?***

Reply: Affected by geological structure and freeze-thaw cycles, the phenomenon of the gravel layer under the thin soil layer is prevalent in the Qinghai-Tibet Plateau (Sun, 1996; Yang et al., 2009; Chen et al., 2015; Deng et al., 2019; ). Moreover, we confirmed this phenomenon in the present study with field observations in the study area, as shown in the figure below.

[Figure]

**References**

[1] Chen H, Nan Z, Zhao L, et al. Noah modelling of the permafrost distribution and characteristics in the West Kunlun area, Qinghai-Tibet Plateau, China[J]. Permafrost and Periglacial Processes, 2015, 26(2): 160-174.

[2] Deng, D., Wang, C., and Peng, P.: Basic Characteristics and Evolution of Geological Structures in the Eastern Margin of the Qinghai-Tibet Plateau, Earth Sciences Research Journal, 23, 283-291, 2019.

[3] Sun H L, The formation and evolution of the Qinghai-Tibet Plateau, China: Shanghai Science and Technology Press, 1996, ISBN: 9787532340231.

[4] Yang K, Chen Y Y, Qin J. Some practical notes on the land surface modeling in the Tibetan Plateau[J]. Hydrology and Earth System Sciences, 2009, 13(5): 687-701.

*3. I also think mentioning "gravel" might be misleading. I think the authors should find a name to describe this type of formation (I see Wang et al. 2013 uses "soil-gravel mixtures" which sounds much more informative to me) because I think most readers reading "gravel" won't think about this unsorted slope deposits but rather to well sorted alluvial gravel formation that are highly conductive and then it might sound very counter intuitive. Especially when the author says "since gravel can neither conduct nor store water" and that it "hinders the movement of water".*

Reply: Thank you for the comment and suggestion. In the revised version, we replaced "gravel" with "soil-gravel mixture (SGM) layer".

*4. Another big problem lies in the presentation of the models. Many points are unclear or don't make sense. I detail this below. A good example of this is the method used to calculate snow melt, which is first said to be based on a snow depth threshold between contour bands rather than on climatic variables (even though it is later the case, when the author mentions a PDD method). Not mentioning the so-called "snow sliding". I suspect the problem doesn't lie in the model itself since the output looks good, but rather in the description. I am happy to be shown I am wrong if it is the case, but from what I read and understand, there are major problems in the identifications of the processes and their representation as they are described (under the hypothesis that what is coded is different and correct).*

Reply: Apologies for not being clear enough about this part in the previous version of the manuscript. There are some problems in our description, which we addressed in detail in answers to questions 10 and 11.

*5. Finally, I started correcting the English but I am not a native speaker and the task here is too important for a scientific reviewer. So I would recommend having the manuscript proofread by a native speaker because I think many formulations can be improved (see the examples I give below for the beginning).*

Reply: Apologies for the grammatical and stylistic errors. The language of the revised manuscript has been improved by a native English speaker/professional science

editing service.

*Key specific comments*

*6. I had a look at Wang 2013 and, unless I am mistaken, one empirical parameter is missing in equation 1, where (1 - Wgravel) should be (1 – B x Wgravel) (Wang writes it B). Is this a mistake or is it me who missed a transformation of the equation ?*

Reply: Thank you for the professional questions. This is a mistake, and we corrected it in the revised version.

*7. Since the model uses different types of equations depending on a threshold for rain, what happens at the transition between normal and heavy rain ? What if during a rain event the threshold is crossed, how smooth is the model in this regard ? Are there some data processing methods to smooth a potentially sharp transition ?*

Reply: Thank you for the insightful comment. The flow in the model was simulated by day. When the daily precipitation exceeded the threshold, the Green-Ampt model was used. A sharp transition in the flow process has not been detected in the current simulations, but your question is very insightful, and we will conduct a series of studies on it in our future research.

*8. L242*

*"The large portion of gravel in the gravel layer causes the formation of macropores, which are connected to form a fast channel for transporting water during heavy rains"*

*I am totally confused here. As I said earlier, it is indeed the general way gravel formations are treated (as a conductive layer). But before the authors wrote :*

*"However, since gravel can neither conduct nor store water, the gravel […] hinders the movement of water and affects the water retention curves"*

*So how does all of this work together? The authors have been, since the beginning,*

*using studies on soils containing rock fragments to support a certain type of behavior from their bottom layer and now they argue towards another behavior because they have been calling these fragments "gravels". And at this stage I am confused. Maybe I missed something here but if so, there is a lack of pedagogy/clarity in the way this model and stratigraphy works together.*

Reply: Apologies for the lack of clarity this part. As for the answer in question 3, according to your suggestion, we used "SGM" instead of "gravel" to distinguish the concept of "gravel" and "soil-gravel mixture". There are macropores in the SGM layer. In a saturated state, the macropores form a fast channel for transporting water. However, when the SGM layer is in an unsaturated state, the water mainly moves under the actions of the matrix potential and gravitational potential. Thus, in an unsaturated state, the macropores do not work, and the gravel will hinder the movement of water.

In Section 2.2.1 of the revised version, we change from

"However, since gravel can neither conduct nor store water, the gravel, which accounts for 50–65% of the gravel layer, hinders the movement of water and affects the water retention curves."

to

"However, in the unsaturated state, since gravel can neither conduct nor store water, the gravel, which accounts for 50–65% of the SGM layer, hinders the movement of water and affects the water retention curves."

Change from

"The large portion of gravel in the gravel layer causes the formation of macropores, which are connected to form a fast channel for transporting water during heavy rains."

to

"The gravel increases the porosity in the SGM layers, and these pores connect to form a fast channel for transporting water during heavy rains."

**9. L255**

*"(the contour bands)"*

*The authors need to explain more clearly how the model works in the main text. I had to read the appendix to get a clearer idea of how this works. It is an unusual approach so it needs to be commented on. What decides the shape and extension of a band ? And it also needs some statistics: How many bands ? Average size of a band ? Average elevation range within a band ? I would also like to see a map with all these bands to see how all this looks like. Otherwise, what is done here remains very abstract.*

Reply: Thank you for your valuable suggestion. We added the working principle of the model in Section 2.2 of the revised version, and the concept of contour bands is also supplemented accordingly. The added sentences are as follows:

"The WEP-COR model is a distributed hydrological model. To consider the impact of the topography and land cover on the water cycle in large basins, the spatial calculation unit of the WEP-COR model is contour bands (i.e. bands at different elevation intervals) inside small sub-basins. Each unit is classified into five classes: water body, soil–vegetation, irrigated farmland, non-irrigated farmland, and impervious area (The water body class includes rivers, lakes and glaciers. The soil–vegetation class includes bare land, grassland and woodland. Impervious area consists of urban buildings and impervious surfaces). The calculation result of the water and heat flux in each type was weighted by area to obtain the water and heat flux of the contour band."

The simulation methods used for each section were also added, accordingly, in Appendix B (Lines 675-681).

The division of sub-basins contour bands was supplemented in Section 3.1 (Lines 334-336) as follows:

"The Niyang River Basin was divided into 217 sub-basins, and each sub-basin was divided into 1~10 contour bands as the basic calculation unit according to the elevation. The basin was divided into 871 contour bands. The average area of the contour bands is 20 $km^2$."

The sub-basins and the contour bands are shown in the figure below.

[Figure]

**10. L256-259**

*"When the snow thickness difference between two calculation units exceeded this threshold, snow meltdown occurred. The snow in the higher-altitude calculation unit slides into the next unit until the two units had the same snow thickness. The daily variation of snow water equivalent was calculated as follows"*

*Why is melt based on thickness difference ? Melt should be based on the climate input. All this makes very little sense, but I suspect these problems lie in the model description and not in the model itself.*

Reply: Apologies for the unclear description. We corrected it in the revised version as follows:

"When the difference in snow thickness between two adjacent contour bands exceeds this threshold, an avalanche occurs between those contour bands. The snow in the higher-altitude contour band slides into the lower band until the two bands equalize in snow thickness."

**11. L262-263**

*"when the difference in snow thickness between contour bands in the same*

*sub-basin exceeds the threshold, the snow slides downwards until the snow thickness"*

*What is snow sliding ? I never heard of that and found nothing relevant on the net. The 2 important redistribution mechanisms I can think of are wind drift and avalanche. Snow creep also exists but is marginal in comparison. So what is the process here ?*

Reply: In our study, "snow sliding" means avalanche, i.e. snow collapse driven by elevation difference. In the model, we generalized the avalanche as the redistribution of snow between two adjacent contour bands. In the revised version, we replaced the terms with "avalanche".

*12. L290-292*

*"G is the heat flux (MJ/m2/d) conducted into the snow or soil, which was determined by the temperature difference between the soil or snow and the atmosphere near the surface. The above equation was combined with the ground heat conduction and energy balance equations"*

*I think this is wrong. G is the energy input in the ground that is used to drive heat conduction after the surface energy balance equation has been applied. So G is not derived from the atmosphere temperature near the surface, but H is. G is what you get when you sum the energy fluxes from the radiations and turbulent fluxes. Another problem is that the end of the sentence talks about heat conduction and energy balance equations. Conduction has not been introduced but energy balance is actually equation 7.*

Reply: Apologies for the inaccurate description of this process. This part introduced the energy balance equation used in the model and the calculation method of each energy flux. *G* was calculated from the temperature difference between the underlying surface and the atmosphere near the surface and was the heat flux conducted into the underlying surface. The heat conduction in the underlying surface was only related to *G*. In Section 2.2.2 of the revised version (Lines 284-289), we simplified this part and provided the calculation method of *G*.

*13. L294*

*"We […] simplified the calculation by solving the H according to the energy balance equation after calculating the LE"*

*Well that is not what the authors say before. Equation 9 is clearly a way to calculate H from temperature inputs. It is impossible to deduce H from the energy balance equation because you deduce G from this equation knowing all the other terms.*

Reply: Apologies for not being clear in the previous version of the manuscript on this part pointed out by you. As we replied in question 12, we calculated $G$ as follows:

$$G = C_{Vu}d_u(T_a - T_u)$$

where, $C_{Vu}$ is the volumetric heat capacity of the underlying surface (MJ/m$^3$/°C); $d_u$ is the depth of the underlying surface affected by heat conduction (m); $T_a$ is the air temperature on the day of simulation (°C); and $T_u$ is the surface temperature of the underlying surface on the day before simulation (°C).

*14. L297*

*"The temperature difference between the atmosphere and the surface is the source of heat conduction"*

*Why say this after calculating the surface energy balance ? The surface energy balance enables to calculate the energy change of the top cell, to work with temperature, the authors can then do ΔT = ΔE/Cp. Saying what I quote here after detailing an SEB module is more than confusing.*

*I don't have this level of problematic issues with the rest of the paper. Yet I think that in general, the text of the result and discussion section could be lighter and more concise.*

Reply: Apologies for not being clear enough in the previous version of the manuscript on the aspects pointed out by you. For clarification of the energy calculation part, you can refer to our replies to questions 12 and 13 here.

*Specific comments*

**15. I don't know where to put it so I write it here: to be able to understand what the new stratigraphy brings we need to have access to the WEP-COR stratigraphy, on Figure 3 for example.**

Reply: The WEP-COR stratigraphy was shown in the previous version. See Figure B1(b) in the Appendix B for details.

**16. L16**

**"The Qinghai-Tibet Plateau has a thin soil layer on top of a thick gravel layer"**

**I have 2 problems with this abstract opening:**

**Problem 1: See my previous comments, this cannot be true at the scale of a region as large as the QTP where one can find mountain peaks, peatlands, moraines, alluvial fans, blocky terrain… I suggest writing something like "For hydrological purposes, simplifying the representation of the QTP subsurface conditions to a thin soil layer on top of a thick gravel layer…" but this needs to be either demonstrated in a previous paper or in the present paper.**

Reply: Thank you for the insightful comments. For the proof of the geological structure in the Qinghai-Tibet Plateau and the generalization method of the different underlying surfaces in the model, you can refer to the answers to questions 1 and 2.

**17. Problem 2: I guess this is just a matter of personal preference, but I would recommend to start the abstract with a bit of context on what big question this study works with. Hydrology in mountainous cold regions and climate change…**

Reply: Thank you for your professional suggestion. In the revised version, we revised the abstract according to your recommendation. The added sentences are as follows:

"Runoff formation and hydrologic regulation mechanisms in mountainous cold regions are the basis for investigating the response patterns of hydrological processes under climate change."

**18. L41-42**

*"plays an important role in ensuring the security of water resources in China and Southeast Asia"*

*Needs to be supported by a reference.*

Reply: We added more references as follows:

[1] Liu X, Yang W, Zhao H, et al. Effects of the freeze-thaw cycle on potential evapotranspiration in the permafrost regions of the Qinghai-Tibet Plateau, China[J]. Science of the Total Environment, 2019, 687: 257-266.

[2] Yu L, Feng C Y. Recent progress in climate change over Tibetan Plateau[J]. Plateau and Mountain Meteorology Research, 2012, 32(3): 84-88.

*19. L43-44*

*"cannot be ignored"*

*Needs also a reference. The sentence is also surprising. The authors could start the sentence by "The extensive glacier…" list the items and end the sentence with "have a major impact on the water cycle…"*

Reply: Thank you for your professional suggestion. We modified this sentence as you suggested:

"The extensive glaciers, snow cover, permanently and seasonally frozen soil have major impacts on the water cycle."

Additional references are as follows:

[1] Yongjian D, Shiqiang Z, Jinkui W U, et al. Recent progress on studies on cryospheric hydrological processes changes in China[J]. Advances in Water Science, 2020, 31(5): 690-702.

[2] Zhiwei L I, Guo'an Y U, Mengzhen X U, et al. Progress in studies on river morphodynamics in Qinghai-Tibet Plateau[J]. Advances in Water Science, 2016, 27(4): 617-628.

*20. L44-45*

*"On the surface of seasonal frozen soil and permafrost, seasonal thaw layers alternately freeze and thaw as seasons change."*

***This is a convoluted way to say that both in permafrost and permafrost free areas, the ground undergoes seasonal freezing.***

Reply: Thank you for pointing out this deficiency. We modified it according to your suggestion.

**21. L63-67**

***My expertise on the topic is limited but this section on the links between tectonics, sedimentology and granulometry of the Quaternary sediments could be better phrased and states obvious things that don't show particular relevance for the study. I don't understand the message the authors want to convey that is important for the paper.***

Reply: Thank you for the comments. This paragraph was used to introduce the formation of the special underlying surface structure in the QTP. In the revised version, we simplified the description as follows:

"During the continuous QTP uplift, a series of ascending areas (denuded areas) and descending areas (deposited areas) have been formed. Quaternary deposits are generally thinner in denuded areas and thicker in deposited areas (valleys, plain). As a result of the surface uplift and collision of the Indian plate with the Eurasian plate, there are many gravel and rock fragments, which are soil-gravel mixtures (SGM), in the QTP Quaternary sediments."

**22. L105**

***"8 °C"***

***Add the average elevation associated to this mean temperature***

Reply: The average elevation of the catchment is 4688.6 m and this added to Section 2.1.1 (Line 106) of the revised version.

**23. L111-112**

***"Permafrost accounts for approximately 23.65%, mainly distributed in the upper reaches ofthe basin and the high-altitude areas on both sides of the mainstream."***

*Reference for this value ? Also I doubt one can reach such a precision in the significant numbers of the percentage.*

Reply: This value was calculated from ground temperature in ArcGIS according to the definition of permafrost: ground that remains at or below 0 °C for at least two consecutive years (Biskaborn et al., 2019; Dobinski W, 2011). We have reduced the significant digits to the whole number, 24% of the area under permafrost.

**References:**

[1] Biskaborn B K, Smith S L, Noetzli J, et al. Permafrost is warming at a global scale[J]. Nature Communications, 2019, 10(1): 1-11.

[2] Chen H, Nan Z, Zhao L, et al. Noah modelling of the permafrost distribution and characteristics in the West Kunlun area, Qinghai-Tibet Plateau, China[J]. Permafrost and Periglacial Processes, 2015, 26(2): 160-174.

[3] Dobinski W. Permafrost[J]. Earth-Science Reviews, 2011, 108(3-4): 158-169.

*24. L113-114*

*"The annual average temperature of the experimental site is 5.28 â„ƒ, which is a seasonally frozen soil area."*

*It is the first time the authors mention this site, maybe introduce it first.*

Reply: Thank you for the comments. In the revised version, we incorporate this sentence into the introduction of experimental site in (Lines 112-113).

*25. Caption of Fig 2*

*Indicate where this is located on the map of Fig. 1*

Reply: Figure 2 was taken near the experimental site shown in Figure 1, and we made notes under this figure in the revised version.

*26. L172-173*

*"The volume of the glacier was calculated by the area-volume empirical formula (Grinsted, 2013; RadiÄ‡ and Hock, 2010)."*

*Does this give access to volume changes along time ?*

Reply: Yes, glacier volume changed over time. The area of glaciers was obtained from four remote sensing images from 1994, 2003, 2009, and 2015. We linearly interpolated the glacier volume calculated from the area, making its temporal change as the model input.

**27. L188-189**

**"According to the geological characteristics of the QTP, this study improved the hydrothermal simulation methods of the non-freeze–thaw period and the freeze–thaw period."**

**This is a conclusion, it should not be part of the methods.**

Reply: Thank you for this comment. We deleted this part from the Methods section in the revised version.

**28. L193-196**

**"In the non-freeze–thaw period, the calculation object of water movement was defined as the dualistic soil–gravel structure (Fig. 3a). The upper layer is soil whose thickness and number of layers are determined by the location of the calculation unit; the thickness of the soil layer gradually decreases from the foot to the peak of the mountain. The lower layer is the gravel layer (mixed layer of soil and gravel)."**

**This is really hard to read/understand, rephrase, with examples and tangible elements.**

Reply: We rewrote this sentence and illustrated it with Figure 3b:

"Under the fully thawed conditions, the calculation object of water movement was defined as the dualistic soil – gravel structure (Fig. 3a). The upper layer was soil, whose thickness is determined by the location of the calculation unit, and which decreased gradually from the foot to the peak of the mountain (Fig. 3b). The lower layer is the SGM layer."

[Figure]

Figure 3b: Snow–soil–gravel layered structure.

*29. L233-234*

*"Until the water has the same potential energy in the soil and the gravel, the INF breaks through the critical surface, and then the infiltration rate stabilizes (Fig. 4)."*

*I don't understand this part. First I am unsure that potential energy is the good terminology (i.e. potential energy of the water in a dam), I assume it is the pressure head. And if it is so, the Green-Ampt model does not calculate the pressure head, it calculates the volume of infiltrated water or the depth of the infiltration front. So I don't understand this sentence. Maybe I did not understand the situation correctly but then please clarify this point.*

Reply: Potential energy here refers to soil water potential, including solute potential (not considered in this study), matric, gravity, and pressure potentials. The Green-Ampt model was derived by combining the Darcy's law with the continuity principle (Green and Ampt, 1911). The volume of infiltrated water, depth of the infiltration front, and capillary suction pressure were used to calculate the potential gradient in Darcy's law. The specific derivation process can be found in the references.

**References:**

Green W H, Ampt G A. The flow of air and water through soils[J]. The Journal of Agricultural Science, 1911, 4: 1-24.

**30. Figure 4: Cumulative infiltration process of the WEP-QTP model**

*I don't understand this figure. Please give an explanation in the caption. Re-explain the letters. Why are there 2 dashed lines, are they different scenarios ? I see now that this part of the figure is modified from Jia et al. (2001). I think that it should be cited as a source element of the figure. Also, now that I found this image from Jia, I understand that what is represented are the successive wetting fronts. Yet what I don't understand is why we see these dashed curves. In the Green-Ampt model, the wetting front is horizontal.*

Reply: Sorry for the obscurity of this part. A dashed line represents the wetting front at a moment, and the dashed lines in the figure represent the wetting fronts at different times: $t_1$, $t_m$, and $t_{itf}$. The dashed line is the actual wetting front, but the Green-Ampt model equates it to a straight line separating the saturated soil above from the soil below. In the revised version, we adjusted this figure as you suggested.

[Figure]

**31. Eq2 and L233-241**

*Here again it is hard to understand what the authors are doing. Where does this equation come from? Classically, the infiltration rate tends towards K because F(t) is at the place of Fitf here. But Fitf is hard to understand as it is a finite quantity and not a variable (i.e. the cumulative infiltration when the front breaks through). Also why ksoil when working with the "gravel" layers ? And What are "error*

*caused by the different soil moisture content of the soil above the interface". I think this paragraph needs more pedagogy to avoid giving the feeling that the authors are doing their own cooking with some well-established equations.*

Reply: Equation 2 was improved from Equation B4, which calculates the stable infiltration rate after the infiltration front penetrates the soil-gravel interface. Because the saturated hydraulic conductivity of the upper soil was the upper limit of the infiltration rate of the lower soil-gravel mixtures layer, $K_{soil}$ was used. The detailed calculation process of the infiltration rate and the calculation method of each parameter can be found in Equations B4-B6 in the appendix.

In the revised version, we added the following sentences to help readers better understand our work:

"Therefore, a new infiltration model was proposed by improving the method of infiltration rate in the multi-layer Green–Ampt equation (Equation B4 in Appendix B). The stable infiltration rate after INF breaking through Layer *itf* was calculated as follows..."

**32. L253**

*"determined by the snow water equivalent and snow density"*

*What is the approach ? Constant values ? Values derived from climate forcing data?*

Reply: Snow density was derived from climate forcing data. The thickness of the snow layer (cm) was calculated as follows:

$$h_{snow} = \frac{S_{snow} \times \rho_l}{10\rho_s}$$

where $S_{snow}$ is cumulative snow water equivalent (mm); $\rho_l$ is the densities of water (1000 kg/m$^3$); $\rho_S$ is the densities of snow (kg/m$^3$), which was derived from climate forcing data, as shown in Equations 11.

**33. Equation 7**

*"RN = LE + H + G"*

*A more comprehensive way to write it is G=… because it shows how what the authors derive from the climate forcing data is used as an input from the model. Also what about this equation when there is snow ? Is it also applied ?*

Reply: Thank you for your professional suggestion. In the reply to questions 12 and 13, we simplified the introduction of the energy calculation and provided the calculation method for *G*.

**34. Equation 10**

*How is the ice content linked to the temperature ? I assume the authors also use a soil freezing curve.*

Reply: The relationship between the water and heat transport of the frozen soil is mainly manifested in the dynamic balance of the moisture content of the unfrozen water and the negative temperature of the soil, which is shown in Equation B14 in the Appendix.

**35. L350**

*"the riverbed conductivity was approximately 5.184 m/d"*

*Where is this value used ? I feel some part of the model description is missing. From what I understood there is the soil and the gravel layers, now it seems there is a riverbed layer. I just looked for the word riverbed, it does not appear before part 3. I have the hardest time understanding how this model works and I am trying hard.*

Reply: The model improvement in this study does not involve the water exchange process of the river channel, but the riverbed conductivity is a sensitive parameter of the model; hence, the parameter calibration results were presented here.

**36. L350-351**

*"The thickness of the soil layer at the mountaintop, mountainside, and foot of the mountain was 0.4 m, 0.6 m, and 1.0 m, respectively."*

*Was there an attempt to characterize the stratigraphy based on the topography/morphology ? If so it is not mentioned in the method. And it is more*

*than necessary for the message this study wants to convey. So if this effort has been made, please explain it in the methods. Also as I said before, the stratigraphic observations should be presented in detail somewhere. They really contribute to the added-value of the paper.*

Reply: Yes, as mentioned in the reply to question 28, we have redrawn the generalized structure of the model and illustrated it in Figure 3b. The stratigraphic observations were detailed in Section 2.1.2 (Lines 142-150).

**37. Figure 11**

*The legend is so small, with the resolution I got for the figure (which is low) I cannot read it.*

Reply: Thank you for your suggestion. In the revised version, we included high-quality figures and improved the Figure 11 legend.

**38. L429-424**

*"There might have been a soil interlayer at cm, and the measured water content was between the simulated values of the WEP-QTP and WEP-COR. The average RE of WEP-COR was 33.74%, and that of WEP-QTP was smaller at -12.11%. WEP-QTP could reflect the influence of gravel on the vertical migration of water."*

*This discussion is really hard to follow. How speculative is the existence of this interlayer ? What is the physical process that makes it a relevant hypothesis ? I think if this suggestion is important it needs to be explained in more detail.*

Reply: Thank you for the insightful comment. The soil-gravel mixtures (SGM) layer under the topsoil is not homogeneous, but in the model, we assume homogeneity of this layer and use a uniform set of parameters to describe its water and heat properties. In the SGM layers, the observed value of water content in the 160-cm layer was smaller than that in other layers and between the simulated values of the WEP-QTP and WEP-COR; hence, we speculate that there may be a soil interlayer. We added the following description in this part:

"However, as the actual condition of the underlying surface cannot be homogeneous

like in the model generalization, there was a discrepancy in the model simulation results. Soil with low gravel content may be present near the 160cm layer. In March and April after the snow melts, the snowmelt water infiltrates to this layer and was more likely to reside there, and the measured values of this layer was between the simulated values of the WEP-QTP and WEP-COR."

*39. Figure 10 and L453-454*

*"was inconsistent with the measured value"*

*If possible the authors should add observations on figure 10. Since the main message of the study is to show the benefits of the new stratigraphy, these benefits are visible only when compared to observations.*

Reply: Apologies for the misunderstanding of the description here. The simulated and measured runoff data are compared in Figures 5 and 9. Due to the limitations of the experimental site conditions, the hydrological cycle fluxes in Figure 10 have no measured values. Figure 10 was provided to compare the effect of model improvement on the hydrological cycle flux.

Relevant sentences have been corrected to avoid ambiguity.

Change from

"However, in the WEP-COR model, this part of water mostly formed the peak flow which was inconsistent with the measured value"

to

"However, in the WEP-COR model, this part of water mostly formed the peak flow during the flood season, which far exceeded the measured value of the flow (Fig. 9)."

*40. L455*

*"In addition, snow cover significantly contributed to the inconsistencies between the temporal and spatial changes of the frozen soil in the two models, which in turn caused variations in the groundwater recharge and discharge process."*

*What supports these results ? Can the author provide a number or a graph that supports this ?*

Reply: We added sensitivity analysis of snow and SGM to temperature simulation in Section 3.2.1. We found that both snow and SGM improved the accuracy of temperature simulation, but the location of their main effects on heat transfer was different. The snow cover reduces the heat transfer and temperature fluctuations of the surface layer, which improves the simulation accuracy of the surface soil temperature. For layers 40 cm and below, the influence of SGM was dominant. Overall, the addition of the snow layer reduces the temperature fluctuations in the top soil and, in conjunction with the SGM layer, corrects the original model's overestimation of the freeze-thaw rate. We added a detailed discussion in the revised version in Lines 396-407.

At the same time, we also revised this sentence as follows:

"The addition of the snow layer not only reduced the temperature fluctuation of the surface soil, but also, with the help of the SGM layers, revised the original model's overestimation of the freeze-thaw speed."

**41. L656**

*"water body, soil–vegetation, irrigated farmland, non-irrigated farmland, and impervious area"*

*What about mountain bare lands above the tree line ? Which type was used?*

Reply: As we replied to question 1, "Soil-vegetation includes bare land, grassland, and woodland". Therefore, mountain bare lands above the tree line belong to soil–vegetation.

*Technical corrections*

**42. L16**

*"while"*

*I don't see the point of this "while", it is not putting 2 ideas in parallel. It is possible to end the first sentence before the while and restart with "This unique…"*

Reply: Thank you for your suggestion; we changed the text accordingly.

*43. L17*

*"To investigate the mechanism of the underlying surface structure on the…"*

*The sentence reads weird. The structure does not have mechanisms. And "to investigate" misses something like "the effect", "the consequences"… I would rephrase it to something like "To understand the impacts of this subsurface structure on the water cycle of QTP catchments…"*

Reply: Thank you for your professional suggestion. This has been revised.

*44. L17*

*"hydrothermal migration"*

*Is this term correct ? I googled it and found papers about the motion of magma. Which makes more sense because heat can trigger density gradients and thus motion. I do not really see it with fresh water on the continent.*

Reply: Thank you for pointing this out. In the revised version, we uniformly changed this term to "water-heat transport".

*45. L22*

*"the single soil"*

*I think "single" is not useful here.*

Reply: Thank you. We changed "single soil" to "uniform soil profile".

*46. L23-24*

*"In the non-freeze–thaw period"*

*When no phase change occurs in the ground*

Reply: Thank you for your suggestion; we applied it in the revised version.

*47. L35*

*"the "tailing" process after October"*

*the observed "tailing" process after October (it if is indeed what the authors mean)*

Reply: Thank you for your suggestion, we revised it in the revised version.

**48. L40**

**"typical"**

**This world carries little meaning. I would write "major"**

Reply: Thank you for your suggestion; we changed it to "major".

**49. L45-46**

**"Almost all ecological, hydrological, soil, and biological activities in the soil in cold regions occur here, hence it has been the focus of hydrological research in cold regions"**

**This sentence could be improved because of the "Almost all", "soil in the soil" etc. I would rephrase "This region has received a lot of attention regarding hydrological research because of the great variety of biological and physical processes occurring at the surface and subsurface…".**

Reply: Thank you for your suggestion; we changed this sentence accordingly.

**50. L84-85**

**"during the non-freeze–thaw period"**

**Under fully thawed conditions**

Reply: Thank you for your suggestion; "under fully thawed conditions" sounds clearer. We have made relevant changes.

**51. L86**

**"develop a hydrothermal coupling method"**

**This is unclear according to me, I would say "develop a modeling framework representing coupled heat and water transfers in the ground"**

Reply: Thank you for your suggestion, we have incorporated your suggestions in the revised version.

*52. L102*

*"litharenitewith"*

Reply: Apologies for this mistake; it should be "litharenite with" here. We have made relevant changes.

*53. L144-143*

*"stations. Avoid"*

Reply: Sorry for this mistake, we have fixed this error in the revised version.

*54. L153-154*

*"from the foot to the peak of the mountain"*

*decrease with elevation. I guess it gets to 0 even before the peaks.*

Reply: Yes, this situation exists. However, for the sake of simulation efficiency, the soil thickness in the model was only divided based on the mountaintop, the mountainside, and the foot of the mountain.

*55. L154*

*"and0"*

Reply: Apologies for this mistake; the 0 here was mistyped and has been deleted.

*56. L279*

*"the upper boundary energy"*

*"energy fluxes"*

Reply: Thank you for your suggestion; we changed this phrase to "energy fluxes".

*57. L279*

*"calculated by meteorological elements"*

*"calculated based on the climate forcing"*

Reply: Thank you for your suggestion; we have made relevant changes.

*58. L289*

*"specific heat"*

*"specific heat capacity"*

Reply: We have modified the phrases according to your suggestions.

*59. L293*

*"the iterative method"*

*There are a few of them, be more specific.*

Reply: Thank you for the comment and suggestion. In the reply to questions 12 and 13, we simplified the introduction of the energy calculation and provided the calculation method for *G*.

*60. L389-390*

*"the heat preservation effect of the snow"*

*I guess the authors refer to the insulation effect of snow*

Reply: Yes, we revised the section to use professional terminology.

*61. L400*

*"the thermodynamic properties"*

*Are the author talking about the temperature ? I am confused.*

Reply: Thermodynamic properties here refer to heat capacity and thermal conductivity, which change the layer temperature by affecting heat transfer.

*62. L429-430*

*"the unstable water-holding capacity"*

*I don't understand why it is unstable.*

Reply: This sentence has been corrected.

Change from

"However, due to the large uncertainty of the compositions of the soil and gravel layer, the unstable water-holding capacity of the soil–gravel layer cannot be accurately

reflected when the model is generalized, which also leads to a certain difference between the WEP-QTP simulation and the measured values."

to

"However, as the actual condition of the underlying surface cannot be homogeneous like in the model generalization, there was a discrepancy in the model simulation results."

---

## Referee Report (RR1)

General comment

I am not convinced by the answers brought to my points. To summarize, I had and still have three main complaints (that I also detail below the general comment).

First, the revised abstract still says : "The Qinghai-Tibet Plateau (QTP) has a thin soil layer on top of a thick soil-gravel mixture (SGM) layer." And this still makes no sense to me and I am still waiting for references which prove that I am wrong and that a significant portion of the QTP has indeed this stratigraphy. As developed below, it seems more reasonable to me to say that it is a relevant simplification for the present watershed based on the field observations of the authors. The conclusion of the manuscript is more cautious in this regard and I like better how it presents the study content than the abstract.

Second, the model description is still very puzzling. The authors confused avalanches and snow melt but most importantly, the surface energy balance description disappeared to the benefit of a peculiar new equation 7 that only considers sensible heat fluxes and in a very strange way that is not supported by the provided references. How come the first version was so wrong about the atmosphere-surface energy fluxes ? As I said before, I suspect the model is fine and only the description has problems but the whole process of deleting the surface energy balance part to replace it by this odd equation leaves me with a weird feeling. Clarifications are needed.

Finally, when answering to me, the authors did not really address my concern about the fact that the demonstration of the improvements brought by the new model needs to be improved (because I did not develop it enough in my detailed comments I guess). But I saw that other reviewers were more thorough on this point. So I'll leave it to be fixed based on their input.

So in the end, I still think the study is interesting but the problems that bother me still need to be addressed. Also I realize now that the title of the study mentions a new model whereas it would be more accurate to mention new improvements brought to an existing model (which is different from creating a new model from scratch). Below are my comments to specific answers from the authors.

Comment 1.

I think that the answer to comment 1 is off. My point was to say that there is no reason such a variety of landscapes and surface processes leads to a uniform stratigraphy at the scale of a catchment and even less at the scale of the QTP. It is no problem to simplify reality if it is acknowledged and framed. Explaining the model class does not address this point.

Comment 2.

If so, can the author provide a proportion of the QTP area for which this stratigraphy applies ? I am not convinced by the references provided. We are discussing real world observations that can assess the validity of the proposed stratigraphy and the authors suggest two papers describing modelling works (Chen et al. 2015 and Yang et al. 2009). Among the 2 others, one I did not find (Sun et al. 1996), so I checked the other one (Deng et al. 2019). Maybe I missed it but I did not find anything about the ubiquity or widespread occurrence of a gravel layer below a thin soil layer over the whole QTP. The paper discusses Pliocene and Pleistocene deposits in the eastern QTP and their connection with tectonics. Figures 6 to 8 of this paper summarize the stratigraphy in different areas, figure 6 shows a lot of lateral variability as a consequence of the activity of a fault, figure 7 shows gravel on

top of sand (for the upper part of the stratigraphy), and figure 8 shows humus on top of clay with limestone fragment. This last one could fit the theory of the authors but nowhere Deng et al. claim that this is ubiquitous. The word fragments do not appear anywhere else in the article. And Deng et al. use the word gravel in its common meaning and not as a rock fragment. So to say, I am still waiting for the proof that this stratigraphy is widespread across the QTP. Again I have no problem with simplifications but then it needs to be presented as such. I would largely prefer to read that it is a relevant simplification for the present watershed based on the field observations of the author. Either this or, as I was saying earlier, then the author should provide the order of magnitude of the coverage of this stratigraphy, a reference that says if it is e.g. 0.8%, 8% or 80% of the plateau that correspond to this stratigraphy, based on relevant references so that we know what we are discussing.

Comment 8.

"In a saturated state, the macropores form a fast channel for transporting water. However, when the SGM layer is in an unsaturated state, the water mainly moves under the actions of the matrix potential and gravitational potential. Thus, in an unsaturated state, the macropores do not work, and the gravel will hinder the movement of water."

Conductivity is known to evolve with saturation but this explanation is a bit puzzling to me. How strong is the matrix potential in a soil with high gravel content ? And how come this matrix potential does not also ampere gravitational drainage ?

Comment 10.

Confusion between snow melt and avalanche is very surprising to me, but now I understand the corrected sentence. Can the author elaborate on the importance of accounting for avalanches for ground thermo-hydrological regime ? It is surprising to me that avalanches play a big role in this regard but I might be wrong.

By the way L258-259 of the revised manuscript still say:

"When the snow thickness difference between two calculation units exceeded this threshold, snow meltdown occurred. The snow in the higher-altitude calculation unit slides into the next unit until the two units have the same snow thickness."

And line L264-265 say:

"when the difference in snow thickness between two adjacent contour bands exceeds this threshold, an avalanche occurs between those contour bands. The snow in the higher-altitude contour band slides into the lower band until the two bands equalize in snow thickness."

This model description is still confusing and it should not be the case at this stage.

Comment 13.

This is extremely weird. In the initial version of the manuscript, the energy fluxes between the atmosphere and the surface was based on surface energy balance calculation with incoming and outgoing radiations, latent and sensible heat fluxes... And now all of this is replaced by this new equation 7 ! What happened to initial equations 7, 8 and 9 ? And how could the first model description be so wrong ? Such a difference implies a massive difference regarding the forcing data that are used. This now comes after the confusion between snow melt and avalanches and gives the impression that all these parts were written with very little knowledge of the model. It is the first time I see something like this and I do not know what to think of that. I still want to believe that only the model description is off.

Finally, the physics of the new equations look questionable to me. First, now it seems that the only energy exchange between the atmosphere and the surface corresponds to sensible heat fluxes so what about radiations ? What about evaporation ? Neither radiations nor evaporation are going to impact the soil thermal regime ? What about the claimed water-heat coupling if evaporation does not impact the energy fluxes between the surface and the atmosphere ? The consequences of such a choice need to be developed and discussed. Also the initial version of WEB COR included a surface energy balance calculation so if this is the new calculation is it a downgrade of WEB COR regarding physical processes and it should be mentioned.

Second, this flux does not depend on wind speed or not even on a bulk parameter such as a convection coefficient or the aerodynamic impedance that was present in the initial draft. This equation should describe a process happening at an interface and looks like something based on the energy variations of a volume of ground (C x V x dT, but in this case, dT would have a sense if it was a transient variation not an instantaneous potential). I am confused and considering what is happening here, I have a hard time believing this equation was used. Third, what value does the "du" parameter take ? Are we talking about several centimeters ? meters ? How is it established ? Because the energy change will vary linearly with this value.

Finally, I checked Jia et al. (2001) and the new equation 7 has nothing to do with Jia et al. (2001), even equation 61 from Jia et al. (2001) is very different. Jia et al. (2001) actually includes surface energy balance, as initially submitted here. I checked Hu and Islam (1995) (the new draft says Hu et al., 2001, I assume it is a typo) but new equation 7 is nowhere to be found either. I have the feeling new equation 7 is not physically valid regarding sensible fluxes for the aforementioned reasons (not counting that radiations and latent fluxes are for now on ignored) so I would need proof that I am wrong (i.e. a reference that established its validity).

Comment 34.

The answer is very nebulous and imprecise and following the answer on Comment 13, it shows a limited knowledge of the model. I am surprised that in a study aiming at bringing model developments, knowledge about the relationship between negative temperature (or energy) and liquid water content is so hard to find. Basically the appendix B14 told me to check Li et al. 2019, which I did. And Li et al. 2019 says "The water–heat continuous equation of frozen soil is solved numerically based on the soil freezing status and empirical formulas." I tried to dig and went from Li et al. 2019 to Wang et al. 2014 and then to Niu et al. 2006, and there I actually found relationships between liquid water content and negative temperatures that are in WEB COR if understood correctly.

Comment 35.

If so please explain where and how you use the riverbed conductivity. And please do so when the model setup is described.

Comment 39.

"… Figure 10 have no measured values. Figure 10 was provided to compare the effect of model improvement on the hydrological cycle flux." How can we know that it is an improvement if there is no field value to compare to ? Unless I missed something, the fact that it is different does not imply that it is better no ? I don't follow this reasoning.

I went through the new draft:

Line 197

"… its higher reflectivity to shortwave solar radiation were also considered"

When talking about snow. So here again I wonder: are the authors using surface energy balance calculation (including radiations) or not ? Because if it is just the new equation 7, radiations are not accounted for in the model…

Line 227

I think it would be nice to have the values of the empirical parameters.

Several lines:

Line 63-64: "the saturated hydraulic conductivity of SGM decreases as the gravel content increases"

Line 222: "since gravel can neither conduct nor store water"

Line 242: "The gravel increases the porosity in the SGM layers"

Line 347-348: "The saturated hydraulic conductivity of the soil layer was 0.648 m/d, that of the SGM layer was 4.32 m/d"

These assertions don't work together or if they do please explain.

Line 279-280

"For the heat transfer process, assuming that the upper boundary of the system is the atmosphere, which controls the input and output of the system energy."

This sentence has neither subject nor conjugated verb relating to the subject.

Line 300-301

Former equation 11 is now equation 9.

Line 373-377

Please explain how you got discontinuous but millimetric values of the snow cover for your observations. Explain also how the comparison was made. The contour bands are 20 km2 in average and we are talking here about a point-wise measurement.

Line 379

If I understood correctly there is just one experimental site, so no S at site.

Line 395

The legend of the figure was cropped (visible on the initial submission). The graphs are left without legend, and cannot be understood.

Line 418

As for figure 7, there is no longer a legend on Figure 8 and the reader cannot know what is observation, QTP and COR.

---

## Author Response (AR2)

**Reviewer 1**

*General comment*

*I am not convinced by the answers brought to my points. To summarize, I had and still have three main complaints (that I also detail below the general comment).*

*First, the revised abstract still says : "The Qinghai-Tibet Plateau (QTP) has a thin soil layer on top of a thick soil-gravel mixture (SGM) layer." And this still makes no sense to me and I am still waiting for references which prove that I am wrong and that a significant portion of the QTP has indeed this stratigraphy. As developed below, it seems more reasonable to me to say that it is a relevant simplification for the present watershed based on the field observations of the authors. The conclusion of the manuscript is more cautious in this regard and I like better how it presents the study content than the abstract.*

Dear Reviewer:

Firstly, thank you very much for your detailed and professional comments during the two reviews. In the abstract, we have made the following change to make our description more precise:

"The Qinghai-Tibet Plateau (QTP) has a thin soil layer on top of a thick soil-gravel mixture (SGM) layer."

To:

"Owing to plate movements and climatic effects, the surface soils of bare lands and grasslands on the Qinghai–Tibet Plateau (QTP) are thin, and the soil below the surface contains abundant gravel."

Secondly, we have also provided additional information about the distribution of this geological structure on the Tibet Plateau based on relevant literature, as detailed in the answers to Comments 1 and 2 below. I hope this will help you better understand the distribution of this structure on the QTP.

*Second, the model description is still very puzzling. The authors confused avalanches and snow melt but most importantly, the surface energy balance*

*description disappeared to the benefit of a peculiar new equation 7 that only considers sensible heat fluxes and in a very strange way that is not supported by the provided references. How come the first version was so wrong about the atmosphere-surface energy fluxes ? As I said before, I suspect the model is fine and only the description has problems but the whole process of deleting the surface energy balance part to replace it by this odd equation leaves me with a weird feeling. Clarifications are needed.*

Reply: We apologize for the confusion. The initial manuscript originally intended to present the specific calculation of each energy flux in the energy balance equation of the model, but it did not introduce the calculation method of $G$ clearly. In this study, the upper boundary of the object of study is the atmosphere, and the temperature difference between the atmosphere and the surface is the source of heat transfer. Therefore, the added equation 7 is used to calculate $G$. In the revised manuscript, we have supplemented the derivation procedure of the equation used to solve $G$ and the corresponding references. This is clarified in detail in the responses to Comment 5 below.

*Finally, when answering to me, the authors did not really address my concern about the fact that the demonstration of the improvements brought by the new model needs to be improved (because I did not develop it enough in my detailed comments I guess). But I saw that other reviewers were more thorough on this point. So I'll leave it to be fixed based on their input.*
*So in the end, I still think the study is interesting but the problems that bother me still need to be addressed. Also I realize now that the title of the study mentions a new model whereas it would be more accurate to mention new improvements brought to an existing model (which is different from creating a new model from scratch). Below are my comments to specific answers from the authors.*

Reply: Thanks to your suggestion, we have changed the title accordingly, as follows:

"Application of an improved distributed hydrological model based on soil–gravel structure in the Niyang River Basin, Qinghai–Tibet Plateau"

Thank you very much for your careful review of our manuscript. Your comments have helped us to greatly improve the quality of the manuscript. For the comments that were not clearly explained in the previous reply, we will provide additional detailed responses below.

**1. Comment 1.**

*I think that the answer to comment 1 is off. My point was to say that there is no reason such a variety of landscapes and surface processes leads to a uniform stratigraphy at the scale of a catchment and even less at the scale of the QTP. It is no problem to simplify reality if it is acknowledged and framed. Explaining the model class does not address this point.*

Reply: The QTP has a variety of landscapes and surface processes, but grassland occupies the largest proportion of land use, followed by bare land, and the sum of the two accounts for up to 81.64% (Table 1).

**Table 1. Proportion of land use in Qinghai-Tibet Plateau**

| Land use | grassland | bare land | forest | water body | cultivated land | built-up land |
|---|---|---|---|---|---|---|
| Proportion (%) | 48.60 | 33.04 | 12.05 | 5.18 | 1.02 | 0.11 |

In these areas, physical weathering is dominant due to long-term low temperatures, mineral decomposition is low, and clay content gradually decreases from top to bottom. In the central and eastern plateau meadow areas of the QTP, due to the slow decomposition of biomass in the soil, the topsoil (typically 0–20 cm) in this region accumulates much denser grassroots and more soil organic matter than does the deep soil. The soil stratification in these areas is very significant compared to that observed in other regions (Yang et al., 2009). In the reply to Comment 2, we provide specific literature evidence and a detailed reply. Accordingly, we have revised the abstract and

the introduction regarding the distribution of this soil stratification structure to make our discussion more rigorous.

In the abstract, change from

"The Qinghai-Tibet Plateau (QTP) has a thin soil layer on top of a thick soil-gravel mixture (SGM) layer."

To:

"Owing to plate movements and climatic effects, the surface soils of bare lands and grasslands on the Qinghai－Tibet Plateau (QTP) are thin, and the soil below the surface contains abundant gravel."

In the introduction, change from

"In addition, under strong freeze-thaw conditions in the cold plateau region, the humus accumulation of herbaceous plants is slow, while the decomposition of minerals is weak, resulting in slow soil development on the surface of Quaternary deposits and a thin soil layer above the SGM (Deng et al., 2019; Yang et al., 2009; Chen et al., 2015; Sun, 1996)."

To:

"In addition, in the cold alpine regions of the QTP, the decomposition of biomass occurs mostly in the surface layers of Quaternary sediments owing to the low temperatures, resulting in the formation of a thin soil layer that is more highly developed and accumulates more organic matter than deeper layers (Sun, 1996). This soil stratification is particularly evident in alpine meadows (Yang et al., 2009; Pan et al., 2017)."

*2. Comment 2.*

*If so, can the author provide a proportion of the QTP area for which this stratigraphy applies ? I am not convinced by the references provided. We are discussing real world observations that can assess the validity of the proposed stratigraphy and the authors suggest two papers describing modelling works (Chen et al. 2015 and Yang et al. 2009). Among the 2 others, one I did not find (Sun et al. 1996), so I checked the other one (Deng et al. 2019). Maybe I missed it but I did not*

*find anything about the ubiquity or widespread occurrence of a gravel layer below a thin soil layer over the whole QTP. The paper discusses Pliocene and Pleistocene deposits in the eastern QTP and their connection with tectonics. Figures 6 to 8 of this paper summarize the stratigraphy in different areas, figure 6 shows a lot of lateral variability as a consequence of the activity of a fault, figure 7 shows gravel on top of sand (for the upper part of the stratigraphy), and figure 8 shows humus on top of clay with limestone fragment. This last one could fit the theory of the authors but nowhere Deng et al. claim that this is ubiquitous. The word fragments do not appear anywhere else in the article. And Deng et al. use the word gravel in its common meaning and not as a rock fragment. So to say, I am still waiting for the proof that this stratigraphy is widespread across the QTP. Again I have no problem with simplifications but then it needs to be presented as such. I would largely prefer to read that it is a relevant simplification for the present watershed based on the field observations of the author. Either this or, as I was saying earlier, then the author should provide the order of magnitude of the coverage of this stratigraphy, a reference that says if it is e.g. 0.8%, 8% or 80% of the plateau that correspond to this stratigraphy, based on relevant references so that we know what we are discussing.*

Reply: In the manuscript, we provide four references (Deng et al., 2019; Yang et al., 2009; Chen et al., 2015; Sun, 1996), two of which you may have missed (Yang et al., 2009; Chen et al., 2015).

Reference 1: Yang K, Chen Y Y, Qin J. Some practical notes on the land surface modeling in the Tibetan Plateau[J]. Hydrology and Earth System Sciences, 2009, 13(5): 687-701.

In section 4.1 of this paper the authors write: "The decomposition of the biomass in the soil is slow due to low temperature over the Plateau, and therefore, the topsoil (~typically 0–20 cm) in the CE-TP region accumulates much denser grassroots and more soil organic matters (SOM) (not shown) than the deep soil does. This soil stratification in the CE-TP should be addressed for the following reasons. First, the soil stratification in the CE-TP is very significant compared to that observed in other

regions."

This paper also provides the soil texture and parameters obtained from laboratory experiments of soil samples taken at Anduo sites (Table 3).

**Table 3.** Soil composition and parameters analyzed by laboratory experiments for Anduo site for five field samples (two at 5 cm, two at 20 cm, and one at 60 cm) (courtesy of N. Hirose).

| Sample No. | Depth (cm) | Sample features | Composition (%) | | | | $\rho_d$ | $\theta_s$ |
| | | | Gravel | Sand | silt | clay | (kg m$^{-3}$) | (m$^3$m$^{-3}$) |
| --- | --- | --- | --- | --- | --- | --- | --- | --- |
| 5A | 5 | dense root | | N/A | | | 0.667 | 0.633 |
| 5B | 5 | dense root | 0.00 | 30.64 | 59.88 | 9.48 | 0.817 | 0.593 |
| 20A | 20 | little root, gravel | 3.69 | 69.02 | 19.83 | 7.46 | 1.378 | 0.440 |
| 20B | 20 | little root, gravel | 4.24 | 67.08 | 19.53 | 9.15 | 1.694 | 0.318 |
| 60 | 60 | little root, gravel | 3.35 | 76.56 | 10.12 | 9.97 | 1.426 | 0.370 |

From Table 3, it can be find that the proportion of gravel is 0% and the proportion of sand is 30.64% when the depth < 20 cm. When the depth ≥ 20 cm, gravel appears, and the proportion of sand increases to more than twice that of the surface soil. A clear soil stratification can be observed.

Reference 2: Chen H, Nan Z, Zhao L, et al. Noah modelling of the permafrost distribution and characteristics in the West Kunlun area, Qinghai‐Tibet Plateau, China[J]. Permafrost and Periglacial Processes, 2015, 26(2): 160-174.

This paper notes the influence of soil characteristics of the West Kunlun region, located on the QTP, on the simulation of permafrost distribution. Sensitive soil parameters of 0~1 m soil layers were provided by the laboratory soil particle size distribution analysis and the empirical model of soil parameters based on soil texture (Table 1).

Table 1 Soil layers and parameters at the TGL station.

| Category | Number of layers | Depth(m) | QTZ | BB | SATDK × 10$^{-6}$ m × s$^{-1}$ | Soil description |
| --- | --- | --- | --- | --- | --- | --- |
| Surface layer | 2 | 0–0.12 | 0.65 | 4.52 | 4.12 | Sandy loam |
| Subsurface layer 1 | 5 | 0.12–0.92 | 0.60 | 3.12 | 2.21 | Sand, loam, gravel |
| Subsurface layer 2 | 6 | 0.92–3.02 | [a]0.4–0.7 | [b]2–5 | [c]2–5 | Sand, gravel |
| Bottom layer | 10 | 3.02–15.82 | [a]0.4–0.7 | [b]2–5 | [c]2–5 | Sand, gravel, rock |

[a]Value range of QTZ; [b]value range of BB; [c]value range of SATDK; value ranges of QTZ, BB and SATDK are determined by the field soil profile, the laboratory soil particle distribution analysis or the original value settings in the Noah soil parameters table. See text for abbreviations.

As can be seen from this table, the soil is sandy loam at 0–0.12 m, but below 0.12 m, the soil becomes a mixed layer with sand and gravel.

Also in another paper:

Reference 3: Pan Y, Lyu S, Li S, Gao Y, Meng X, Ao Y, and Wang S: Simulating the role of gravel in freeze‒thaw process on the Qinghai‒Tibet Plateau, Theoretical and applied climatology, 127, 1011-1022, 2017.

This paper pays attention to the gravel in the soil of the QTP. Through the sampling results in Madoi (Table 1) and Nagqu (Table 2), it can be find that the content of gravel in the surface soil is relatively low ($\leq 10\%$), and the content of gravel under the surface soil is higher (around 30%). The soil also has an obvious stratified structure.

**Table 1** Soil texture and soil organic content at Madoi

| Layer | Depth $z$ (m) | Sand (%) | Clay (%) | Gravel (%) | Soil organic (kg m$^{-3}$) |
|---|---|---|---|---|---|
| 1 | 0.0175 | 30.20 | 30.0 | 5 | 85 |
| 2 | 0.0451 | 36.47 | 25.5 | 5.70 | 75.12 |
| 3 | 0.0906 | 51.09 | 14.03 | 18.14 | 40.14 |
| 4 | 0.1656 | 49.36 | 13.59 | 27.08 | 31.37 |
| 5 | 0.2891 | 59.05 | 9.42 | 26.37 | 18.14 |
| 6 | 0.4930 | 71.97 | 2.47 | 24.54 | 1.92 |
| 7 | 0.8289 | 64.06 | 1.82 | 33.07 | 1.18 |
| 8 | 1.3828 | 75.56 | 2.96 | 20.62 | 1.1 |
| 9 | 2.2961 | 72.99 | 3.2 | 22.73 | 0 |
| 10 | 3.4331 | 63.80 | 2.95 | 31.80 | 0 |

**Table 2** Soil texture and soil organic content at Nagqu

| Layer | Depth $z$ (m) | Sand (%) | Clay (%) | Gravel (%) | Soil organic (kg m$^{-3}$) |
|---|---|---|---|---|---|
| 1 | 0.0175 | 63.68 | 4.13 | 10 | 100.4 |
| 2 | 0.0451 | 63.68 | 4.13 | 10 | 100.4 |
| 3 | 0.0906 | 63.68 | 4.13 | 10 | 100.4 |
| 4 | 0.1656 | 63.68 | 4.13 | 10 | 70.53 |
| 5 | 0.2891 | 43.50 | 10.99 | 10 | 45.15 |
| 6 | 0.4930 | 71.94 | 3.58 | 19.03 | 24.91 |
| 7 | 0.8289 | 67.08 | 0.88 | 28.46 | 13.52 |
| 8 | 1.3828 | 64.75 | 1.87 | 28.46 | 3 |
| 9 | 2.2961 | 64.75 | 1.87 | 28.46 | 0 |
| 10 | 3.4331 | 64.75 | 1.87 | 28.46 | 0 |

We plotted the sampling sites of these studies and our study areas in the figure below so that you can better understand the distribution of the current stratified research on the QTP.

[Figure]

Reference 4: Sun, H., The formation and evolution of the Qinghai-Tibet Plateau. Shanghai Science and Technology Press, 1996, ISBN:9787532340231.

The reference you didn't find (Sun et al. 1996) is a book written by Sun Honglie, an academician of the Chinese Academy of Engineering, who was the leader of the first Qinghai–Tibet scientific expedition. This book introduced the formation mechanism of the QTP, the uplift process, and its impact on the natural environment and human activities.

On page 263, when introducing the basic natural features of the QTP, it states that, "In the unique soil-forming environment of the QTP, most soils are characterized by thin soil layer...":

上游方向推进,使侵蚀裂点以下地形的切割程度更加明显。正是由于隆起抬升的速度快,且幅度大,青藏高原边缘山地地貌外营力以侵蚀作用占绝对优势,堆积作用仅是局地或暂时的。陡峭的山地、深切的河谷、间断的古高原夷平面残留是边缘山地的主要地貌类型,反映出高原的地形发育处于初始发展的活跃阶段。由地表物质不稳定而导致的坡面滑塌、土壤水蚀及泥石流等则是这一区域普遍的地形发育现象。青藏高原东部与南部边缘深切割山地地形分布范围宽阔,而北部和西部边缘深切割山地地形范围相对较窄。这种地形发育的不对称除受地质因素影响外,山地所处地域的气候条件起了重要作用[2]。

高原的隆起,改变了这一地区原有的行星环流,形成了具有地区特征的高原天气系统。环流状况的变化,改变了地形发育外营力条件的地域格局,使得高原内、外流水系发生显著的变迁。现代青藏高原南部和东南部面向南来的暖湿气流,地势高度自南向北逐渐抬升,降水较为丰沛,河流切蚀能力强,水系朝溯源侵蚀的方向演进。如高原东部金沙江、澜沧江和怒江等均溯源侵蚀至接近于高原腹地,高原南部朋曲河切穿喜马拉雅山直延伸至山脉北麓。与上述情况相反,在青藏高原内部以及受巨大山脉屏障的雨影区,由于气候偏干,河流水量减少,部分外流水系转变为内流水系,出现时令河甚至于有的河流、湖泊退缩消失。

青藏高原现代土壤发育也仍处于新的成土过程中。由于高原迅速的抬升,使成土条件分阶段向高寒方向转化,土壤发育也在不断与新的环境相适应。在活跃的山地侵蚀与堆积作用下,地表物质迁移频繁,土壤发生层的物质组成相当不稳定,土壤发育常受到土层剥蚀或掩埋,成土过程多具间断性。受高寒作用的影响或由于湖泊、冰川退缩,地表物质风化过程缓慢或新出露地面风化度很浅,许多土壤才开始发育。在青藏高原独特的成土环境下,大部分土壤具有土层薄、粗骨性强,风化程度较低的特点。越是干旱、高寒和坡度陡峭的地域,土壤发育的这些特点越突出。在青藏高原干旱、半干旱地区,土壤砂砾化现象普遍,风砂的堆积和推移也往往造成原始土壤发育过程不连续。

依所处地理位置和温度、水分条件的差异,青藏高原土壤分为大陆性与季风性这两大系统。大陆性土壤系统分布于高原内部高寒山地及干旱、半干旱区,包括的土类有寒冻土、高山草甸土、山地灌丛草原土、高山草原

On page 303, when describing the soil characteristics of the alpine meadow with the largest proportion of the QTP, it also states that, "The long-term low temperature makes the physical weathering dominant, the degree of mineral decomposition is not

high. The content of clay is very low, and gradually decreases from top to bottom. In line with this, the soil layer of meadow is shallow, and coarse in texture.....":

物共同形成的毡状草皮层覆盖地表,发育良好。多年测定表明,高寒草甸鲜草产量为 1.0～3.0t/hm²,其中可食牧草占 60%～90%。嵩草草甸光合产物主要集中于地下器官,约为地上部分的 3 倍。通常地上部分叶片生长密集,牧草品质优良,适口性好,营养价值高,草地植物粗蛋白质含量平均为 18%。

**3. 土壤的主要特征**

高山草甸土(寒毡土)属 AC 型,最主要的特征是土壤表层有一为嵩草等死根和活根密集纠结而成的草皮层(Ac 层),厚约 10cm,其下是腐殖层(A₁),B 层发育不明显,C 层则明显地受基岩性质所制约。草皮层形成的原因,主要在于低温条件下,植物生理干旱的持续时间较长,微生物活动受到抑制,植物残体分解缓慢,因而嵩草等庞大根系缠结成层。在植物萌发及生长期间,土壤中温度水分条件较好,植物残体的分解得以进行,有腐殖质的积累,土壤结构在腐殖质层较为优良。但长期低温使物理风化作用占优势,矿物分解程度不高,粘粒含量甚低,向下逐渐减少,粘土矿物以水化云母为主。与此相适应,草甸土的土层浅薄,质地轻粗,剖面中矿物组成的差异较小。[2]

高山草甸土表层常有冻胀裂缝,沿裂缝土体常于向阳面翘起,而形成草皮层块。草皮层块于冻结或解冻期间,由于不同物质胀缩程度和导热速度的不同,造成草根与其下土层断开并形成滑面。因而,草皮层块常形成向下滑塌的现象,有些草皮层块甚至滑离土面,形成斑块状脱落。[2]

在相对温暖的暖季正逢雨季,由于植毡强大的蓄水能力,土体处于嫌气状态,不利于有机物的强烈分解。雨季结束后,土层含水量逐渐降低,通气条件有所好转,但土温亦随之下降,冬半年土壤长期冻结,同样有碍于有机物的矿化。寒毡土有机物质的存在状态和数量比例与同纬度平原地区土壤有很大的不同,主要表现为有机物的数量多和根系比例高。[34]

据实测资料,寒毡土土体中 A 层中各类有机质与整个土体中有机质总贮量的比值表明,有机物在剖面中分布不均匀,主要集中在上部。有机物的形态组成随深度而变化,土层中相对稳定的腐殖物质比重随深度增加而上升。

In addition, during our review we also found, in another book, a statement consistent

with the findings of our study.

Reference 5: Sun Z Y, Zhou A G, Bu J W, et al., Research on geological environmental carrying capacity evaluation method for mineral resources development in Qinghai-Tibet Plateau, China University of Geosciences Press, 2016, ISBN:9787562539940

On page 27, when introducing the soil of the QTP, the authors state that, "The soil of the Qinghai-Tibet Plateau, especially the alpine soil, mostly shows the characteristics of small thickness and simple layers. The forest soil at the edge of the plateau is relatively well developed, but its thickness is generally only 50 to 90cm, and it is relatively rare to see more than 100cm. As for alpine soils, the thickness is even only about 30cm."

**1. 土壤发育历史短**

由于高原近代的自然条件变得愈来愈严酷,土壤发育的速度减缓,现代土壤形成的历史也比较短暂。因此,青藏高原土壤,特别是高山土壤,大都表现出厚度不大、层次简单的特点。高原边缘的森林土壤相对来说发育较好,但其厚度一般也只有50～90cm,超过100cm的比较少见。至于高山土壤,厚度更是只有30cm左右。

由于形成时间短,土壤剖面的分化比较差。以高山草甸土为例,它的表层是大量草根交错盘结、相互交织而成的草皮层。这种草皮层的形成与年内气温低、生物作用比较微弱有关。草皮层直接与母质相连接,部分虽然有过渡层次,但其发育很原始。

另外,因土壤非常年轻,质地也比较粗疏,砾石含量很高。大体上砾石含量超过30%的土壤要占2/3,个别土壤所含石砾超过50%。砾石含量低于5%的基本无砾石土壤仅有1/10。除了砾石以外,土壤中大量含砂,一般含量达40%～50%。由于细土物质少,土壤养分含量比较低,这种土壤十分容易引起沙化。

**2. 土壤的垂直分带性**

青藏高原地域广大、地形复杂,导致高原气候明显的空间分异,并进一步引起植被和土壤类型的变化和区域差异。随着高原各地的地势起伏变化,土壤的垂直分布规律明显,形成类型多样的土壤立体分布形式。

一般地说,山体愈高、相对高差愈大,其垂直带也就愈完整。如青藏高原东缘的贡嘎山,其东坡海拔1300m以下的河谷至山巅,依次为黄红壤、山地黄棕壤、山地棕壤、山地暗棕壤、亚高山漂灰土、亚高山草甸土、高山草甸土、高山寒漠土。而昆仑山南麓山体虽高大,但相对高差较小,因此,在这里一般分布的是高山草甸土和高山荒漠土,向上只有高山寒漠土,垂直带较为简单。

山地坡向对土壤垂直带有明显的影响,处于不同湿润状况分界地区的山体,其坡向影响尤为突出。以屏障作用显著的中喜马拉雅山脉为例,南北两坡水分状况不同,南坡湿润、北坡属半干旱,除去相对高度不同而引起的土壤垂直带的繁简差别外,在同一海拔高度上,南坡是亚高山灌丛草甸土,北坡则是高山草原土。就小范围的阴阳坡而言,在祁连山山地就有明显的差异,如山地阳坡为栗钙土,阴坡则为灰褐土,而且灰褐土的分布下限也明显降低。各种各样的土壤垂直带,按照土壤形成和分布特点,可以归纳为两大类型,即大陆性垂直结构类型和海洋性垂直结构类型。

海洋性垂直结构类型主要分布在高原的东南和南部边缘。土壤垂直结构的特点是:森林土壤类型发达,分布界线很高,垂直结构中完全没有出现草原土壤。自下而上依次分布着红壤、山地黄壤、山地黄棕壤、山地漂灰土、山地酸性棕壤、亚高山灌丛草甸土与高山草甸土,直至寒漠土与永久冰雪。以高原东缘二郎山为例,海拔1700m以下为山地黄壤,海拔1700～2100m一带的谷坡为山地黄棕壤,2100～3700m为山地棕壤,3700～3900m为山地泥炭质暗棕壤,二郎山顶3900m为亚高山灌丛草甸土及高山草甸土。

大陆性垂直结构类型分布在高原内部,土壤垂直结构中高山草原及山地草原土壤分布广泛,森林土壤仅在边缘山地阴坡呈小片分布,高原腹地根本没有森林土壤存在。例如昆仑山中段北翼就是典型的大陆性垂直结构类型,它以山地棕漠土为主,垂直结构简单。

The limited experiments we have conducted on the QTP do not yet allow us to give a definite percentage. However, from the above literature, we conclude that this soil stratification structure is not unique to the Niyang River basin and, at least in the meadow area accounting for 48.60% of the QTP, this kind of soil stratification structure is very significant. In the bare land, which accounts for 33.04% of the QTP, the stratification of the soil was not as significant as that in the meadows due to the thinner surface soil, but there is also abundant gravel beneath the surface soil.

**3. Comment 8.**

*"In a saturated state, the macropores form a fast channel for transporting water. However, when the SGM layer is in an unsaturated state, the water mainly moves under the actions of the matrix potential and gravitational potential. Thus, in an unsaturated state, the macropores do not work, and the gravel will hinder the movement of water."*

*Conductivity is known to evolve with saturation but this explanation is a bit puzzling to me. How strong is the matrix potential in a soil with high gravel content ? And how come this matrix potential does not also ampere gravitational drainage ?*

Reply: According to Darcy's law, the amount of water transported in the soil is:

$$q = K(\theta_l)\nabla H$$

where $K(\theta_l)$ is the hydraulic conductivity (cm/s) of the soil when the liquid water content is $\theta_l$; $\nabla H$ is the gradient of water potential (containing gravitational potential and matrix potential); and $q$ is the amount of water transported in the soil (cm/s).

Hydraulic conductivity and water potential gradients together affect the amount of water transported in the soil.

In the unsaturated state, the matrix potential and gravity potential work together to drive water transport. The effect of gravel content on the soil matrix potential can be specified in Equation 1:

$$\frac{\theta_l - \theta_r}{\theta_s - \theta_r} = A_m h^{-\lambda}\left(1 - B_m \omega_{gravel}\right)$$

In our study, for every 10% increase in gravel percentage with constant value of $\frac{\theta_l - \theta_r}{\theta_s - \theta_r}$, the matric suction decreases by approximately 10%.

In the saturated state, the matrix potential is 0, and water movement is driven only by the gravitational potential.

**4. Comment 10.**

*Confusion between snow melt and avalanche is very surprising to me, but now I understand the corrected sentence. Can the author elaborate on the importance of*

*accounting for avalanches for ground thermo-hydrological regime ? It is surprising to me that avalanches play a big role in this regard but I might be wrong.*

*By the way L258-259 of the revised manuscript still say: "When the snow thickness difference between two calculation units exceeded this threshold, snow meltdown occurred. The snow in the higher-altitude calculation unit slides into the next unit until the two units have the same snow thickness."*

*And line L264-265 say:*

*"when the difference in snow thickness between two adjacent contour bands exceeds this threshold, an avalanche occurs between those contour bands. The snow in the higher-altitude contour band slides into the lower band until the two bands equalize in snow thickness."*

*This model description is still confusing and it should not be the case at this stage.*

Reply: As can be seen from figures A1 and A2 in Appendix A, in the study area, precipitation is greater and temperature is lower at higher elevations, where the snow accumulation rate far exceeds its melting rate. If there were no avalanches, precipitation would be stored more and more as snow or ice at the top of the mountain, which does not correspond to the actual situation. In the real world, avalanches allow a portion of the snow to first collapse to a lower, warmer elevation and then gradually melt.

Specifically, if avalanches are not taken into account, the effects are as follows: 1) simulated runoff will be reduced (snowmelt runoff is an important source of runoff in this region); 2) the insulation effect of snow on permafrost at lower elevations will be reduced and the area of seasonally frozen soil will increase; and 3) soil moisture and temperature at lower elevations will also be affected by reduced snow cover during freezing-thawing periods.

We rewrote the description of the avalanche and retained the following corrected sentence:

"On the QTP, variations in temperature and precipitation caused by altitude differences result in more snow accumulation and less melting at higher altitudes. Therefore, avalanches are common in this region. In this model, we established a

snow thickness threshold. When the difference in snow thickness between two adjacent calculation units (the contour bands) exceeds this threshold, an avalanche will occur. The snow in the higher-altitude calculation unit slides into the next unit until the two units have the same snow thickness."

5. *Comment 13.*

*This is extremely weird. In the initial version of the manuscript, the energy fluxes between the atmosphere and the surface was based on surface energy balance calculation with incoming and outgoing radiations, latent and sensible heat fluxes… And now all of this is replaced by this new equation 7 ! What happened to initial equations 7, 8 and 9 ? And how could the first model description be so wrong ? Such a difference implies a massive difference regarding the forcing data that are used. This now comes after the confusion between snow melt and avalanches and gives the impression that all these parts were written with very little knowledge of the model. It is the first time I see something like this and I do not know what to think of that. I still want to believe that only the model description is off.*

*Finally, the physics of the new equations look questionable to me. First, now it seems that the only energy exchange between the atmosphere and the surface corresponds to sensible heat fluxes so what about radiations ? What about evaporation ? Neither radiations nor evaporation are going to*

*impact the soil thermal regime ? What about the claimed water-heat coupling if evaporation does not impact the energy fluxes between the surface and the atmosphere ? The consequences of such a choice need to be developed and discussed. Also the initial version of WEB COR included a surface energy balance calculation so if this is the new calculation is it a downgrade of WEB COR regarding physical processes and it should be mentioned.*

*Second, this flux does not depend on wind speed or not even on a bulk parameter such as a convection coefficient or the aerodynamic impedance that was present in the initial draft. This equation should describe a process happening at an interface*

*and looks like something based on the*

*energy variations of a volume of ground (C x V x dT, but in this case, dT would have a sense if it was a transient variation not an instantaneous potential). I am confused and considering what is happening here, I have a hard time believing this equation was used. Third, what value does the "du" parameter take ? Are we talking about several centimeters ? meters ? How is it established ? Because the energy change will vary linearly with this value.*

*Finally, I checked Jia et al. (2001) and the new equation 7 has nothing to do with Jia et al. (2001), even equation 61 from Jia et al. (2001) is very different. Jia et al. (2001) actually includes surface energy balance, as initially submitted here. I checked Hu and Islam (1995) (the new draft says Hu et al., 2001, I assume it is a typo) but new equation 7 is nowhere to be found either. I have the feeling new equation 7 is not physically valid regarding sensible fluxes for the aforementioned reasons (not counting that radiations and latent fluxes are for now on ignored) so I would need proof that I am wrong (i.e. a reference that established its validity).*

Reply: In the original manuscript, we simply wanted to show that the heat conduction into soil ($G$) and the sensible heat flux ($H$) can be solved using the joint solution of equations. However, $H$ is much larger than $G$ and the calculation method of $H$ is relatively crude, which makes this solution method unstable. Therefore, a simplified method was used in this model to obtain $G$ by climate forcing first. In the revised manuscript, we found that this statement was easily misunderstood by the reader, so we deleted this part and reintroduced the method of calculating $G$.

In this study, the upper boundary of our study object is the atmosphere, which controls the input and output of energy in the system. The temperature difference between the atmosphere and the surface is the source of heat conduction. For periodic forcing, the heat flux into the soil could be parameterized by the sum of a temperature-derivative term and the difference between ground surface and deep soil temperature (Hu and Islam, 1995). By integrating the thermal diffusion equation, equation 11 from Hu and Islam (1995) can be obtained:

$$C_{Vu}\delta\frac{dT_u}{dt} = G(0, t) - G(\delta, t)$$

where $C_{Vu}$ is the volumetric heat capacity of the underlying surface (MJ/m³/°C); $T_u$ is the temperature of the underlying surface; $\delta$ is the thickness of the underlying surface; and $G(\delta, t)$ is the heat flux at the depth $\delta$ at the moment $t$.

When $\delta$ is equal to the damping depth of the diurnal temperature wave ($d_u$) ($d_u = (2k/\omega)^{1/2}$ (Hu and Islam, 1995), where $k$ is the thermal diffusivity of the underlying surface (m²/s), $\omega$ is the fundamental frequency), $G(d_u, t)$ can be neglected, and the daily average heat flux conduction into the underlying surface ($G$) can be obtained by discretizing equation 11 from Hu and Islam (1995) as follows:

$$G = C_{Vu}d_u\Delta T_u$$

where $\Delta T_u$ is the daily temperature variation of the underlying surface (°C), which is approximated by the difference in temperature between the atmosphere and the underlying surface.

This is not a downgrade of the original model. The same approach was used in the WEP-COR model. It can be found in Li et al. (2019):

"The force-restore method (FRD) (Hu & Islam 1995) is used to solve $G$ and the surface temperature of different land covers."

In the new revised manuscript, we have rewritten this section by adding the derivation procedure for the equations used to solve $G$ and the corresponding references (Lines 278-293).

**6. Comment 34.**

*The answer is very nebulous and imprecise and following the answer on Comment 13, it shows a limited knowledge of the model. I am surprised that in a study aiming at bringing model developments, knowledge about the relationship between negative temperature (or energy) and liquid water content is so hard to find. Basically the appendix B14 told me to check Li et al. 2019, which I did. And Li et al. 2019 says "The water–heat continuous equation of frozen soil is solved numerically based on the soil freezing status and empirical formulas." I tried to dig and went*

*from Li et al. 2019 to Wang et al. 2014 and then to Niu et al. 2006, and there I actually found relationships between liquid water content and negative temperatures that are in WEB COR if understood correctly.*

Reply: We apologize for the lack of clarity. We have added the exact reference in the Appendix B, Equation B14.

**7. Comment 35.**

*If so please explain where and how you use the riverbed conductivity. And please do so when the model setup is described.*

Reply: This parameter was used in the calculation of groundwater outflow. Groundwater outflow is calculated according to the hydraulic conductivity $k_b$ of riverbed material and the difference between river water stage $H_r$ and groundwater level $h_u$ (Jia et al., 2001):

$$RG = \begin{cases} k_b A_b (h_u - H_r)/d_b & h_u \geq H_r \\ -k_b A_b [1 + (H_r - Z_b)/d_b] & h_u < H_r \end{cases}$$

where $A_b$ is the seepage area of the riverbed, $Z_b$ the elevation of the riverbed, and $d_b$ is the thickness of the riverbed material.

We have made the following supplements in the model structure section of Appendix B:

"The groundwater outflow was calculated according to the hydraulic conductivity of the riverbed material and difference between the river water stage and groundwater level (Jia et al., 2001)."

**8. Comment 39.**

*"… Figure 10 have no measured values. Figure 10 was provided to compare the effect of model improvement on the hydrological cycle flux." How can we know that it is an improvement if there is no field value to compare to ? Unless I missed something, the fact that it is different does not imply that it is better no ? I don't follow this reasoning.*

Reply: We did not intend here to say which water cycle flux change process is better.

The model performance has been discussed in Section 3.1 through the comparison of flow processes. In Section 3.3, the comparison of the changes in water cycle fluxes between the two models was only intended to explore the influence of gravel on the water cycle process.

Therefore, in order to avoid ambiguity, we change the title of Section 3.3 from:

"Simulation and comparison of watershed flow process"

To:

"Analysis of the snow–soil–gravel layer continuum effects on the process of water cycling"

Additionally, the following sentences were added before the comparative analysis:

"By comparing the hydrological cycle fluxes simulated by the two models, the influence of gravel on hydrological processes and the contribution of gravel to enhancing the simulation can be revealed, to some extent. Figure 10 shows the comparison and analysis of hydrological cycle flux changes across the basin simulated using the WEP–QTP and WEP–COR models."

***9.  I went through the new draft:***

***Line 197***

***"… its higher reflectivity to shortwave solar radiation were also considered"***

***When talking about snow. So here again I wonder: are the authors using surface energy balance calculation (including radiations) or not ? Because if it is just the new equation 7, radiations are not accounted for in the model…***

Reply: The surface energy balance equation was not used in the calculation of $G$, and we deleted this sentence accordingly.

***10.  Line 227***

***I think it would be nice to have the values of the empirical parameters.***

Reply: Flowing the introduction of the equation parameters, we supplemented the parameter taking values accordingly, as follows:

"Wang et al. (2013) provided a full description of the factors and parameters used in

Equation (1) ($A_m = 1.45$, $B_m = 0.2$, $\lambda = 0.18$) in this study."

**11. Several lines:**

**Line 63-64: "the saturated hydraulic conductivity of SGM decreases as the gravel content increases"**

**Line 222: "since gravel can neither conduct nor store water"**

**Line 242: "The gravel increases the porosity in the SGM layers"**

**Line 347-348: "The saturated hydraulic conductivity of the soil layer was 0.648 m/d, that of the SGM layer was 4.32 m/d"**

**These assertions don't work together or if they do please explain.**

Reply: Lines 63-64 refer to the case of low gravel content, but the soil of the QTP generally has a high gravel content. The abundance of gravel allows the formation of many interconnected pore channels within these sediments, thus increasing their saturated hydraulic conductivity. Therefore, the result of parameter calibration was that the saturation water conductivity of the g-layer (the lower gravel and soil mixed layer) is greater than that of the s-layer (the upper soil layer).

In order to avoid ambiguity, we have rewritten this part in the introduction as follows:

"As a result of the collision of the Indian and Eurasian plates, there are many gravel and rock fragments within QTP Quaternary sediments ( Chen et al., 2015; Deng et al., 2019). The abundance of gravel allows the formation of many interconnected pore channels within these sediments, thus increasing their saturated hydraulic conductivity (Beibei et al., 2009)."

We also removed sentences that were likely to cause misunderstanding (Lines 63-64 and Line 222).

**12. Line 279-280**

**"For the heat transfer process, assuming that the upper boundary of the system is the atmosphere, which controls the input and output of the system energy."**

**This sentence has neither subject nor conjugated verb relating to the subject.**

Reply: Thank you for pointing out this mistake, we have made the following

corrections to this sentence:

"For the heat transfer process, we assumed that the upper boundary of the heat transfer system is the atmosphere, which controls the input and output of the system energy."

**13. Line 300-301**

*Former equation 11 is now equation 9.*

Reply: We have corrected this error and checked all equation references.

**14. Line 373-377**

*Please explain how you got discontinuous but millimetric values of the snow cover for your observations. Explain also how the comparison was made. The contour bands are 20 km2 in average and we are talking here about a point-wise measurement.*

Reply: These snow thicknesses were manually measured in the field during each inspection of the experimental site, so they were discontinuous. The snow thickness of the experimental site was calculated according to the precipitation and temperature of the site through equations 4, 5, and 10, and then compared with the measured values. After validation of the actual measurements, these equations were then applied to the model to calculate the average thickness of snow in the contour band.

In Section 2.1.1, we added the method of snow thickness measurement:

"The monitoring instruments were inspected regularly during the experiment, along with manual measurement of snow thickness."

**15. Line 379**

*If I understood correctly there is just one experimental site, so no S at site.*

Reply: Thank you for pointing this out. We have corrected this in the revised manuscript.

**16. Line 395**

***The legend of the figure was cropped (visible on the initial submission). The graphs are left without legend, and cannot be understo*od.**

Reply: Thank you for pointing this out. We have replaced it with new figure in the revised manuscript.

*17 Line 418*

*As for figure 7, there is no longer a legend on Figure 8 and the reader cannot know what is observation, QTP and COR.*

Reply: Thank you for pointing this out. We have replaced it with new figure in the revised manuscript.

**References**

[1] Hu, Z., Islam, S.: Prediction of ground surface temperature and soil moisture content by the force‐restore method. Water Resources Research, 31(10): 2531-2539, 1995.

[2] Li, Z., Yu, G., Xu, M., Hu, X., Yang, H., and Hu, S.: Progress in studies on river morphodynamics in Qinghai-Tibet Plateau. Advances in Water Science, 27(4): 617-628, 2016

[3] Yang K, Chen Y Y, Qin J. Some practical notes on the land surface modeling in the Tibetan Plateau[J]. Hydrology and Earth System Sciences, 2009, 13(5): 687-701.

**Reviewer 2**

*This study developed a hydrological model (WEP-QTP) to consider soil－gravel structure (SGS) for areas in Qinghai-Tibet Plateau (QTP). The model employed different infiltration approaches to represent the impact of the dualistic SGS under fully thawed conditions, and it coupled heat and water transfer process for the snow soil－gravel layer continuum system. The model was well evaluated in a watershed in QTP regarding streamflow, snow thickness, soil-gravel temperature, and moisture. This model may be applicable in QTP to represent heat and water fluxes. However, large uncertainties exist from the forcing data, the model parameters. It is not a difficult task to obtain acceptable performance if the model was evaluated in a few sites of observations. The paper is readable, but language editing is strongly recommended to make the paper concise and professional.*

Dear Reviewer:

Due to the limitation of the field experiment environment on the QTP, we only used one site for model validation of the water-heat transport in soil. However, the flow process was validated using data from three hydrological stations. To make up for this deficiency, we have added the values of relevant parameters and reference notes in the revised manuscript. We thank you for your suggestion, which we will take into consideration in our subsequent study by selecting more typical experimental sites to optimize our model.

Our detailed responses to your comments are provided below.

*1. The primary motivation of developing WEP-QTP is the dualistic SGS in QTP. The authors provided site-specific example of the structure (Fig. 2). But it is unclear about that SGS is extensively distributed in QTP. So the authors should discuss the extensive distribution of SGS.*

Reply: The dualistic soil–gravel structure of the QTP was formed under the combined effect of long-term low temperature and plate collision, so this structure is commonly found in cold alpine regions. In these areas, physical weathering is dominant due to

long-term low temperature, mineral decomposition is low, and clay content gradually decreases from top to bottom. In the plateau meadow areas of the QTP, due to the slow decomposition of biomass in the soil, the topsoil (typically 0–20 cm) in this region accumulates much denser grassroots and more soil organic matter than does the deep soil (Yang et al., 2009).

Similar soil stratification structures were also observed by Yang et al. (2009) in Anduo, Chen et al. (2015) in TGL, and Pan et al. (2017) in Madoi and Nagqu. We plotted the sampling sites of these studies and our study areas in the figure below so that you can better understand the distribution of the current stratified research on the QTP.

[Figure]

We also revised the introduction regarding the distribution of this soil stratification structure accordingly to make our discussion more rigorous.

The following change was made:

"In addition, under strong freeze-thaw conditions in the cold plateau region, the humus accumulation of herbaceous plants is slow, while the decomposition of minerals is weak, resulting in slow soil development on the surface of Quaternary deposits and a thin soil layer above the SGM (Deng et al., 2019; Yang et al., 2009; Chen et al., 2015; Sun, 1996)."

To:

"In addition, in the cold alpine regions of the QTP, the decomposition of biomass occurs mostly in the surface layers of Quaternary sediments owing to the low temperatures, resulting in the formation of a thin soil layer that is more highly developed and accumulates more organic matter than deeper layers (Sun, 1996). This soil stratification is particularly evident in alpine meadows (Yang et al., 2009; Pan et

al., 2017)."

***2. If it is true the SGS is quite extensive in QTP, but the authors did not show the thicknesses of the top-layer soil and the underlying gravel. The thickness of the two structure is important to determine the related model parameters. Certainly, model calibration will improve the model performance, but can the calibrated parameters represent the physical structure of the SGS?***

Reply: For the thickness of the upper soil layer, we found through field sampling that they gradually decreases from the foot to the peak of the mountain in the study area (Lines 149-151). Higher elevations on the mountainside are generally alpine meadows with a soil layer thickness of about 40 cm. The bare lands further up the mountaintop are difficult to reach, but we speculate that they may be thinner. Therefore, in the model parameter setting, we set the soil thickness to 0.2 m, 0.4 m, and 1.0 m in the upper, middle, and lower contour zones, respectively (Lines 352-353).

For the depth of the g-layer (the lower gravel and soil mixed layer), due to the difficulty of actual measurement, we set it in the model as a multiple of the soil layer thickness, which is determined by the model parameter calibration. Its thickness is the total thickness of the aquifer and vadose zone above the impermeable boundary minus the surface soil thickness.

The calibrated model performs well in both the flow simulation and the water−heat transfer simulation. Especially for the simulation of the vertical variation of the moisture content (Fig. 3), from which we can observe that the parameter generalization of the model structure adequately reflects the actual situation (simulated soil moisture content varies discontinuously between above and below 40 cm).

Accordingly, we provided the following supplementary explanations in Section 2.2 of the manuscript:

"Under fully thawed conditions, the calculation object of water movement was defined as the dualistic soil–gravel structure (Fig. 3a). The upper layers were soil (s-layers), whose thickness was determined by the location of the calculation unit, and

which gradually decreased from the foot to the peak of the mountain (Fig. 3b). The lower layers were a mixture of gravel and soil (g-layers), dominated by gravel, the thickness of which was the total thickness of the aquifer and vadose zone above the impermeable boundary minus the surface soil thickness."

***3. The so-called improved model also introduced many new parameters as shown in Eqs. (1-3), for example, Am, Bm, and h. How were these parameters estimated in the application? What are the ranges of these parameters? Moreover, what about the sensitivity of the model performance to these parameters? The information regarding parameter sensitivity will be very important if the model is used in other watersheds. It would be better to provide a list of these new introduced parameters and their ranges.***

Reply: For Equation 1, $A_m$, $B_m$, and $\lambda$ are the empirical parameters to be estimated, and we determined the values of these parameters based on the recommended values combined with the simulation effects according to Wang et al. (2013) ($A_m$=1.45, $B_m$=0.2, and $\lambda$=0.18). Wang et al. (2013) obtained a range of values for these equation parameters by fitting water retention curves to soil mixtures with different gravel sizes and gravel contents and provided a complete description of the factors and parameters used in the equation.

We added the following supplementary notes to this equation in the revised manuscript:

"Wang et al. (2013) provided a full description of the factors and parameters used in Equation (1), $A_m$ =1.45, $B_m$ =0.2, $\lambda$=0.18 in this study."

For Equation 2, $A_{itf}$ and $B_{itf}$ are not parameters to be estimated, but intermediate parameters calculated from water content, capillary suction pressure, and hydraulic conductivity. Their specific calculation methods are shown in Appendix B by Equations B5 and B6.

For clarity, we added the following supplementary notes to the equation in the revised manuscript:

"...$A_{itf}$ is the total water capacity of the s-layers above the interface (mm); and $B_{itf}$ is

the error caused by the different soil moisture content of the s-layers above the interface (mm). A full description of the two parameters $A_{itf}$ and $B_{itf}$ has been provided by Jia and Tamai (1998), and their calculation is shown in Appendix B by Equations B5 and B6."

**4. WEP-QTP coupled a water-heat transfer process. Actually, a few other hydrological models (e.g., VIC) have considered this process (including the effect of freeze－thaw soil) so they are applicable to simulate hydrological processes in cold regions. What is the advantage of WEP-QTP?**

Reply: These models, like the WEP-COR model, define the simulated object of the water-heat transport process as a homogeneous medium, which is applicable to the simulation of the water-heat transport process in typical cold regions. However, the underlying surface of the QTP is different from that of typical cold regions, and the soil stratification structure affects the water-heat transport process. When these models are directly applied to this region, the soil moisture content of the topsoil and the surface air temperature gradient are significantly underestimated (Yang et al., 2009).

Our study also shows that the soil-gravel layer structure in the QTP affects flow processes. Neglecting the presence of gravels will underestimate the groundwater regulation, which would be detrimental to the accurate estimation of surface and subsurface water resources in this region under future climatic conditions.

The improved WEP-QTP model takes into full account the influence of the soil-gravel layer structure on the water-heat transport process and water cycle process, and its performance is significantly improved compared with the hydrological model (WEP-COR) applicable to typical cold regions.

**5. Lines 160-161, why were both LAI and NDVI used to calculate ET? The two are changeable in many cases.**

Reply: In this model, LAI was used in the Penman-Monteith formula to calculate transpiration. NDVI was used to calculate the fractional vegetation cover (*Veg*), which

was used as a coefficient to calculate transpiration from the dry part of vegetation leaves (Jia et al., 2001):

$$E_{tr} = Veg(1 - \delta)E_{PM}$$

where $E_{tr}$ is the transpiration from the dry part of vegetation leaves; $\delta$ is the fraction coefficient of the foliage covered by a water film; and $E_{PM}$ is the Penman–Monteith transpiration.

**6. Lines 190-193: was the hydrological process in farmland improved in the study? Does this study area contain farmland?**

Reply: Yes, farmlands are generally located at the foot of mountains where there is a thicker soil layer, and these areas were also improved as a calculation unit with soil thickness of 100 cm. However, the share of farmland is relatively small, accounting for only 1% of our study area.

**7. What is the difference between subsections 3.1 and 3.2? Both of the subsections described the comparison of streamflow.**

Reply: I think you may be confused by the difference between sections 3.1 and 3.3 (and not section 3.2, which presents a comparative analysis of soil temperature and moisture). Section 3.1 only introduces the parameters of model calibration and the simulation results. Section 3.3 is designed to explore the effect of the snow-soil-gravel layer continuum on the water cycle processes by comparing the simulated differences between the hydrological cycle flux before and after the model improvement.

To better organize the paper, we have changed the title of Section 3.3 from:

"Simulation and comparison of watershed flow process"

To:

"Analysis of the snow–soil–gravel layer continuum effects on the process of water cycling".

Also, we have added the following exposition at the beginning of this section:

"To explore the influence of the snow–soil–gravel layer continuum on the process of water cycling and the reasons behind the improvement of the model simulation, 2014

(a year for which all measured data were available) was selected as a typical year to compare and analyze the simulation results before and after model improvement (Fig. 9)."

**8. *Please provide legends for the lines in Figs. 7, 8.***

Reply: Thank you for pointing out these errors. We have replaced these figures with new figures in the revised manuscript.

**9. *Please give language editing to make the paper concise professional.***

Reply: Thank you for your suggestion. The language of the revised manuscript has been improved by a native English speaker/professional science editing service.

**References**

[1] Chen H, Nan Z, Zhao L, et al. Noah modelling of the permafrost distribution and characteristics in the West Kunlun area, Qinghai‑Tibet Plateau, China[J]. Permafrost and Periglacial Processes, 2015, 26(2): 160-174.

[2] Jia Y, Ni G, Kawahara Y, et al. Development of WEP model and its application to an urban watershed[J]. Hydrological Processes, 2001, 15(11): 2175-2194.

[3] Pan Y, Lyu S, Li S, et al. Simulating the role of gravel in freeze‑thaw process on the Qinghai‑Tibet Plateau[J], Theoretical and applied climatology, 2017, 127, 1011-1022

[4] Yang K, Chen Y Y, Qin J. Some practical notes on the land surface modeling in the Tibetan Plateau[J]. Hydrology and Earth System Sciences, 2009, 13(5): 687-701.

---

## Author Response (AR3)

**Reviewer 1:**

*General comment*

***This third round of review is much more satisfying than the previous one regarding my two main concerns (e.g. the stratigraphy and the model description). I think that now the authors have convincing arguments to say that their stratigraphy deserves to be considered over the QTP even though they should still avoid to say that the whole plateau presents it since, as we talked about it before, it does not make much sense (e.g. where you have bedrock, fluvial deposits, moraine…). It does not need to be 100% of the plateau to be scientifically significant. I saw that so far the authors have implemented a lot of my suggestions in the appendix rather than in the main text but I think putting several of their last arguments to support the relevance of this stratigraphy in the main text will vastly improve the paper.***

Dear Reviewer:

We thank you for your thoughtful suggestions and insights, which have enriched the manuscript and produced a better and more balanced account of the research. From the comments presented from prior discussion, it has come to our attention that the information presented on the geological stratification based on the QTP was not careful enough. Therefore, we have revised the introduction by adding arguments from the appendix to support the relevance of this stratigraphy.

Change from

"In addition, in the cold alpine regions of the QTP, the decomposition of biomass occurs mostly in the surface layers of Quaternary sediments owing to the low temperatures, resulting in the formation of a thin soil layer that is more highly developed and accumulates more organic matter than deeper layers (Sun, 1996). This soil stratification is particularly evident in alpine meadows (Yang et al., 2009; Pan et al., 2017)."

To:

"In addition, due to environmental constraints, physical weathering dominates the soil formation process on the QTP, resulting in a low level of soil mineral decomposition and slow soil development. Although the QTP has a variety of landscapes and surface processes, the grasslands occupy the largest proportion of the land, followed by bare land, the sum of which exceeds 80% (Zhang and Zhou, 2021). In these areas, the decomposition of biomass occurs mostly in the surface layers of Quaternary sediments because of the low temperatures, resulting in the formation of a thin soil layer that is highly developed and accumulates more organic matter than deeper layers (Sun, 1996). This soil stratification is widely spread on the QTP (Yang et al., 2009; Pan et al., 2017). The topsoil (typically 0 – 20 cm) in these areas is generally a sandy loam with a mixture of sand and gravel below it (Chen et al., 2015)." (Lines 64–71).

*Regarding the model description, I am still not fully sure but I think we went through an imbroglio because in the first place, the model description included the equation for surface energy balance calculation (as well as reference to the effect of radiations on snow) even though the model does not perform SEB. I think it is terribly misleading and a bad practice in general. The model description (and the equations inside) should describe what is inside the model. So this needs to be fixed (more precisions below) and it is important for a cryo-hydrology paper to discuss the fact that latent heat fluxes between the atmosphere and the surface are ignored in the model. This should also be part of the main text, not in the appendix.*

Reply: We apologize for the confusion. Considering that the upper boundary of the study object is the atmosphere, we did not use SEB in the calculation of the heat flux conducted into ground ($G$), but used the forcing recovery method directly through meteorological forcing. The SEB was included in the model, but instead of calculating $G$, it was used to calculate sensible heat and then give the breakdown of energy. This is indeed very misleading to the reader. Therefore, we removed the

sentences related to SEB that may mislead readers and introduced the calculation method of *G* as per your suggestion. Additionally, the implications of this current simplified calculation method for model simulations has been stated in the results and discussion section. Please refer to Comment 4 for more details.

*I have a few smaller points apart from these 2 and some phrasing suggestions.*

*Comment 1*

*Answer to points 1 and 2*

*The author have addressed my concerns about how the stratigraphy of the QTP is handled in a much more complete and convincing way than previously. I am now satisfied by the provided references and I thank the author for this detailed explanation. I think part of these explanations should be included in the main text to explain the motivation for these developments, because to me (and likely to other readers), until today, this stratigraphic choices did not sound motivated enough. I like Table 1 of the answer, I think it really helps to understand the point of this study. Yet, I miss the source of the data. Where does it come from ?*

Reply: We are pleased to hear that our explanation has addressed your concerns. Thank you for your suggestion, we have added the explanations in the Appendix to the Introduction accordingly. The revised information has been presented under the general comment.

Furthermore, we would like to apologize to the reviewer for the missing citation for the source data here, the data in Table 1 was calculated statistically from the Multi-Period Land Use Land Cover Remote Sensing Monitoring Dataset in China (CNLUCC) (Xu et al., 2018).

[1] Xu X L, Liu J Y, Zhang S W, et al. Multi-Period Land Use Land Cover Remote Sensing Monitoring Dataset in China (CNLUCC)[J]. Resource and Environmental Science Data Registration and Publication System. Available online: http://www. resdc. cn/DOI, 2018.

*Comment 2*

*Answer to point 3*

*I understand better what the author means. But I still think saying that in unsaturated conditions "the macropores do not work" is a bit strange and maybe not very appropriate. If this is the conveyed idea, I would recommend something more standard like "In unsaturated conditions, the hydraulic conductivity is lower and this decreases the drainage ability of the soil". Same for "and the gravel will hinder the movement of water" if a higher gravel content implies a lower matrix suction (and higher hydraulic conductivity, cf discussions on point 11 below), isn't it rather the opposite, that gravels promote the drainage ability of the soil?*

Reply: The two sentences are in response to comment 8 of the first round and we apologize that our initial colloquial explanation has brought about some confusion based on the concepts of hydraulic conductivity, drainage capacity, and water transport fluxes in the soil.

Hydraulic conductivity ($K$) is the water flux per unit water potential gradient in soil, the higher the soil hydraulic conductivity, the better the soil drainage capacity. In both saturated and unsaturated conditions, the gravel on the QTP are conducive to hydraulic conductivity (drainage capacity).

The soil water transport flux $q$ is the actual water flux between soil layers (not the soil drainage capacity) and is influenced by both the water potential gradient ($\nabla H$) and the $K$ (obtained by Darcy's law: $q = K(\theta_l)\nabla H$). Under unsaturated conditions, the gravel reduces the water potential gradient by reducing the matrix potential thereby affecting the water transport flux.

In short, gravel is a promoter of $K$ and a reducer of $\nabla H$. Whether it increases or decreases the soil water transport flux, it is dependent on the actual water content.

*Comment 3*

*Answer to point 4*

*Thanks for the explanation. The authors also deleted "When the snow thickness difference between two calculation units exceeded this threshold, snow meltdown occurred" and that was an important thing to do as well.*

Reply: We thank the reviewers for the acceptance of our explanation. This error was removed when the sentence was revised.

*Comment 4*

*Answer to point 5*

*I am amazed that every time I point a problem in the model description a new equation appears. But I think I see what the problem has been since the beginning now. The thing is WEB-COR does not do surface energy balance calculation neither in Li et al. (2019) nor in the present draft. In their paper, Li et al. (2019) presents a surface energy balance equation (Eq. 1) that they do not use in their model. And the author of the present draft did the same in their initial submission… Why would you put in your model description an equation that your model ignores? This is extremely misleading, because it is normal to expect that your model includes SEB if you put the SEB equation in the model description. But I think the blame goes to Li et al. (2019) as I can tell the authors of the present draft largely got inspired from Li et al. (2019) to write their model description.*

*So I think it is time to make this point clear and to add in the model description (not in the appendix) that, consistently with Li et al. (2019):*

*- the model does not perform surface energy balance*

*- the climatic forcing affecting the surface and driving heat conduction in the ground only result from temperature difference between the surface and the atmosphere*

*- radiation are thus not explicitly taken into account*

*- the evaporation values that the model computes are purely hydrological and have no impact on the surface thermal regime, which is an important lack regarding the given objective to "develop a modeling framework representing coupled water and heat transfer in the ground" (cf introduction).*
*The discussion should also include a section discussing the consequences of not accounting for the thermal effect of evaporation on their results.*

Reply: We apologise for any misleading information. After several rounds of reviewing and revision, all misleading sentences relating to energy balance and radiation have been removed from the manuscript. To make this part clearer, we have made a supplementary modification to the introduction of the $G$ calculation as per the reviewer's suggestion by explicitly stating that the temperature difference between the surface and the atmosphere is the only source of heat conduction.

Change from

"For the heat transfer process, we assumed that the upper boundary of the heat transfer system is the atmosphere, which controls the input and output of energy in the system. The temperature difference between the atmosphere and the surface is the source of heat conduction."

To:

"Due to the zonal variability in altitude, land surface features, and vegetation characteristics, spatial differences exist in meteorological elements and aerodynamic parameters on the QTP. The use of the energy balance method, which involves multiple meteorological elements and aerodynamic parameters to calculate the heat flux of each contour band, can lead to extensive calculations and an unstable solution process. Therefore, we assumed that the upper boundary of the heat transfer system is the atmosphere, which controls the input and output of energy in the system. The temperature difference between the atmosphere and the surface is the only source of heat conduction" (Lines 278–283).

In addition, since the objective of the study is "to develop a modeling framework representing coupled water and heat transfer in the ground based on the

snow-soil-gravel layer structure", the focus is on the water and heat transfer in the snow-soil-gravel layer. Therefore, the calculation of the energy input to the upper boundary is simplified and the energy balance formula was not used. As per your suggestion, we added the following discussion on this defect in Section 3.2 accordingly:

"It should be pointed out that, the model improvement is mainly concerned with the water and heat transfer within the seasonal thaw layer. Coupled with the fact that the amount of evaporation during the freeze–thaw period is generally less and the latent heat of evaporation accounts for a small proportion of the net radiation. Therefore, the model simplified the calculation of the energy input to the upper boundary of the seasonal thaw layer by using the temperature difference between the atmosphere and the surface as the only source of heat conduction, without quantitatively considering the influence of sensible heat and latent heat of evaporation on the heat flux conducted into the ground. The model needs to be further improved in subsequent studies by systematically considering the influence of the radiation and climate characteristics of the QTP on each energy component." (Lines 440–447).

**Comment 5**

**Answer to point 9**

**I appreciate the deletion but I still don't manage to understand why this was written in the first place, it was again very misleading, and gave the impression that the model included surface energy balance calculation even though it is not the case.**

Reply: As per the response under the general comment, the surface energy balance was included in the model, but instead of calculating $G$, it was used to calculate sensible heat and then give the breakdown of energy. The manuscript has been revised and all misleading information have been removed.

*Comment 6*

*Answer to point 11*

*I see the first 2 quotes of my initial remarks are now out of the manuscript, removing the ambiguity.*

Reply: We thank you for your review and suggestions, as it has helped us to improve the manuscript.

*Comment 7*

*I went through the draft again (since it is the 3rd time, I ignored the part for which we did not discuss anything recently)*

*L40 (and 78): I would replace "unique" by "characteristic" or "typical"*

*L75: I would replace "hidden" by "compensated"*

*L82: "(when the temperature of the calculation…)"*

*L188 "a key link" between what and what ? maybe a key feature*

Reply: Thank you for your suggestions, these have been amended accordingly.

*Comment 8*

*L199: the Wiktionary says that a continuum is: "A continuous series or whole, no part of which is noticeably different from its adjacent parts, although the ends or extremes of it are very different from each other." and I am not sure it applies well to the superposition of 3 different media separated by interfaces.*

Reply: We modified the "snow–soil–gravel layer continuum" to "snow–soil–gravel layer structure" by reason of rigor, this change also allows for uniformity with "dualistic soil–gravel structure".

*Comment 9*

*L252: "only moisture simulation was performed during this period" what is the ground temperature profile when the freeze thaw period restarts if it was not calculated during fully thawed period?*

Reply: The initial ground temperature profile for a new freeze-thaw period set in the current model was inherited from the ground temperature profile at the end of the previous freeze-thaw period.

*Comment 10*

*L257: "Therefore, avalanches are common in this region" common in steep/mountainous regions. As for the soil structure, the author should be, once more, more careful and avoid saying "this process is like this for the whole QTP". A plateau presents by definitions some areas with flat or gentle slope that does not trigger avalanches.*

Reply: Thank you for pointing this out. We have revised it accordingly and have checked the full text for rigour.

*Comment 11*

*L288: the fundamental frequency of what? The frequency of the sinusoidal signal used to establish this theory I guess.*

Reply: Yes, it is the angular frequency ($\omega = \frac{2\pi}{86400}$).

*Comment 12*

*L295: why not directly give the values rather than give how they compare to 0, it would be more informative. Is it different for each contour band?*

Reply: Yes it is different, the topography of many places on the QTP is undulated. Our calculation unit is based on the contour band, the bottom boundary of each contour band is not at the same altitude and maintains a different constant temperature under the influence of geothermal and climatic forcing. The higher the altitude, the lower the bottom temperature.

**Comment 13**

**L359: I would not use "considerably" if you don't give numbers to support that it is considerable.**

**L387: "Owing to" -> because of**

Reply: Thank you for your suggestions, these have been amended accordingly.

**Comment 14**

**L412-413: "During the freezing period (December–March), the liquid water content of the upper layer first decreased owing to the decrease in temperature". I am not sure I fully understand. The freezing process converts liquid water in ice and once freezing is completed, temperature can decrease again. Is that what is implied here? Because it is different from saying that because the liquid wc decrease, the temperature decrease.**

Reply: We apologize for the misunderstanding; here we simply wanted to describe the reduction of the water content in the soil profile layer by layer as the atmospheric temperature decreases. To reduce ambiguity, we rewrote this sentence as:

"After entering the freeze–thaw period, the water in the soil started to freeze from the top layer to the bottom as the atmospheric temperature dropped, thereby decreasing the liquid water content layer by layer until February" (Lines 417–419).

*Comment 15*

*L415: "melt" -> Thaw, unless you say "the ice in the upper layer began to melt".*

*Same L416*

*L455-457: no need for decimal number in the percentage.*

Reply: Thank you for your suggestions. these have been amended accordingly.

*Comment 16*

*L478: "observed", earlier in the review processed I pointed out that on Fig 10 you show differences between QTP and COR but without the reference point given by some observations and I remember saying "different does not necessarily implies better". The answer was to say that it was not the point of the figure to compare with observations. But now the paragraph presenting the figure ends up mentioning that the QTP version shows a feature that is "observed". So either there are observations and there should be on the graph or there are not. Or is it some kind of general statement ? Please clarify this point.*

Reply: We thank the reviewer for bringing up this point, to clarify, this is a general statement referring to a slow flow reduction process which is in accordance with the observed flow changes in Figure 10 (original Fig. 9). To reduce ambiguity, we added the definition of "tailing" process (a slow flow reduction process consistent with observed flows, Lines 458-459) and rewrote this sentence as follows:

"Water in the WEP–QTP model had more time to complete groundwater and river recharge, and thus exhibited a better 'tailing' process than that in the WEP–COR model (Fig. 10)" (Lines 490–492).

*Comment 17*

*L484: "In cold regions, climate change affects high and low flows in different ways" Which flow are we talking about here? River flow I assume that because you cite Song et al., 2021 which seems to talk about river runoff? If so precise it in the sentence.*

Reply: We apologize to the reviewer for not clarifying this. This refers to river flow. We have amended the sentence accordingly to improve clarity:

"In cold regions, climate change affects high and low river flows in different ways." (Line 497).

*Comment 18*

*L489-490: "Our study demonstrates that the soil–gravel layer structure on the QTP is different from the soil structure in other cold regions."*

*This is not exact, you assume it is different based on other studies and then you demonstrate its impact on the cryo-hydrology with your model.*

Reply: Thank you for pointing this out and we have amended this sentence to: "When simulating the water cycle process in the QTP, it is not appropriate to simply generalize the active layer as a homogeneous soil structure and ignore the influence of g-layers beneath the soil-layers. This study shows that..." (Lines 502–503).

*Comment 19*

*L509: I would also remove decimal numbers to the % values*

Reply: Thank you for your suggestion and we have amended this accordingly.

*Comment 20*

*L519: "the unique geological structure and climatic characteristics of the QTP". This is again a statement that is not careful enough "the widely spread geological structure" would be a bit more reasonable. Even though this structure might be often present, it still makes no sense to claim that it covers 100% of the QTP.*

Reply: Thank you for your suggestion. We have revised it accordingly and have checked the full text for rigour.

**Reviewer 2:**

*The authors made substantial improvement to the paper. The WEP–QTP model seems to be an effective tool to simulate water-heat balances in the dualistic soil–gravel structure in QTP. I recommend the paper could be acceptable after minor revision.*

*I understand this study improved the hydrological model by considering freeze–thaw processes in the dualistic soil–gravel structure, but the authors should address uncertainties regarding the model structure and the newly-introduced parameters. For example, how the parameters or the degree-day factor and the critical temperature were prescribed or estimated in the modeling. The authors should give a brief discussion on the uncertainties.*

Dear Reviewer:

We are glad that our improvements to this paper have been recognized by you.

How to estimate the parameters involved in the new formula has been introduced in the parameter description following the formula.

The new parameters added due to model improvement, including the degree-day factor, the critical temperature, and the critical value of non-heavy versus heavy rain periods, are all sensitive parameters for the improved model and have clear physical significance. The degree-day factor and critical temperature values were estimated by the modelling of snow thickness, while the critical value of non-heavy versus heavy rain periods was determined by the parametric calibration of the flow process.

We made the following supplements in section 3.1:

"The new sensitive parameters for the model improvement included the degree-day factor of snow, the critical temperature of snow and the critical value of non-heavy versus heavy rain periods, where the degree-day factor and the critical temperature of snow were estimated by the modelling of snow thickness, and the critical value of non-heavy versus heavy rain periods was determined by the

parametric calibration of the flow process. The values of these parameters were: 4 mm/[°C·day], −1 °C, 15 mm/day respectively." (Lines 356–360).

Thank you for your advice and suggestions, we look forward to hearing from you and would be happy to make further changes, if required.

---

## Author Response (AR4)

*Dear Authors,*

*Thank you for considering all the comments from two reviewers. The paper starts to shape up. Congratulations! My only concern is the less comprehensive discussion about existing cold region hydrological modeling studies on the TP. Most referred models in the Introduction are dated back to twenty years ago. I'd like to suggest to add more current new development in this field, including but not limited to the models of GBHM, WEB-DHM, FLEXG, FLEX-Topo-FS, and also the improved SWAT in cold regions.*

*I am happy and looking forward to receiving your revised MS.*

*Best regards,*

*Hongkai*

Dear Editor:

We are grateful to you and the reviewers for your thoughtful suggestions and insights; the manuscript has greatly benefited from them. We apologize for not adding current new developments in the field to the introduction in the previous rounds of revisions.

As per your suggestion, we have supplemented the introduction with cold region hydrological modeling studies as follows:

Change from:

"The existing hydrological models for cold regions, such as the SHAWDHM model (Zhang et al., 2013), GEOtop model (Rigon et al., 2006), cold regions hydrological model (CRHM) (Pomeroy et al., 2007), and variable infiltration capacity (VIC) model (Cherkauer and Lettenmaier, 2003), consider the processes of water and heat transport in the soil and can simulate water cycling in cold regions to a certain extent. However, these models define the simulated object of the water–heat coupled transport process as a homogeneous medium, while overlooking the stratified soil structure of the QTP. By calibrating certain parameters, the effect of gravel on water and heat transfer can be hidden to some extent, and the simulation effect can be improved. However, some errors remain in the simulation, and it is difficult to objectively reflect the influence of the stratified geological structure on water–heat transport and the hydrological cycle."

To:

"Many hydrological models have been applied to the QTP, including those developed specifically for water cycle processes in cold regions, such as the CRHM (Zhou et al., 2014), WaSiM (Sun et al., 2020), GEOtop (Pan et al., 2016) and DWHC models (Chen et al., 2018). As these models were constructed for cold regions from the outset, modeling soil freeze-thaw processes and accumulation and melting processes of snow and glaciers is detailed and based on physical mechanisms. However, these models require more input parameters and are generally suitable for small catchments. Some models are improved for the characteristics of cold regions based on non-cold hydrological models, such as the improved SWAT (Sun et al., 2013), VIC (Cuo et al., 2015), HBV (Bergström et al., 2015), WEB-DHM (Wang et al., 2010), and GBEHM models (Gao et al., 2018), amongst others. These models exhibited improved performance in cold regions by using heat transfer models or temperature-index models to simulate the freeze-thaw process of soil and melting process of snow (Ala-Aho et al., 2021; Gao et al., 2021). A few studies have also coupled soil freeze-thaw processes or accumulation and melting processes of snow and glaciers

with conceptual hydrological models, such as the FLEX-Topo-FS (Gao et al., 2022) and FLEX-SG models (Gao et al., 2020), based on the perceptual model and FLEX-Topo model (Savenije., 2010), the flexible modeling framework of which can improve the performance of the model in information-poor cold regions while avoiding over-parameterization. However, these models above generally define the simulated object of the water–heat coupled transport process as a homogeneous medium and ignore the stratified soil structure when applied to the QTP. Further, the effect of gravel on water and heat transfer can be hidden to some extent by calibrating certain parameters. The bias in the simulation of interlayer water and heat transport due to the differences in the hydrodynamic and thermodynamic properties between soil and gravel remain, making it difficult for these models to objectively reflect the hydrological cycle processes under geologically stratified structural conditions in the QTP."

Thank you for your consideration. We look forward to hearing from you.

Sincerely,

Pengxiang Wang